# Lifelong Audio-video Masked Autoencoder with Forget-robust Localized Alignments

## Abstract

We present a lifelong audio-video masked autoencoder that continually learns the multimodal representations from a video stream containing audio-video pairs, while its distribution continually shifts over time. Specifically, we propose two novel ideas to tackle the problem: *(1) Localized Alignment*: We introduce a small trainable multimodal encoder that predicts the audio and video tokens that are well-aligned with each other. This allows the model to learn only the highly correlated audiovisual patches with accurate multimodal relationships. *(2) Forget-robust multimodal patch selection*: We compare the relative importance of each audio-video patch between the current and past data pair to mitigate unintended drift of the previously learned audio-video representations. Our proposed method, **FLAVA** (**F**orget-robust **L**ocalized **A**udio-**V**ideo **A**lignment), therefore, captures the complex relationships between the audio and video modalities during training on a sequence of pre-training tasks while alleviating the forgetting of learned audiovisual correlations. Our experiments validate that FLAVA outperforms the state-of-the-art continual learning methods on several benchmark datasets under continual audio-video representation learning scenarios.

## 1 Introduction

Multimodal learning is an important problem for various real-world applications, as many realistic data types are inherently multimodal, consisting of multiple modalities, such as *text-image* (Liao et al., 2022; Lee et al., 2023), *text-video* (Li et al., 2018; Villegas et al., 2022; Hu et al., 2022b), and *audio-video* (Korbar et al., 2018; Xiao et al., 2020) pairs. While most language-vision learning (Li et al., 2020; Yan et al., 2023; Liu et al., 2023) require alignments across the modalities via human-annotated descriptions/explanations, audiovisual domain (Zhou et al., 2019; Gong et al., 2023) has a unique and practical advantage that it does not require additional labels as most videos naturally come with accompanying audios. Thanks to this property, audio-video multimodal learning models can leverage web-scale raw videos (e.g., YouTube, TikTok, Instagram, etc.) for training with minimal human efforts in data preprocessing, and have achieved impressive success in learning effective audio-video multimodal representations (Tang et al., 2022; Huang et al., 2022a; Lin et al., 2023).

Despite the success, most audio-video models still struggle to learn from real-world data since they assume that the distribution of incoming multimodal data is static. However, in real-world scenarios, the model should handle a **dynamic shift of audio-visual data distribution** when training on videos, as the agent's surroundings can continuously change over time, with past environments no longer relevant. For example, in Figure 1, we illustrate a scenario where people are conversing outdoors. Initially, the data includes human voices and visuals of human interaction (yellow). However, as a volcanic eruption takes place, the data shifts to erupting sounds and mountain visuals (red).

Learning audio-video data with continuously changing semantic categories is a nontrivial problem due to two critical challenges: *1) sparse spatiotemporal correlation between the audio-video pairs*, and *2) representational forgetting of audio-video relationships*. First, we find that only a few objects/regions (i.e., sound sources) in a video are strongly correlated with audio and vice versa (Figure 2 (b)). Audio-video representation models also suffer from the problem of forgetting not only the representations for each domain (audio and video), but also their correlations. For instance, the model may initially learn the correct correlation in the audio-video data pair of cars' engine sound (Figure 2 (b)), but

Figure 1: **Example of data distribution shift.** Audio-visual data distribution dynamically shifts after an event.

after learning on a sequence of tasks[1], it forgets the learned cross-modal correlation and highlights inaccurate regions: the middle caption in the video and an unnecessarily large region in the audio spectrogram (Figure 2 (c)). Note that this forgetting becomes more severe for multimodal models since the modalities are sparsely intertwined, as discussed above, and the unintended knowledge shift in an audio-video relationship captured via inter-modal attention could make the model misunderstand the environments/tasks (Please see Figure 3).

To overcome these challenges in learning multiple audio-video representations continuously, we propose *FLAVA (Forget-robust Localized Audio-Video Alignments)*, a novel method that dynamically harnesses audio-video attention maps generated by a small trainable multimodal encoder to capture interrelated audio-video patches. Furthermore, our method compares attention maps from the ongoing task with those from previous time steps to identify forget-robust patches in order to preserve accurate audio-video correlation on previous semantic concepts while training on new data. As a result, we can visually see that the past audio-video semantics are sustained in Figure 2 (d). To the best of our knowledge, this is the first work that addresses a continual representation learning problem for audio-video tasks and identifies the crucial challenges that the new problem has. We validate our approach on VG-GSound and AudioSet datasets against recent audio-video representation learning methods equipped with recent continual learning approaches. Our *FLAVA* outperforms such strong baselines by 1.52%p, 1.80%p, and 0.31%p in accuracy on the downstream audio-to-video and video-to-audio zero-shot retrieval and audiovisual classification task.

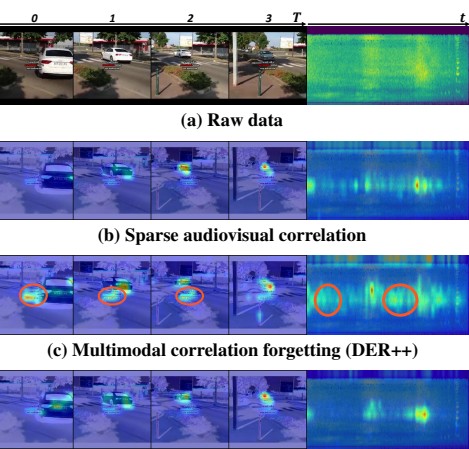

(a) Raw data

(b) Sparse audiovisual correlation

(c) Multimodal correlation forgetting (DER++)

(d) Multimodal correlation forgetting (Ours)

Figure 2: **Visualization of cross-attention maps.**: **(a)** presents raw data pair. **(b)** shows cross-attention maps at the end of the corresponding task. Subsequently, we pre-train the model with a series of tasks after **(b)**. While DER++ attends to entirely different parts (orange circle) in **(c)**, our method maintains consistent attention as shown in **(d)**. More examples are in Figure 17.

We summarize our contributions threefold:

- We introduce a practical problem of audio-video representation learning on continuously changing its data distributions, which poses new critical challenges: *sparse spatiotemporal correlation between the audio-video pairs* and *representational forgetting of audio-video relationships*.

- We propose a novel method, **F**orget-robust **L**ocalized **A**udio-**V**ideo **A**lignments (**FLAVA**), to adaptively capture sparse audio-video attention to learn accurate audio-video relationships while mitigating forgetting from previously learned relationships without requiring task identification.

- We demonstrate the efficacy of our proposed method on multiple retrieval and classification tasks against strong baselines, and provide extensive in-depth analyses with visualizations.

## 2 RELATED WORK

**Audiovisual understanding**    Self-supervised learning on audiovisual data aims to learn transferable representations that can be applied to a variety of audio-image/video downstream tasks, including action recognition/event classification (Nagrani et al., 2021; Lee et al., 2021), sounding object

---

[1]We use *task* and *category* interchangeably in the manuscript.

localization (Qian et al., 2020; Hu et al., 2022a; Liu et al., 2022), and multimodal retrieval (Gong et al., 2023; Huang et al., 2022a). Inspired by the success of Masked AutoEncoders (MAE) in visual pre-training (He et al., 2022), recent audiovisual representation learning adopts masked modeling for comprehending audiovisual semantics (Tang et al., 2022; Gong et al., 2023). TVLT (Tang et al., 2022) adopts the MAE structure and incorporates audiovisual matching to predict whether audio and visual data originated from the same video. CAV (Gong et al., 2023) introduces an efficient way of combining the MAE with audiovisual contrastive learning, which pulls matching audiovisual pairs closer and pushes non-matching pairs apart. Their methods assume that the distribution of the input data is fixed and does not shift during training. However, in the real world, a machine/agent will continuously encounter new (i.e., changing distribution) audio-video tasks/semantics. If not well managed, the methods will suffer severe performance degradation if they encounter the aforementioned shift in multimodal continual learning, a challenging and realistic scenario for multimodal learning.

**Multimodal continual learning** Continual learning (CL) (Kirkpatrick et al., 2016; Rebuffi et al., 2017; Ahn et al., 2019) refers to a learning paradigm in which a model learns an unlimited number of different tasks/domains in a sequential manner. It aims to continuously adapt to new tasks while preserving or improving previously learned knowledge/skills, which is crucial in deploying an AI for real-world usage. A number of works have addressed supervised learning for vision tasks (Zenke et al., 2017; Yoon et al., 2018; Lee et al., 2020), and very recently, a few approaches have been explored continual learning with self-supervised learning (Madaan et al., 2022; Cossu et al., 2022; Fini et al., 2022; Yoon et al., 2023), and multimodal learning (Yan et al., 2022; Pian et al., 2023). IncCLIP (Yan et al., 2022) introduced the technique which employs a generative model to make pseudo-negative text data for self-supervised image-text continual learning. AV-CIL (Pian et al., 2023) tackled the problem of supervised continual learning for audio-video tasks. However, training these models requires dense human annotations, such as text or audiovisual labels. They also require task boundary information to know when new tasks are introduced during continual learning. On the other hand, our proposed FLAVA focuses on lifelong pre-training of audio-video multimodal models without any human-effort labels or task boundary information.

## 3 CONTINUAL AUDIO-VIDEO REPRESENTATION LEARNING

### 3.1 PROBLEM STATEMENT

To tackle real-world multimodal continual learning scenarios where the task/category of surrounding audiovisual information continuously changes over time, we introduce a learning paradigm that aims to train a model on a sequence of $\mathcal{T}$ disjoint unsupervised audio-video datasets $\mathcal{D} = \{\mathcal{D}_i\}_{i=1}^{\mathcal{T}}$. Note that we assume a task-free (or task-agnostic) setup (Aljundi et al., 2019b) where the model performs the pre-training and inference without the explicit knowledge of task boundaries, which is challenging yet realistic because the model does not need any human guidance. For training the $i$-th task, the model iteratively samples $B$ audio-video pairs $(\mathcal{X}_a^i, \mathcal{X}_v^i) \sim \mathcal{D}_i$[2]. Here, $\mathcal{X}_a \in \mathbb{R}^{B \times t \times f}$ represents the audio spectrogram with time ($t$) and frequency ($f$) dimension, and the corresponding video clip $\mathcal{X}_v \in \mathbb{R}^{B \times T \times C \times H \times W}$ consists of $T$ successive frames, where $C$, $H$, and $W$ indicate the dimension of its channel, height, and width, respectively. Let $\boldsymbol{x}_a \in \mathbb{R}^{B \times M \times p \times p}$ and $\boldsymbol{x}_v \in \mathbb{R}^{B \times N \times p \times p}$ be patches of audio and video inputs with the patch size of $p$, where $M = |t/p| \cdot |f/p|$ and $N = |T| \cdot |H/p| \cdot |W/p|$. We obtain $D$-dimensional embedding patches $\boldsymbol{a}$ and $\boldsymbol{v}$ given the audio-video mini-batch as follows:

$$\begin{aligned}
\boldsymbol{x}_a &= \texttt{patchfy}\left(\mathcal{X}_a, p\right), \quad \boldsymbol{x}_v = \texttt{patchfy}\left(\mathcal{X}_v, p\right), \\
\boldsymbol{a} &= \texttt{Conv2d}\left(\boldsymbol{x}_a, \boldsymbol{w}\right), \quad \boldsymbol{v} = \texttt{Conv2d}\left(\boldsymbol{x}_v, \boldsymbol{w}\right),
\end{aligned} \quad (1)$$

where $\boldsymbol{w}$ denotes the weights of a convolutional layer, $\boldsymbol{a} \in \mathbb{R}^{B \times M \times D}$, and $\boldsymbol{v} \in \mathbb{R}^{B \times N \times D}$. We then pretrain the model $f_{\theta, i-1}$, which is pre-trained on consecutive datasets $\{\mathcal{D}_j\}_{j=1}^{i-1}$, with $(\boldsymbol{a}, \boldsymbol{v})$ in order to learn task-specific representations from $\mathcal{D}_i$. Following the recent works on audio-visual representation learning (Gong et al., 2023; Huang et al., 2022a), we adopt two loss terms, *reconstruction loss* for masked inputs and *masked contrastive loss*, denoted as $\ell^r$ and $\ell^c$, respectively. In the end, we update the model weights on audio-video embeddings $(\boldsymbol{a}, \boldsymbol{v})$ by minimizing the

---

[2]For the rest of the paper, unless otherwise stated, we omit the task index for brevity.

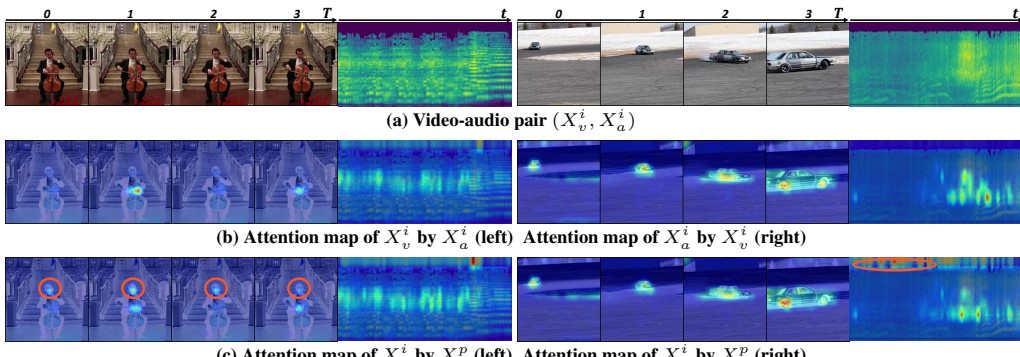

Figure 3: **Cross-attention maps obtained by matched or unmatched data pairs. (a)** Visualization of raw data pairs. **(b)** Cross-attention maps when the pairs are matched. **(c)** Cross-attention with past audio and video (unmatched). We highlight inaccurate attention in orange circles. Please see Figure 16 for more examples.

following objective: $\mathcal{L} = \ell^r + \lambda\ell^c$, where $\lambda$ denotes a hyperparameter for balancing the two losses. The detailed mathematical expressions of the two loss functions are explicated in Appendix C.

## 3.2 Forgetting of the Alignment Between Audio-Visual Modalities

Based on our findings in Figure 2, we raise two challenges in continual audio-video representation learning: *sparse spatiotemporal correlation* and *representational forgetting of audio-video relationships*. In this section, we perform further investigations to gain a better insight into the root cause of the forgetting of audio-video multimodal representation. Let $(X_a, X_v)$ be an audio-video pair of the current task, and $(X_a^p, X_v^p)$ be from the previous time steps. We visualize the cross-attention maps of $(X_a, X_v)$ in Figure 3 (b) and compare it with the cross-attention of $(X_a, X_v^p)$ (Figure 3 (c) Left) and $(X_a^p, X_v)$ (Figure 3 (c) Right). In both modalities, the cross-attention maps with the past data often concentrate on misleading locations when the past data is largely different from the current task. For instance, if the current and past videos both describe human activities but with different audio sources (people participating in sports, playing cello), cross-attention maps associated with past data tend to focus on objects that are not concerned with the current task (Figure 3 (c) Left). This suggests that the model may incorrectly learn audio-video relationships for new data distribution depending on learned multimodal knowledge, which makes the learned audio-video correlation drift to a spurious correlation. Moreover, training a model on the current task potentially results in losing the past multimodal alignment, since previous audio-video correlation is susceptible to being overwritten by the spurious audio-video correlation from the current task.

## 4 Lifelong Audio-Video Masked Autoencoder with FLAVA

To overcome critical challenges in earlier sections, we introduce a novel lifelong audio-video masked autoencoder, dubbed ***Forget-robust Localized Audio-Video Alignments (FLAVA)***. We introduce a lightweight trainable audio-video matching (AVM) module to direct the model to focus on locally aligned audio-visual regions (§4.1). Next, we propose the selection process to identify forget-robust patches (§4.2). Finally, we introduce our selective and forget-robust continual audio-video pre-training framework (§4.3). The overview of our framework is illustrated in Figure 4.

### 4.1 Localized Audio-Video Alignment with Audio-video Matching Module

In this section, we explain our approach to obtain the importance score, which guides the model to focus on locally aligned (i.e. highly correlated) audio-video patches. Let $(\tilde{\boldsymbol{o}}_a, \tilde{\boldsymbol{o}}_v)$ be an audio-video representation by audio/video encoders and $\mu(\boldsymbol{q}, \boldsymbol{k}; \boldsymbol{\theta})$ be an Audio-Video Matching module (AVM), parameterized with $\boldsymbol{\theta}$. We pass them through our AVM module to measure the cross-attention between different modality embeddings. For computing an audio-to-video attention map, $\tilde{\boldsymbol{o}}_a$ is treated as a query, and $\tilde{\boldsymbol{o}}_v$ acts as a key and value, and for video-to-audio attention map, they operate oppositely. This process generates keys and queries for $\tilde{\boldsymbol{o}}_a$ and $\tilde{\boldsymbol{o}}_v$, denoted as $(\boldsymbol{k}_a, \boldsymbol{q}_a)$, and $(\boldsymbol{k}_v, \boldsymbol{q}_v)$,

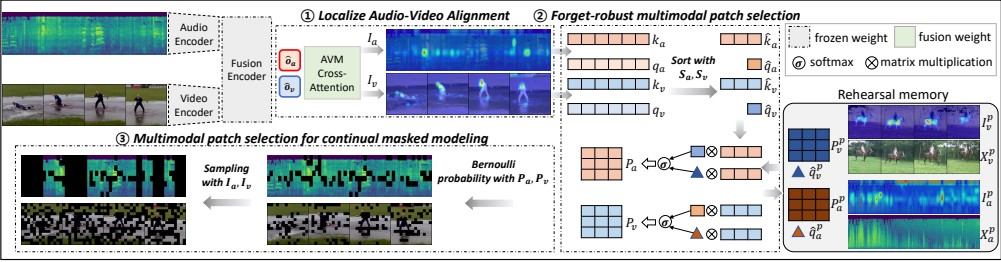

Figure 4: **Overview of our approach.** Our method harnesses cross-modal attention maps from the AVM module to compute importance scores in order to identify highly correlated patches (**Localized Audio-Video Alignment**). Comparing the attention maps created by the current queries with those generated by past queries, we generate a pruning probability matrix that compares the relative importance of each patch between the current task and past tasks (**Forget-robust multimodal patch selection**). Finally, we select patches to continually pre-train the model (**Multimodal patch selection for continual masked modeling**).

respectively. Therefore, we can measure cross-attention for each modality as follows:

$$
\begin{aligned}
\boldsymbol{A}_a &= \mathtt{Softmax}\left(\mu_{\boldsymbol{\theta}}(\boldsymbol{q}_v, \boldsymbol{k}_a)\right) = \mathtt{Softmax}\left(\boldsymbol{q}_v \boldsymbol{k}_a^\top / \beta * \sqrt{d}\right), \\
\boldsymbol{A}_v &= \mathtt{Softmax}\left(\mu_{\boldsymbol{\theta}}(\boldsymbol{q}_a, \boldsymbol{k}_v)\right) = \mathtt{Softmax}\left(\boldsymbol{q}_a \boldsymbol{k}_v^\top / \beta * \sqrt{d}\right),
\end{aligned}
\tag{2}
$$

where $\beta$ is the temperature hyperparameter and $\boldsymbol{A}_a \in \mathbb{R}^{B \times H \times N \times M}$, $\boldsymbol{A}_v \in \mathbb{R}^{B \times H \times M \times N}$. Here, $H$ denotes the number of heads, and the detailed structure of AVM module is illustrated in Appendix D.

These audio and video cross-attention maps associated with each other imply the multimodal significance; that is, the highlighted audio/video embedding queries (i.e., input patches) from the attention indicate that they are more closely associated with the paired modality data than other embeddings. Then, we can compute the importance score matrices, $\boldsymbol{I}_a \in \mathbb{R}^{B \times M}$, $\boldsymbol{I}_v \in \mathbb{R}^{B \times N}$ as follows:

$$
\boldsymbol{I}_a = \mathtt{MeanPool}\left(\boldsymbol{A}_a\right), \ \boldsymbol{I}_v = \mathtt{MeanPool}\left(\boldsymbol{A}_v\right),
\tag{3}
$$

These matrices illustrate the correlation between each patch and its corresponding modality pair, where higher values in the matrices signify the greater significance of the corresponding patches.

## 4.2 FORGET-ROBUST MULTIMODAL PATCH SELECTION

Next, we identify the most informative and forget-robust subset from audio-video patches to use them in learning strongly correlated audio-video representations in a continual learning process, which requires a careful balance between retaining previous knowledge and adapting new multimodal correlations. Thus, we suggest exploiting attention maps activated by data from the current and previous timesteps to understand the relative importance of each patch to the current and past tasks. The process of calculating current and past attention maps is summarized in Equation 4. We aim to select $\kappa_a$ audio and $\kappa_v$ video patches, where $\kappa_a = M \cdot \rho_a$ and $\kappa_v = N \cdot \rho_v$. Here, $\rho_a$ and $\rho_v$ represent sampling ratios of audio and video data, respectively. First, we extract locally aligned patches using sorted indices $\boldsymbol{S}_a = \mathtt{argsort}(\boldsymbol{I}_a)$ and $\boldsymbol{S}_v = \mathtt{argsort}(\boldsymbol{I}_a)$. With these sorted indices, we collect discriminative keys $(\widehat{\boldsymbol{k}}_a, \widehat{\boldsymbol{k}}_v)$ (Equation 4 line 1) and compute weighted mean queries $(\widehat{\boldsymbol{q}}_a, \widehat{\boldsymbol{q}}_v)$ from the AVM module using $\boldsymbol{I}_a$, and $\boldsymbol{I}_v$ respectively as the input (Equation 4 line 2-3). Subsequently, we utilize the discriminative keys and queries to compute new attention maps, denoted as $\widehat{\boldsymbol{A}}_a = \mu_{\boldsymbol{\theta}}(\widehat{\boldsymbol{q}}_v, \widehat{\boldsymbol{k}}_a, \beta=1) \in \mathbb{R}^{B \times H \times \kappa_a}$, $\widehat{\boldsymbol{A}}_v = \mu_{\boldsymbol{\theta}}(\widehat{\boldsymbol{q}}_a, \widehat{\boldsymbol{k}}_v, \beta=1) \in \mathbb{R}^{B \times H \times \kappa_v}$. In order to assess the relative importance of current data and past data, we further compute past-data-induced attention maps $\widehat{\boldsymbol{A}}_a^p = \mu_{\boldsymbol{\theta}}(\widehat{\boldsymbol{q}}_v^p, \widehat{\boldsymbol{k}}_a, \beta=1)$, $\widehat{\boldsymbol{A}}_v^p = \mu_{\boldsymbol{\theta}}(\widehat{\boldsymbol{q}}_a^p, \widehat{\boldsymbol{k}}_v, \beta=1)$, by combining the past discriminative queries $\widehat{\boldsymbol{q}}_a^p$, $\widehat{\boldsymbol{q}}_v^p$ from the rehearsal memory and the current discriminative keys (Equation 4 line 4). In the end, we obtain the attention reflecting inter-task cross-modality for continual representation learning.

$$
\begin{aligned}
\widehat{\boldsymbol{k}}_n[i,:,j] &= \boldsymbol{k}_n[i,:,\boldsymbol{S}_n[i,j]], \ \boldsymbol{I}_n^s[i,j] = \boldsymbol{I}_n[i,\boldsymbol{S}_n[i,j]], \\
\widehat{\boldsymbol{q}}_m[i,:,k] &= \boldsymbol{q}_m[i,:,\boldsymbol{S}_m[i,k]], \ i=1,\dots,B, \ j=1,\dots,\kappa_n, \ k=1,\dots,\kappa_m, \\
\widehat{\boldsymbol{q}}_m &= \mathtt{MeanPool}\left(\widehat{\boldsymbol{q}}_m, \mathtt{weight}=\boldsymbol{I}_m^s\right), \\
\widehat{\boldsymbol{A}}_n &= \mu_{\boldsymbol{\theta}}(\widehat{\boldsymbol{q}}_m, \widehat{\boldsymbol{k}}_n, \beta=1), \ \ \widehat{\boldsymbol{A}}_n^p = \mu_{\boldsymbol{\theta}}(\widehat{\boldsymbol{q}}_m^p, \widehat{\boldsymbol{k}}_n, \beta=1), \ \text{ where } \ (n,m) \in \{(a,v),(v,a)\}.
\end{aligned}
\tag{4}
$$

For the purpose of assessing the relative importance, we concatenate the audio $(\widehat{\boldsymbol{A}}_a, \ \widehat{\boldsymbol{A}}_a^p)$ and video attention maps $(\widehat{\boldsymbol{A}}_v, \ \widehat{\boldsymbol{A}}_v^p)$, respectively. Subsequently, we apply Softmax normalization for each index, resulting in pruning probability matrices $\boldsymbol{P}_a$ and $\boldsymbol{P}_v$. Each index value of $\boldsymbol{P}_a$ and $\boldsymbol{P}_v$ approaches one if the corresponding patch exhibits a higher correlation with past data. Consequently, indices with higher values would be more likely to be candidates for pruning. The above procedure is expressed as follows:

$$\boldsymbol{P}_n = \texttt{MeanPool}\left(\texttt{Softmax}\left([\widehat{\boldsymbol{A}}_n, \widehat{\boldsymbol{A}}_n^p]\right)\right), \quad \text{where} \quad n \in \{a, v\}. \tag{5}$$

### 4.3 MULTIMODAL PATCH SELECTION FOR CONTINUAL MASKED MODELING

Based on the importance score $\boldsymbol{I}_a$ and pruning probability $\boldsymbol{P}_a$ for audio inputs, now we can find the most helpful audio patches to learn. In order to preserve the local correlation among audio patches by temporal continuity, we select audio patches in time chunks. To measure the audio patch importance time-wise, we reshape the audio importance score of $i$-th sample $\boldsymbol{I}_{a,i}$ into a time-frequency shape of $|t/p| \times |f/p|$. The reshaped map is then averaged along the frequency dimension, resulting in $\boldsymbol{I}_{a,i}^t \in \mathbb{R}^{|t/p|}$. For $\boldsymbol{P}_{a,i}$, we first apply Bernoulli probability distribution to generate $\boldsymbol{F}_{a,i}$, which is an indicator matrix for pruning. Then, we reshape $\boldsymbol{F}_{a,i}$ and sum over frequency dimension to obtain the number of pruning indices in the temporal domain, denoted as $\boldsymbol{F}_{a,i}^t \in \mathbb{R}^{|t/p|}$.

We subsequently process $\boldsymbol{I}_{a,i}^t$ to estimate the audio patch importance within time chunks. To do so, we apply average pooling to $\boldsymbol{I}_{a,i}^t$ with a kernel size denoted as $\boldsymbol{L}_c$, where $\boldsymbol{L}_c$ is set to a default value of 4. This operation yields $\boldsymbol{I}_{a,i}^c$, which indicates the importance of each time chunk to select time chunks based on their importance score, we employ multinomial probability distribution with weights from $\boldsymbol{I}_{a,i}^c$ to extract time chunk indices $\boldsymbol{t}_{select}$. We then iterate through $\boldsymbol{t}_{select}$ to select time chunks using $\boldsymbol{I}_{a,i}^c$ and $\boldsymbol{F}_{a,i}^t$ until the number of selected audio patches reaches $\kappa_a$. By iterating the above process through all the samples, we generate forget-robust audio patch indices in the minibatch, denoted as $\tilde{\boldsymbol{S}}_a \in \mathbb{R}^{B \times M}$. The details of audio patch selection are summarized in Algorithm 2.

Next, we utilize the importance score $\boldsymbol{I}_v$ and pruning probability $\boldsymbol{P}_v$ for video inputs in an effort to generate forget-robust video patch indices. In contrast to the aforementioned audio patch selection, the video patch selection process does not need additional constraints. Instead, we employ a straightforward approach by initializing zero for each element in $\boldsymbol{I}_v$ corresponding to indices with a True value in $\boldsymbol{F}_v$ based on a Bernoulli distribution defined by $\boldsymbol{P}_v$, to create $\tilde{\boldsymbol{I}}_v$. Then, we subsequently apply a multinomial probability distribution with weights from $\tilde{\boldsymbol{I}}_v$, and finally obtain $\tilde{\boldsymbol{S}}_v \in \mathbb{R}^{B \times N}$, which is forget-robust video patch indices. The process can be expressed as follows:

$$\tilde{\boldsymbol{I}}_v[i,j] = \begin{cases} 0 & \text{if } \boldsymbol{F}_v[i,j] \\ \boldsymbol{I}_v[i,j] & \text{otherwise} \end{cases}, \ i = 1, \dots, B, \ j = 1, \dots, N,$$

$$\tilde{\boldsymbol{S}}_v = \texttt{Multinomial}\left(\tilde{\boldsymbol{I}}_v, \right), \tag{6}$$

Finally, based on $\tilde{\boldsymbol{S}}_a$, $\tilde{\boldsymbol{S}}_v$, we select $\kappa_a$, $\kappa_v$ of audio, video patches to gather new input $(\widehat{\boldsymbol{x}}_a, \ \widehat{\boldsymbol{x}}_v)$. By substituting $(\boldsymbol{x}_a, \ \boldsymbol{x}_v)$ into $(\widehat{\boldsymbol{x}}_a, \ \widehat{\boldsymbol{x}}_v)$, we enable the model to better focus on learning audio-video relationship with more efficiency. The final patch selection is performed as follows:

$$\widehat{\boldsymbol{x}}_a[i,j] = \boldsymbol{x}_a[i, \tilde{\boldsymbol{S}}_a[i,j]], \ \ \widehat{\boldsymbol{x}}_v[i,j] = \boldsymbol{x}_v[i, \tilde{\boldsymbol{S}}_v[i,k]], \ i = 1, \dots, B, \ j = 1, \dots, \kappa_a, \ k = 1, \dots, \kappa_v, \tag{7}$$

With selected patches, we perform continual pre-training based on the DER++ framework with the penalty loss ($\ell^p$), which encourages the model to maintain the features of the rehearsal memory during future training while mitigating their drifts. Hence, our final pre-training objective is $\mathcal{L} = \ell^r + \lambda\ell^c + \alpha\ell^p$, where $\alpha$ is a hyperparameter for the penalty loss.

Efficient utilization of the rehearsal memory is important in rehearsal-based continual learning, particularly in audio-video continual learning scenarios due to the substantially larger size of videos compared to images. The effective storage of past data can notably augment the diversity of data within the memory. To address this, we propose FLAVA+, an extension of FLAVA, where its memory stores the selected patches instead of raw data. The introduction of FLAVA+ represents a distinct and complementary direction to FLAVA, demonstrating the efficacy of efficient memory utilization.

The proposed patch selection (yellow background) and the weight update process (gray background) are summarized in Algorithm 1.

---

**Algorithm 1** Foget-robust Localized Audio-Video Alignment (FLAVA)

---

**input** Dataset $\{\mathcal{D}_t\}_{t=1}^T$, transformer model $f_\theta$ VAM module $h_\Theta$, rehearsal memory $\mathcal{M}$.

  1: **for** task $\mathcal{T}_t = \mathcal{T}_1, \ldots, \mathcal{T}_T$ **do**
  2:    **for** batch $(\boldsymbol{x}_a, \ \boldsymbol{x}_v) \sim \mathcal{D}_t$ **do**
  3:       $\boldsymbol{k}_a, \boldsymbol{q}_a, \boldsymbol{A}_a, \ \boldsymbol{k}_v, \boldsymbol{q}_v, \boldsymbol{A}_v \leftarrow$ AVM-ATT$(\boldsymbol{x}_a, \ \boldsymbol{x}_v)$                     ▷ AVM Attention in Eq. 10
  4:       $\boldsymbol{I}_a, \ \boldsymbol{I}_v \leftarrow$ IMPORTANCE$(\boldsymbol{A}_a, \ \boldsymbol{A}_v)$                          ▷ Importance measure in Eq. 3
  5:       $\widehat{\boldsymbol{k}}_a, \widehat{\boldsymbol{q}}_a, \ \widehat{\boldsymbol{k}}_v, \widehat{\boldsymbol{q}}_v \leftarrow$ SORT$(\boldsymbol{k}_a, \boldsymbol{q}_a, \boldsymbol{I}_a, \ \boldsymbol{k}_v, \boldsymbol{q}_v, \boldsymbol{I}_v)$      ▷ Core information in Eq. 4
  6:       **##FLAVA##**
  7:       $\boldsymbol{x}_a^p, \boldsymbol{x}_v^p, \widehat{\boldsymbol{q}}_a^p, \widehat{\boldsymbol{q}}_v^p, \boldsymbol{I}_a^p, \boldsymbol{I}_v^p, \boldsymbol{P}_a^p, \boldsymbol{P}_v^p \leftarrow \mathcal{M}$                  ▷ Load past data
  8:       $\boldsymbol{P}_a, \ \boldsymbol{P}_v \leftarrow$ PRUNE-PROB$(\widehat{\boldsymbol{k}}_a, \widehat{\boldsymbol{q}}_v, \widehat{\boldsymbol{q}}_v^p, \ \widehat{\boldsymbol{k}}_v, \widehat{\boldsymbol{q}}_a, \widehat{\boldsymbol{q}}_a^p)$       ▷ Pruning probability in Eq. 4, 5
  9:       $\widehat{\boldsymbol{x}}_a, \widehat{\boldsymbol{x}}_a^p \leftarrow$ AUDIO-SELECT$([\boldsymbol{x}_a, \boldsymbol{x}_a^p], [\boldsymbol{I}_a, \boldsymbol{I}_a^p], [\boldsymbol{P}_a, \boldsymbol{P}_a^p])$   ▷ Audio selection in Alg. 2, Eq. 7
10:       $\widehat{\boldsymbol{x}}_v, \widehat{\boldsymbol{x}}_v^p \leftarrow$ VIDEO-SELECT$([\boldsymbol{x}_v, \boldsymbol{x}_v^p], [\boldsymbol{I}_v, \boldsymbol{I}_v^p], [\boldsymbol{P}_v, \boldsymbol{P}_v^p])$   ▷ Video selection in Eq. 6, 7
11:       $\mathcal{M} \leftarrow \mathcal{M} \cup (\boldsymbol{x}_a, \boldsymbol{x}_v, \widehat{\boldsymbol{q}}_a, \widehat{\boldsymbol{q}}_v, \boldsymbol{I}_a, \boldsymbol{I}_v, \boldsymbol{P}_a, \boldsymbol{P}_v)$    ▷ Rehearsal memory update
12:       **##FLAVA+##**
13:       $\widehat{\boldsymbol{x}}_a^p, \widehat{\boldsymbol{x}}_v^p, \widehat{\boldsymbol{q}}_a^p, \widehat{\boldsymbol{q}}_v^p \leftarrow \mathcal{M}$                             ▷ Load past data
14:       $\boldsymbol{P}_a, \ \boldsymbol{P}_v \leftarrow$ PRUNE-PROB$(\widehat{\boldsymbol{k}}_a, \widehat{\boldsymbol{q}}_v, \widehat{\boldsymbol{q}}_v^p, \ \widehat{\boldsymbol{k}}_v, \widehat{\boldsymbol{q}}_a, \widehat{\boldsymbol{q}}_a^p)$       ▷ Pruning probability in Eq. 4, 5
15:       $\widehat{\boldsymbol{x}}_a \leftarrow$ AUDIO-SELECT$(\boldsymbol{x}_a, \boldsymbol{I}_a, \boldsymbol{P}_a,)$                 ▷ Audio selection in Alg. 2, Eq. 7
16:       $\widehat{\boldsymbol{x}}_v \leftarrow$ VIDEO-SELECT$(\boldsymbol{x}_v, \boldsymbol{I}_v, \boldsymbol{P}_v)$                     ▷ Video selection in Eq. 6, 7
17:       $\mathcal{M} \leftarrow \mathcal{M} \cup (\widehat{\boldsymbol{x}}_a, \widehat{\boldsymbol{x}}_v, \widehat{\boldsymbol{q}}_a, \widehat{\boldsymbol{q}}_v)$                 ▷ Rehearsal memory update
18:       $\theta \leftarrow \theta - \eta \nabla f_\theta \left(([\widehat{\boldsymbol{x}}_a, \widehat{\boldsymbol{x}}_a^p], [\widehat{\boldsymbol{x}}_v, \widehat{\boldsymbol{x}}_v^p])\right)$               ▷ Backbone update
19:       $\Theta \leftarrow \Theta - \eta \nabla h_\Theta \left((\boldsymbol{x}_a, \boldsymbol{x}_v)\right)$                     ▷ AVM update
20:    **end for**
21: **end for**

---

## 5   EXPERIMENTS

In this section, we experimentally validate the effectiveness of our method in task-free lifelong audio-video representation learning scenarios, comparing it against recent methods. To accomplish this, we commence by presenting our experimental setup in §5.1, which encompasses datasets, evaluation methods, evaluation metrics, and baseline methods utilized for our experiments. Then we present the experimental results and conduct a comprehensive analysis of these results in §5.2.

### 5.1   EXPERIMENTAL SETUP

**Evaluation Protocol for Audio-Video Continual Pre-Training**   To test the efficacy of our method on continual audio-video representation learning, we validate our method over VGGSound dataset (Chen et al., 2020) and AudioSet (Gemmeke et al., 2017) dataset. Both datasets contain 10s YouTube videos. For evaluation, we conduct two types of downstream tasks: an audiovisual zero-shot retrieval task and an audiovisual classification task. The details of the evaluation protocol including data split, task dataset statistics, and downstream tasks are explicated in Appendix B.

**Baselines**   To quantitatively assess our method, we test the performance of other task-free continual learning methods in the same setting: ER (Rolnick et al., 2019), MIR (Aljundi et al., 2019a), DER++ (Buzzega et al., 2020), GMED (Jin et al., 2021), CLS-ER (Arani et al., 2022), and LUMP (Madaan et al., 2022). The details of the baseline methods are explicated in Appendix A. Both our method and other baselines utilize reservoir sampling (Vitter, 1985) to sample past instances from the rehearsal memory. The memory randomly samples and replaces the instances during training. All methods employ reservoir sampling for memory updates and maintain an identical number of instances in the memory, except for FLAVA+. FLAVA+ accommodates an increased number of instances based on sampling ratios $(\rho_a, \rho_v)$ to match the memory size of FLAVA. Additionally, we compare our approach with the model continually pre-trained without any other additional method, denoted as *Finetune*, and the model pre-trained with the entire pre-training datasets, denoted as *Multitask*. Each acts as lower-bound and upper-bound in assessing learned representation, respectively.

**Evaluation Metrics**   After each end of pre-training on $\mathcal{D}_t$, we estimate task-specific performances $\{acc_{t,i}\}_{i=1}^t$, where $acc_{t,i}$ denotes the performance of the downstream task associated with $\mathcal{D}_i$

Table 1: Results of audiovisual zero-shot retrieval task on VGGSound and AudioSet. R@K means top-K recall. The best and the second best results are highlighted in **bold** and underline, respectively.

| | Method | VGGSound R@1 $\mathcal{A}\uparrow$ | $\mathcal{F}\downarrow$ | R@5 $\mathcal{A}\uparrow$ | $\mathcal{F}\downarrow$ | R@10 $\mathcal{A}\uparrow$ | $\mathcal{F}\downarrow$ | Avg $\mathcal{A}\uparrow$ | $\mathcal{F}\downarrow$ | AudioSet R@1 $\mathcal{A}\uparrow$ | $\mathcal{F}\downarrow$ | R@5 $\mathcal{A}\uparrow$ | $\mathcal{F}\downarrow$ | R@10 $\mathcal{A}\uparrow$ | $\mathcal{F}\downarrow$ | Avg $\mathcal{A}\uparrow$ | $\mathcal{F}\downarrow$ |
|---|---|---|---|---|---|---|---|---|---|---|---|---|---|---|---|---|---|
| Audio-to-Video | Finetune | 0.98 | 4.16 | 3.75 | 11.98 | 6.17 | 15.35 | 3.63 | 10.50 | 1.48 | 2.90 | 3.84 | 11.34 | 5.41 | 17.83 | 3.58 | 10.69 |
| | ER | 4.09 | 3.66 | 11.66 | 9.17 | 17.78 | 10.20 | 11.18 | 7.68 | 4.94 | 2.97 | 12.33 | 7.46 | 17.60 | 11.17 | 11.62 | 7.20 |
| | MIR | 4.59 | 3.14 | 12.26 | 8.34 | 17.51 | 11.17 | 11.45 | 7.55 | 5.21 | 2.93 | 13.16 | 7.10 | 18.04 | 9.14 | 12.14 | 6.39 |
| | DER++ | 4.03 | 3.62 | 13.74 | 6.31 | 19.79 | 7.11 | 12.52 | 5.68 | 4.51 | 3.75 | 12.15 | 8.42 | 16.85 | 11.86 | 11.17 | 8.01 |
| | GMED | 4.17 | 2.73 | 12.01 | 6.84 | 18.95 | 6.33 | 11.71 | 5.30 | 4.71 | 2.27 | 12.83 | 7.45 | 18.44 | 9.18 | 11.99 | 6.30 |
| | CLS-ER | 4.61 | 3.20 | 14.07 | 6.77 | 19.54 | 8.92 | 12.74 | 6.30 | 4.17 | 4.50 | 11.28 | 11.06 | 16.85 | 12.55 | 10.77 | 9.37 |
| | LUMP | 3.56 | 2.79 | 11.68 | 7.65 | 17.40 | 8.52 | 10.88 | 6.32 | 3.73 | 3.03 | 13.74 | 5.29 | 19.50 | 8.17 | 12.32 | 5.50 |
| | FLAVA (Ours) | 5.34 | 2.04 | 15.04 | 5.20 | 22.10 | 5.90 | 14.16 | 4.38 | 5.22 | 2.26 | 13.09 | 7.95 | 18.75 | 10.65 | 12.35 | 6.95 |
| | FLAVA+ (Ours) | 5.39 | 2.71 | 16.76 | 5.15 | 24.18 | 5.99 | 15.44 | 4.62 | 5.36 | 4.24 | 16.76 | 5.54 | 23.65 | 7.44 | 15.26 | 5.74 |
| | Multitask | 6.45 | – | 20.19 | – | 29.01 | – | 18.55 | – | 8.28 | – | 24.14 | – | 33.74 | – | 22.05 | – |
| Video-to-Audio | Finetune | 1.22 | 4.47 | 4.17 | 11.23 | 6.95 | 14.67 | 4.11 | 10.12 | 1.50 | 3.23 | 4.08 | 10.04 | 6.33 | 14.43 | 3.97 | 9.23 |
| | ER | 3.28 | 3.94 | 11.30 | 8.86 | 16.40 | 11.37 | 10.33 | 8.06 | 3.70 | 4.36 | 10.76 | 10.34 | 15.68 | 15.06 | 10.05 | 9.92 |
| | MIR | 3.54 | 3.47 | 11.82 | 9.11 | 16.69 | 12.90 | 10.68 | 8.49 | 4.26 | 4.59 | 11.29 | 9.87 | 15.97 | 13.73 | 10.51 | 9.40 |
| | DER++ | 3.49 | 3.86 | 13.22 | 7.09 | 19.03 | 9.04 | 11.91 | 6.66 | 4.23 | 4.50 | 11.66 | 10.10 | 16.24 | 13.97 | 10.71 | 9.52 |
| | GMED | 3.71 | 2.61 | 11.87 | 6.46 | 17.20 | 9.57 | 10.93 | 6.21 | 3.99 | 4.42 | 10.65 | 10.39 | 15.41 | 14.78 | 10.02 | 9.86 |
| | CLS-ER | 4.09 | 3.11 | 13.30 | 6.96 | 19.43 | 9.68 | 12.27 | 6.58 | 4.25 | 4.58 | 9.78 | 11.65 | 13.45 | 17.65 | 9.16 | 11.29 |
| | LUMP | 3.24 | 3.30 | 11.02 | 7.55 | 16.91 | 9.13 | 10.39 | 6.66 | 3.13 | 3.91 | 10.60 | 8.63 | 16.02 | 12.26 | 9.92 | 8.27 |
| | FLAVA (Ours) | 5.30 | 2.40 | 15.43 | 4.84 | 21.47 | 6.70 | 14.07 | 4.65 | 4.49 | 3.39 | 12.08 | 9.00 | 17.31 | 12.75 | 11.29 | 8.38 |
| | FLAVA+ (Ours) | 5.86 | 1.56 | 17.21 | 4.09 | 23.53 | 6.02 | 15.53 | 3.89 | 5.48 | 4.06 | 15.65 | 7.13 | 22.29 | 8.92 | 14.47 | 6.70 |
| | Multitask | 6.85 | – | 21.93 | – | 30.63 | – | 19.80 | – | 8.05 | – | 25.81 | – | 35.60 | – | 23.15 | – |

when evaluated with $f_{\theta,t}$. Here, we do not use any task boundary knowledge when estimating the performance. For the evaluation, we adopt two conventional metrics in continual learning: **(1) Average accuracy**($\mathcal{A}$) is the average accuracy across all tasks after the completion of pre-training on $\mathcal{D}_{\mathcal{T}}$. Hence, it is formulated as $\mathcal{A} = \frac{1}{\mathcal{T}} \sum_{i=1}^{\mathcal{T}} acc_{\mathcal{T},i}$. **(2) Average Forgetting**($\mathcal{F}$) is the metric that quantifies the average amount of catastrophic forgetting for each task, measured as the difference between its maximum accuracy and accuracy at the completion of pre-training on $\mathcal{D}_{\mathcal{T}}$, calculated as, $\mathcal{F} = \frac{1}{\mathcal{T}-1} \sum_{i=1}^{\mathcal{T}-1} \max_{t \in \{1,\dots,\mathcal{T}-1\}} (acc_{t,i} - acc_{\mathcal{T},i})$.

## 5.2 Quantitative Analysis for Audio-Video Continual Pre-training

**FLAVA achieves superior Zero-shot Audiovisual Retrieval performance** compared to strong baselines. In order to quantitatively assess the audio-video correlation knowledge learned from the continual pre-training, we conduct an audiovisual zero-shot retrieval task. Table 1 summarizes the audio-to-video and video-to-audio retrieval results for both VGGSound and AudioSet. For all methods, we basically set $2k$ and $5k$ for a rehearsal memory size on VGGSound and AudioSet experiments. For the VGGSound, Both FLAVA and FLAVA+ significantly outperform other baselines, exhibiting substantial enhancements of 1.52%p, 2.80%p and 1.80%p, 3.26%p in average audio-to-video and video-to-audio retrieval scores, respectively. In the AudioSet experiments, FLAVA and FLAVA+ exhibit prominent performance advantages, with 0.03%, 2.94% and 0.58%p, 3.76%p improvements in average audio-to-video and video-to-audio retrieval scores, respectively. FLAVA demonstrates competitive performance compared to LUMP in the audio-to-video retrieval task. It is important to highlight that FLAVA achieves a notably higher R@1 score in the audio-to-video task than LUMP, where the R@1 score increases only when the audio correctly retrieves its video pair. These results imply that our approach not only preserves past audio-video correlation but also enables the model to comprehend the high-level semantics of the audio-video relationship. To conduct a more rigorous investigation, we perform additional experiments using shuffled task orders in Appendix E.

**FLAVA is significantly efficient in terms of GPU Memory and Throughput**. Pre-training on the locally aligned subset of audio-video patches also enhances efficiency during the pre-training phase. In Table 2, we summarize GPU memory occupancy and throughput estimated during the pre-training across different methods. Note that FLAVA consumes significantly less GPU memory than baselines, even surpassing *Finetune* in efficiency. Compared to *DER++*, FLAVA achieves an efficiency gain of 43.59%, further enhancing throughput. Additionally, we emphasize that the additional memory required to store the importance scores ($\boldsymbol{I}_a$, $\boldsymbol{I}_v$) and pruning probabilities ($\boldsymbol{P}_a$, $\boldsymbol{P}_v$) in the rehearsal memory is negligible (+ 0.16 GB) compared to the size of the rehearsal memory itself ($\sim$ 5.47 GB).

Table 2: **Efficiency analysis.** GPU memory occupancy (GPU M.) is measured in GB. Throughput (T.P.) of baselines is measured in sample/sec. Both are estimated in single V100 with batch size of 9.

| METHOD | A→V | | V→A | | GPU M.↓ | T.P. ↑ |
|---|---|---|---|---|---|---|
| | $\mathcal{A}$ ↑ | $\mathcal{F}$ ↓ | $\mathcal{A}$ ↑ | $\mathcal{F}$ ↓ | | |
| Finetune | 3.63 | 10.50 | 4.11 | 10.12 | 18.34 | **31.84** |
| ER | 11.18 | 7.68 | 10.33 | 8.06 | 30.95 | 18.10 |
| MIR | 11.45 | 7.55 | 10.68 | 8.49 | 31.17 | 5.75 |
| DER++ | 12.52 | 5.68 | 11.91 | 6.66 | 30.95 | 17.01 |
| GMED | 11.71 | 5.30 | 10.93 | 6.21 | 32.03 | 5.65 |
| CLS-ER | 12.74 | 6.30 | 12.27 | 6.58 | 32.50 | 16.20 |
| LUMP | 10.88 | 6.32 | 10.39 | 6.66 | 18.36 | 23.44 |
| FLAVA | **14.16** | **4.38** | **14.07** | **4.65** | **17.45** | 17.43 |

Table 3: **Sampling methods.** Retrieval results by various methods of sampling on VGGSound dataset. LAVA: Localized audio video alignment in §4.1, FRS: Forget-robust selection in §4.2.

| Method | LAVA | FRS | A→V | | V→A | | GPU M.↓ | T.P.↑ |
|---|---|---|---|---|---|---|---|---|
| | | | $\mathcal{A}$ ↑ | $\mathcal{F}$ ↓ | $\mathcal{A}$ ↑ | $\mathcal{F}$ ↓ | | |
| Random | — | — | 12.64 | 6.46 | 12.55 | 6.58 | **16.63** | **27.98** |
| MATS | — | — | 12.91 | 6.55 | 12.70 | 6.80 | 21.30 | 21.42 |
| FLAVA | — | — | 12.52 | 5.68 | 11.91 | 6.66 | 30.95 | 14.13 |
| | ✓ | — | 13.44 | 5.50 | 13.27 | 5.94 | 17.48 | 17.45 |
| | — | ✓ | 13.40 | 5.30 | 12.94 | 5.44 | 17.48 | 17.08 |
| | ✓ | ✓ | **14.16** | **4.38** | **14.07** | **4.65** | 17.45 | 17.43 |

**FLAVA outperforms continual learning baselines on audiovisual classification tasks.** We conduct the experiment on audiovisual classification tasks to evaluate the joint audio-video representation from the continual audio-video pretraining. We follow the same rehearsal memory size setting in the retrieval task experiments. As depicted in Table 4, FLAVA and FLAVA+ consistently outperform baselines in both the VGGSound and AudioSet datasets, demonstrating gains of 0.31%p, 0.73%p and 0.28%p, 0.67%p, respectively. Hence, our method excels at acquiring a superior joint audio-video representation compared to other baselines. This implies that

Table 4: VGGSound & AudioSet audiovisual classification task. The best and the second best results are highlighted in **bold** and underline, respectively.

| Method | VGGSound (Acc) | | AudioSet (mAP) | |
|---|---|---|---|---|
| | $\mathcal{A}$ ↑ | $\mathcal{F}$ ↓ | $\mathcal{A}$ ↑ | $\mathcal{F}$ ↓ |
| Finetune | 57.01 (± 0.31) | 2.15 (± 0.51) | 63.98 (± 0.18) | 2.12 (± 0.15) |
| ER | 58.30 (± 0.13) | 1.28 (± 0.07) | 65.10 (± 0.32) | 1.11 (± 0.25) |
| MIR | 58.07 (± 0.30) | 1.49 (± 0.19) | 65.17 (± 0.20) | 0.86 (± 0.06) |
| DER++ | 58.30 (± 0.19) | 1.27 (± 0.51) | 65.17 (± 0.44) | 1.39 (± 0.45) |
| GMED | 58.21 (± 0.18) | 1.16 (± 0.38) | 65.24 (± 0.07) | 0.82 (± 0.17) |
| CLS-ER | 58.34 (± 0.14) | 1.22 (± 0.04) | 65.21 (± 0.08) | 1.17 (± 0.28) |
| LUMP | 58.03 (± 0.24) | 1.17 (± 0.14) | 65.38 (± 0.23) | 1.24 (± 0.23) |
| FLAVA (Ours) | 58.65 (± 0.28) | **0.64** (± 0.27) | 65.66 (± 0.16) | 0.89 (± 0.23) |
| FLAVA+ (Ours) | **59.07** (± 0.14) | 0.80 (± 0.34) | **66.05** (± 0.28) | **0.78** (± 0.27) |
| Multitask | 59.90 (± 0.35) | — | 67.81 (± 0.12) | — |

minimizing masked reconstruction loss with the locally aligned subset of audio-video patches gives an advantage in learning the joint audio-video representation. It is noteworthy that the average forgetting in the classification tasks is small when compared to the average forgetting in zero-shot retrieval tasks. In order to investigate the forgetting robustness of the audiovisual classification task, we conduct an ablation study in Appendix E. In addition, we also study the effect of rehearsal memory size, and the ability of the model to adapt to various audiovisual downstream tasks in Appendix E.

**Core components in FLAVA contribute to improving evaluation performance.** To validate the efficacy of our patch selection method, we study the effect of our core components as well as compare them with an adaptive patch selection method, MATS (Hwang et al., 2022), which aims to discard redundant patches during video representation learning. Random denotes a simple random patch selection method. We decompose FLAVA into the Localized audio-video alignment (LAVA) and Forget-robust selection (FRS). All the methods followed the default sampling ratio and were built upon DER++. As shown in Table 3, notably, LAVA and FRS outperform all baselines in VGGSound zero-shot retrieval tasks (Table 1). LAVA enhances the model's comprehension of audio-video semantics during pre-training but shows susceptibility to forgetting. In contrast, FRS demonstrates more robustness in forgetting but has a lower average retrieval score, suggesting a need for improved guidance in understanding audio-video semantics. MATS also performs competitively, highlighting the benefit of selecting informative video patches. However, it suffers from severe forgetting compared to LAVA. We posit that this can be attributed to the learning of redundant audio patches within MATS, thus shedding light on the potential causes of forgetting as discussed in §3.2. Random, on the other hand, yields results similar to DER++, suggesting no advantage in understanding multimodal semantics with random patch selection.

**FLAVA can preserve the modality gap between audio and video embeddings even after continual learning**. Recent research in multimodal learning (Liang et al., 2022) reveals that embeddings cluster by modality in representation space. Such modality-dependent clustering behavior introduces the concept of modality gap, which refers to the distance between these clusters. A larger modality gap is generally considered favorable under well-separated modality clusters since it indicates that the model can distinguish between different modalities effectively. Hence, in the context of continual audio-video representation learning, maintaining a large modality gap between the two modalities corresponding to previous tasks is desirable, as deviating from it suggests a departure from the optimal state. In order to understand the performance enhancements achieved by our approach, we

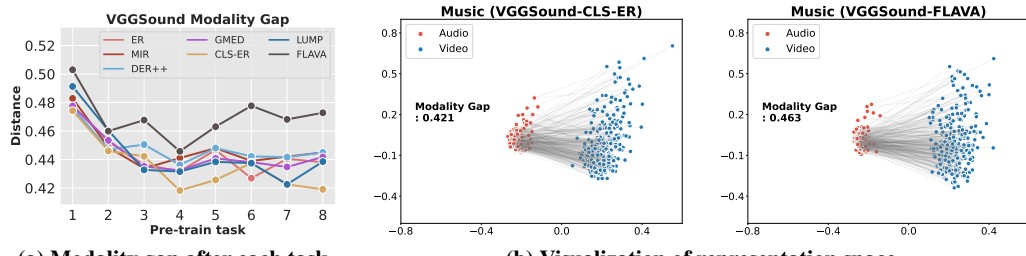

(a) Modality gap after each task         (b) Visualization of representation space

Figure 5: **Modality gap estimation.** **(a)**: Estimation of the modality gap after the completion of each task. **(b)**: Visualizations of the modality gap corresponding to the music task with the model pre-trained up to the last task in VGGSound dataset with CLS-ER method (Left) and our method (Right).

estimate the modality gap at the end of each task, utilizing evaluation data from tasks where the model was trained. The observed changes in the modality gap for the various baselines are presented in Figure 5 (a). It is notable that FLAVA consistently maintains the highest modality gap compared to other approaches, which experience substantial performance drops. This decline indicates the distinction between audio and video embeddings has deteriorated, which supports the performance improvements in our approach in Table 1 and Table 4. To delve deeper into the maintenance of the modality gap, we visualize the modality gap using embeddings extracted from the model at the completion of the last task. In Figure 5 (b), we visualize the task-specific (music, second task) modality gap and compare the modality gap between CLS-ER and our approach. It is evident that our approach enables better clustering while preserving a larger modality gap. This observation elucidates the superior forget-robustness of our approach. Appendix G provides more analysis using the modality gap including AudioSet and each component of our approach.

## 6 CONCLUSION

Audio-video multimodal learning is becoming more important due to its ability to leverage web-scale data with minimal human intervention. Nevertheless, most audio-video models still struggle to learn from real-world scenarios where the model should handle a dynamic shift of audio-visual data distribution over time, as the agent's surroundings can continuously change over time, with the past environments no longer relevant. In this paper, we investigate the critical challenges in audio-video representation learning with task-free continual learning scenarios, where the model continuously learns a course of audio-video multimodal tasks sequentially and cannot access previous tasks and task oracle both on pre-training and fine-tuning. We empirically observe that the audio-video models suffer from the issue of sparse spatiotemporal correlation and representational forgetting of audio-video relationships. To overcome these limitations, we propose a novel continual audio-video multimodal representation learning method for the first time that adaptively captures sparse audio-video attention to learn accurate audio-video relationships while mitigating forgetting from previously learned relationships without requiring task identification.

## 7 REPRODUCIBILITY STATEMENT

Our codes are based on the publicly available RepLAI (Mittal et al., 2022), TVLT (Tang et al., 2022), and CAV (Gong et al., 2023). The experimental setup and details can be found in §5 and Appendix A. We have included our codes in the supplementary material and publicly release our code.

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

# Appendix

**Organization**    The supplementary file is organized as follows: First, we explain the implementation details for our experiments in Appendix A. Then, we outline the evaluation protocol of our experiments in Appendix B. In Appendix C, we elaborate on the audio-video self-supervised objectives used for pre-training the model. Additionally, Appendix D presents a detailed account of the training procedure for the VAM module. We provide additional experimental results in Appendix E. Appendix F showcases the outcomes of our hyperparameter tuning process. Furthermore, in Appendix G, we conduct more analysis on our experimental results using the modality gap. We present PyTorch-like pseudo code for audio patch selection in Appendix H. In both Appendix I and Appendix J, we provide more examples of visualization that show challenges in audio-video lifelong pre-training. Finally, Appendix K outlines the limitations of our study.

## A    IMPLEMENTATION DETAILS

**Hyperparameter configurations.**    We referred to the original papers for initial settings of hyperparameters of continual learning methods. Based on the initial settings, we tune the hyperparameters for our audio-video continual representation learning. Searched hyperparameters are listed in Table 5. In our method, $\alpha$ denotes a multiplier for the penalty loss to minimize the distance between obtained logits from the buffer instances and their logits stored at the past timestep. We also listed our pre-training and fine-tuning hyperparameters in Table 6.

Table 5: **Continual learning method hyperparameters.**

| METHOD | VGGSound | AudioSet |
|---|---|---|
| ER | - | - |
| MIR | $C : 5$ | $C : 5$ |
| DER++ | $\alpha : 0.5$ | $\alpha : 1.0$ |
| GMED | $\alpha : 0.1\ \beta : 0.05\ \gamma : 1.0$ | $\alpha : 0.1\ \beta : 0.01\ \gamma : 1.0$ |
| CLS-ER | $\lambda : 0.1\ \alpha_S : 0.999\ \alpha_P : 0.999\ r_S : 0.6\ r_P : 0.8$ | $\lambda : 0.1\ \alpha_S : 0.999\ \alpha_P : 0.999\ r_S : 0.6\ r_P : 0.8$ |
| LUMP | $\lambda : 0.1$ | $\lambda : 0.05$ |
| FLAVA (Ours) | $\alpha : 0.5\ \beta : 0.4\ \rho_a : 0.5\ \rho_v : 0.5$ | $\alpha : 0.5\ \beta : 0.1\ \rho_a : 0.5\ \rho_v : 0.5$ |

Table 6: **Audio-Video pre-training and fine-tuning hyperparameters.**

| | Pretrain | | Finetune | | |
|---|---|---|---|---|---|
| Dataset | VGGSound | AudioSet | VGGSound | AudioSet | AVE |
| Optimizer | Adam | | AdamW | | |
| Optimizer momentum | $\beta_1, \beta_2 = 0.95, 0.999$ | | | | |
| Learning rate | 1e-4 | | 5e-4 | | 1e-3 |
| Weight decay | 5e-7 | | 5e-6 | | |
| Learning rate schedule | - | | CosineScheduler | | |
| Warmup epochs | - | | 2 | | |
| Epoch | 10 | 15 | 15 | | |
| Batch size | 48 | 36 | 48 | 36 | 12 |
| GPUs | 4 A100 or 4 V100 | | 4 Titan X Pascal | | |
| Audio Random Time Shifting | | | yes | | no |
| Audio Random Noise | | | yes | | no |
| Audio Random Time masking | no | | yes | | no |
| Audio Random Frequency masking | no | | yes | | no |
| Audio Norm Mean | -5.081 | | | | |
| Audio Norm STD | 4.485 | | | | |
| Video MultiScaleCrop | yes | | | | |
| Video Norm Mean | [0.485, 0.456, 0.406] | | | | |
| Video Norm STD | [0.229, 0.224, 0.225] | | | | |

**Baselines.**   ER (Rolnick et al., 2019) employs rehearsal memory and learns the past data in the memory during training on the current task to mitigate forgetting. All the baselines below employ the rehearsal memory to store the subset of past data. MIR (Aljundi et al., 2019a) introduces a strategy that retrieves data the model is likely to forget during the current task and trains the model with the retrieved data. To retrieve the data, it pseudo-updates the model with the data in the current step and finds the mini-batch of past data that gives the highest training loss. DER++ (Buzzega et al., 2020) matches stored logits in the rehearsal memory from past tasks with the current ones, ensuring a smoother transition and preventing abrupt changes in the logits during training. In our setting, we store both audio and video logits in the rehearsal memory and apply the method independently. GMED (Jin et al., 2021) tackles forgetting by using gradient information to update past data in the rehearsal memory. The data is updated to maximize interference of the current task to help the model retain past knowledge. Hence, it virtually updates the model with data from the current step and calculates the relative gradient by the past data to update the past data. CLS-ER (Arani et al., 2022) draws inspiration from the complementary learning system theory and maintains two models to retain short-term memories and long-term memories; one quickly adapts to new tasks and the other is slowly updated to retrain past knowledge. The slowly updated model transfers retained knowledge to the adaptable one, ensuring the retention of past information. Lastly, LUMP (Madaan et al., 2022) integrates past and current data by mixing the two data, rather than replaying the past data together with data from the current task to handle the forgetting issue. In our setting, we integrate the past and current video and audio respectively with the same ratio.

## B   CONTINUAL PRE-TRAINING EVALUATION PROTOCOL

**Audiovisual Dataset Configuration**   In this section, we specify how we design our continual pre-training experiments using two benchmark datasets: VGGSound and AudioSet. First, in the VG-GSound dataset, in order to mimic the dynamic shift of data distribution due to environmental change described in §1, we split the VGGSound dataset into eight tasks based on the category labels (Chen et al., 2020). Each task dataset consists of 6k-8k video clips from 20 different classes as in Figure 6 (a). We construct a pre-training dataset by combining the unused training dataset in VGGSound with the AudioSet-20k (Gemmeke et al., 2017), resulting in a total of 104k video clips. Before continual pre-training. all baselines and our FLAVA initialize the backbone weights using the model pre-trained on this merged dataset. We took care to exclude the unused VGGSound video samples whose class labels are present in the tasks during continual pre-training. This measure ensured that the model did not acquire any task-specific knowledge during this stage. During continual pre-training, we followed the task sequence: sports→music→vehicle→people→animals→home&nature→others part1(tools&others)→others part2(remaining others).

Similarly, we divided the AudioSet dataset into seven tasks, following class hierarchy information (Gemmeke et al., 2017). Unlike the VGGSound, each task exhibits significant variation in the number of clips, as demonstrated in Figure 6 (b). Moreover, the number of clips in each task is much larger than those from the VGGSound tasks. To ensure proper pre-training for the AudioSet experiments, we pre-trained the model with the entire VGGSound dataset to avoid any potential performance issues during the initial stages of continual pre-training. To pre-train the model with imbalanced dataset sizes for each task, we randomly shuffle the pre-train order and follow the task sequence: human→vehicle→nature→animal→others→home→music.

**Comparison between SCL setup and UCL setup**   To clarify our continual audio-video representation learning setup, we compare the conventional supervised continual learning with our setting as shown in Figure 7. The supervised continual learning (Pian et al., 2023; Mo et al., 2023) revolves around updating a model based on new tasks that introduce unseen classes or domains while ensuring the preservation of past knowledge within the model. This adaptation requires human-annotated labels, but acquiring high-quality human annotations is expensive and not scalable to large datasets.

Conversely, unsupervised continual learning (Madaan et al., 2022)(Yan et al., 2022) focuses on adapting new representations from new datasets with altered data distributions while maintaining the learned representations from previous tasks. Notably, this approach doesn't rely on human-annotated information. In our specific scenario, we align with the principles of unsupervised continual learning. Our evaluation process involves subjecting the continually pre-trained models to various downstream

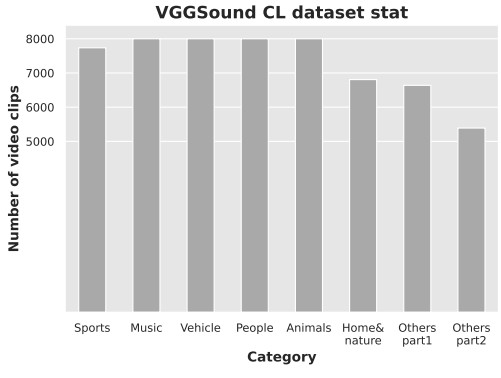

(a) VGGSound CL dataset statistics

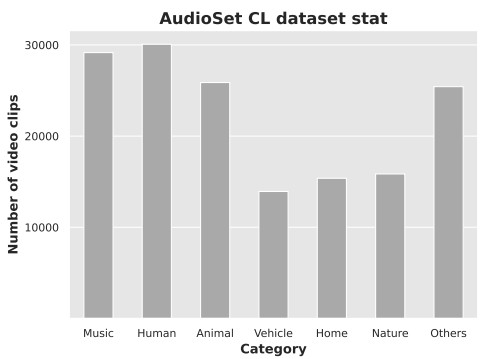

(b) AudioSet CL dataset statistics

| | VGGSound | | | | | | | | AudioSet | | | | | |
| Category | Sports | Music | Vehicle | People | Animals | Home&nature | Others part1 | Others part2 | Music | Human | Vehicle | Home | Nature | Others |
|---|---|---|---|---|---|---|---|---|---|---|---|---|---|---|
| Train | 7737 | 8000 | 8000 | 8000 | 8000 | 6807 | 6639 | 5390 | 29144 | 30073 | 25865 | 13924 | 15373 | 15848 25416 |
| Evaluate | 886 | 907 | 906 | 862 | 873 | 837 | 863 | 810 | 1188 | 1523 | 1011 | 680 | 864 | 403 1585 |
| Retrieval | 119 | 288 | 139 | 209 | 317 | 92 | 85 | 104 | 130 | 145 | 126 | 103 | 81 | 42 175 |

(c) Overall statistics

Figure 6: **Stastics of Audio-Video datasets (a)**: We split the VGGSound dataset into 8 tasks, where each subset dataset consists of similar amounts of video clips. **(b)**: We split the AudioSet into 7 tasks. Unlike the VGGSound dataset, the number of video clips varies significantly for each task. **(c)**: the number of video clips per each task.

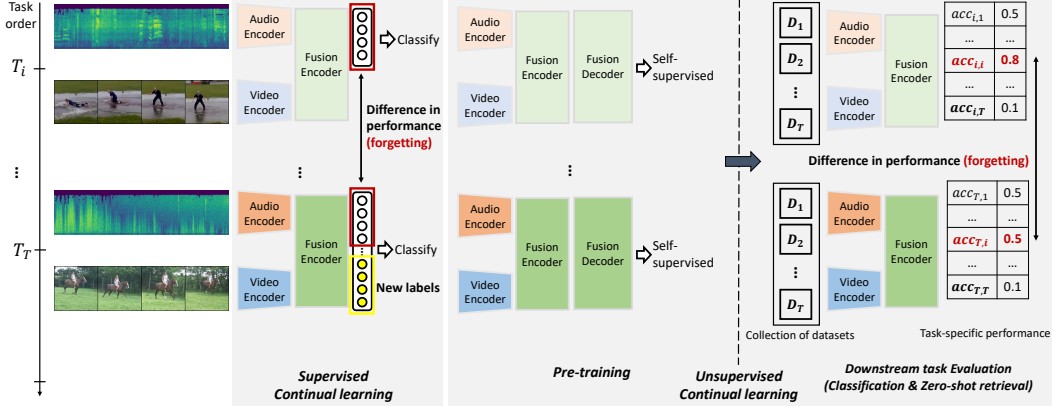

Figure 7: **Comparison with supervised continual learning (SCL) and unsupervised continual learning (UCL)**: We illustrate the difference between the SCL setup and the UCL setup that we used in our experiments.

tasks, as illustrated on the right side of Figure 7. This approach allows for a quantitative assessment of the acquired representations.

**Audiovisual downstream task configuration**    When constructing audiovisual zero-shot retrieval tasks for model performance evaluation, we refer to the CAV (Gong et al., 2023) for both the VGGSound and AudioSet experiments. We employ the zero-shot retrieval task in the CAV but exclude evaluation samples that belong to the classes that are not included in any of the tasks. In the audiovisual classification task, we follow the same training and evaluation dataset used in the pre-training. Moreover, in the fine-tuning stage, we freeze the backbone model and connect it to a randomly initialized linear classification head in order to evaluate the acquired representation.

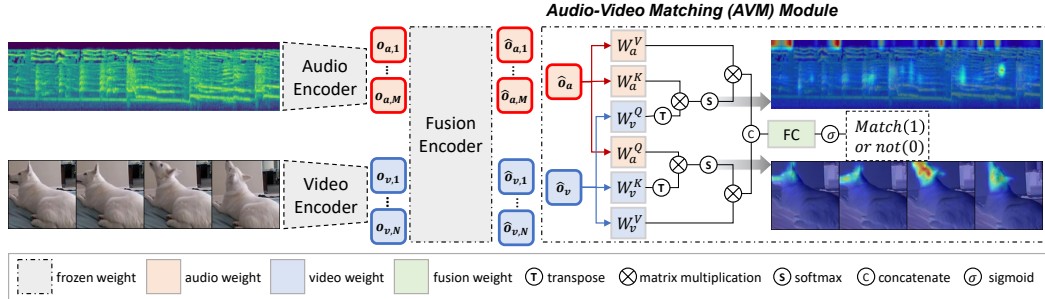

Figure 8: **Overview of AVM module**: The AVM (Audio-Visual Matching) module is self-supervised with the audio-video matching objective. It classifies if the given audio-video pair is positive(audio and video are from the same video) or negative(audio and video are from different videos).

## C  AUDIO-VIDEO SELF-SUPERVISED OBJECTIVES

The backbone Transformer consists of an audio encoder ($E_a(\cdot)$), a video encoder ($E_v(\cdot)$), a multi-modal fusion encoder ($E_f(\cdot)$), and a decoder ($D(\cdot)$). Given audio-video data ($\boldsymbol{x}_a, \boldsymbol{x}_v$) and corresponding audio-video tokens ($\boldsymbol{a}, \boldsymbol{v}$), we pre-train the model by minimizing the mask reconstruction loss $\ell^r$:

$$\tilde{\boldsymbol{a}}, \, \tilde{\boldsymbol{v}} = E_f\left(E_a\left(\boldsymbol{m}_a \otimes \boldsymbol{a}\right), \, E_v\left(\boldsymbol{m}_v \otimes \boldsymbol{v}\right)\right),$$

$$\ell^r = \ell^r_a + \ell^r_v = \frac{1}{B}\sum_{i=1}^{B}\left[\frac{\left(D\left(\tilde{\boldsymbol{a}}_i\right) - \boldsymbol{m}_{a,i} \otimes \boldsymbol{x}_{a,i}\right)^2}{|\boldsymbol{m}_{a,i}|} + \frac{\left(D\left(\tilde{\boldsymbol{v}}_i\right) - \boldsymbol{m}_{v,i} \otimes \boldsymbol{x}_{v,i}\right)^2}{|\boldsymbol{m}_{v,i}|}\right]. \tag{8}$$

where $\otimes$ denotes vector-matrix multiplication while preserving the input's dimensionality. Random audio $\boldsymbol{m}_a$ and video mask $\boldsymbol{m}_v$ are drawn by a binary distribution. In this paper, we set a probability of 0.8 for masking, consistent with Huang et al. (2022a). Using the unmasked patches, we aim to learn the model to reconstruct the masked audio and video patches.

In addition, we also minimize masked contrastive loss to learn the semantic relationship between audio and video representation pairs by pulling those that share the same semantics while pushing the others. Following by Gong et al. (2023), we pass the masked input tokens to audio and video encoders, and subsequently map obtained features (i.e., outputs) to the fusion encoder with modality-specific layer normalization for the masked contrastive learning:

$$\boldsymbol{c}_a = \texttt{MeanPool}\left(E_f\left(E_a\left(\boldsymbol{m}_a \otimes \boldsymbol{a}\right), LN_a\right)\right), \quad \boldsymbol{c}_v = \texttt{MeanPool}\left(E_f\left(E_v\left(\boldsymbol{m}_v \otimes \boldsymbol{v}\right), LN_v\right)\right),$$

$$\ell^c = -\frac{1}{B}\sum_{i=1}^{B}\left[\texttt{log}\left(\frac{\exp(\boldsymbol{c}_{a,i}^{\top}\boldsymbol{c}_{v,i}/\tau)}{\sum_{j=1}^{B}\exp(\boldsymbol{c}_{a,i}^{\top}\boldsymbol{c}_{v,j}/\tau)}\right) + \texttt{log}\left(\frac{\exp(\boldsymbol{c}_{v,i}^{\top}\boldsymbol{c}_{a,i}/\tau)}{\sum_{j=1}^{B}\exp(\boldsymbol{c}_{v,i}^{\top}\boldsymbol{c}_{a,j}/\tau)}\right)\right] \tag{9}$$

, where $\tau$ is temperature hyperparameter, and $LN_a$ and $LN_v$ indicate modality-specific layer normalization for audio and video each.

## D  TRAINING OF AUDIO-VIDEO MATCHING MODULE

**AVM training procedure.**  In the following section, we describe the training process of the AVM module, as illustrated in Figure 8. Given audio-video token pairs ($\boldsymbol{a}, \boldsymbol{v}$) with the batch size of $B$, we propagate token inputs to the frozen encoder for each modality and obtain audio-video representation pairs ($\boldsymbol{o}_a, \boldsymbol{o}_v$). In order to update the module to capture the multimodal correlation between audio and its video pair, we randomly split them into positive and negative pairs, where we construct negative pairs by randomly shuffling the audio tokens to pair with unmatched video tokens. Next, we project the obtained positive and negative pairs into fusion space ($\tilde{\boldsymbol{o}}_a, \tilde{\boldsymbol{o}}_v$) through the fusion encoder. Subsequently, the input pairs are fed into the AVM module. They are projected to keys, queries, and values for the cross-attention operation, by passing through trainable projection layers.

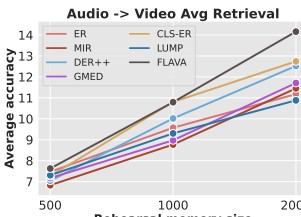 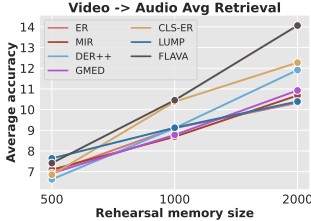 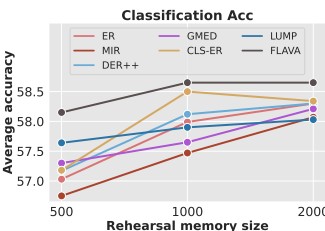

Figure 9: **Downstream performance on various rehearsal memory sizes.** We evaluated downstream task performances on the pre-trained models with various rehearsal memory sizes on the VGGSound dataset.

The above process can be summarized as follows:

$$
\begin{aligned}
\boldsymbol{o}_a, \boldsymbol{o}_v &= E_a(\boldsymbol{a}), E_v(\boldsymbol{v}), \quad \tilde{\boldsymbol{o}}_a, \tilde{\boldsymbol{o}}_v = E_f(\boldsymbol{o}_a, \boldsymbol{o}_v) \\
\boldsymbol{q}_a = \tilde{\boldsymbol{o}}_a \mathcal{W}_a^Q, \ \boldsymbol{k}_a = \tilde{\boldsymbol{o}}_a \mathcal{W}_a^K, \ \boldsymbol{v}_a = \tilde{\boldsymbol{o}}_a \mathcal{W}_a^V, \quad & \boldsymbol{q}_v = \tilde{\boldsymbol{o}}_v \mathcal{W}_v^Q, \ \boldsymbol{k}_v = \tilde{\boldsymbol{o}}_v \mathcal{W}_v^K, \ \boldsymbol{v}_v = \tilde{\boldsymbol{o}}_v \mathcal{W}_v^K, \\
\boldsymbol{A}_a &= \texttt{Softmax}\left(\mu_{\boldsymbol{\theta}}(\boldsymbol{q}_v, \boldsymbol{k}_a, \beta{=}1)\right), \quad \boldsymbol{A}_v = \texttt{Softmax}\left(\mu_{\boldsymbol{\theta}}(\boldsymbol{q}_a, \boldsymbol{k}_v, \beta{=}1)\right), \\
\boldsymbol{V}_a &= \boldsymbol{A}_a \boldsymbol{v}_a, \quad \boldsymbol{V}_v = \boldsymbol{A}_v \boldsymbol{v}_v
\end{aligned}
\tag{10}
$$

where the projections $\mathcal{W}_a^Q, \mathcal{W}_a^K, \mathcal{W}_a^V, \mathcal{W}_v^Q, \mathcal{W}_v^K, \mathcal{W}_v^V \in \mathbb{R}^{D \times H \times d}$ are trainable parameter matrices; $D = H * d$. $\boldsymbol{V}_a \in \mathbb{R}^{B \times H \times N \times D}$, $\boldsymbol{V}_v \in \mathbb{R}^{B \times H \times M \times D}$ are values highlighted by the cross-attention maps.

Next, we average the values head-wise and token-wise, and concatenate the resulting two values into $\boldsymbol{va} \in \mathbb{R}^{B \times 2D}$ in order to merge the multimodal information. Then it is passed to fully connected (FC) layers, which serve as the classification head. These FC layers take $\boldsymbol{va}$ as input, generating a vector $\hat{\boldsymbol{y}} \in \mathbb{R}^B$ that predicts whether each input pair corresponds to a negative of positive pair. For training the AVM module, we employ the binary cross-entropy loss to classify audio-video pairs, i.e.,

$$
\begin{aligned}
\widehat{\boldsymbol{V}}_{av} &= \texttt{Concat}\left(\texttt{MeanPool}(\boldsymbol{V}_a), \texttt{MeanPool}(\boldsymbol{V}_v)\right), \\
\widehat{\boldsymbol{y}} &= \texttt{Sigmoid}\left(\texttt{FC}(\widehat{\boldsymbol{V}}_{av})\right), \ \mathcal{L}^{avm} = -\boldsymbol{y}\left(\log(\widehat{\boldsymbol{y}})\right),
\end{aligned}
\tag{11}
$$

Here, $\boldsymbol{y} = \{0,1\}^B$ represents ground truth labels, with $\boldsymbol{y}_i$ taking the value 0 when the $ith$ input audio-video pair is a negative pair and 1 otherwise. We pre-train the AVM module along with the backbone model. During the weight update process in the AVM module, the gradient computed from the audio-video matching objective does not propagate through the backbone encoder. This design choice ensures exploiting the AVM at a low cost. Moreover, the AVM only increases 3.18% of the total backbone model size (707.8 MB), which is efficient compared to methods like CLS-ER which require two backbones during training.

# E  ADDITIONAL EXPERIMENTAL RESULTS

**Effect of rehearsal memory size**  We explore the impact of rehearsal memory size on the downstream task performance and report the results in Figure 9. We find that our method mostly surpasses baselines in various memory sizes. However, as the memory size decreases, the performance gap in retrieval tasks also narrows. This shows the necessity of diverse audio-video samples from past time steps in order to preserve audio-video correlation, underscoring the susceptibility of exemplar-based methods in the context of continual learning. In the classification task, with a small memory size (500), baselines except for LUMP show similar average accuracy to that of Finetune, which is 57.01. However, LUMP and our approach exhibit low sensitivity to variations in memory size. This could be ascribed to the inherent data augmentation in both methods; LUMP employs a Mixup operation with the current and past data, while our approach samples a subset of patches from the past data.

**Audio patch selection strategy.**  When executing the selection of audio patches guided by the audio importance score $\boldsymbol{I}_a$, our approach involves selecting patches in time-wise segments, following the procedure detailed in Algorithm 2. As spectrogram patches exhibit local correlation driven by their temporal continuity (Huang et al., 2022b), the strategy for audio patch selection becomes pivotal in maintaining these intrinsic properties. The challenge lies in striking a balance between retaining time continuity and eliminating redundant information within the spectrogram.

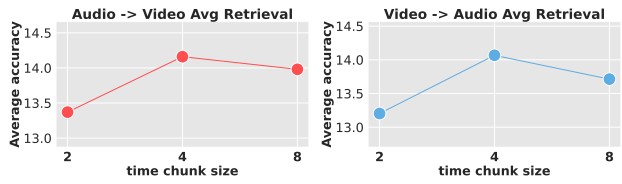
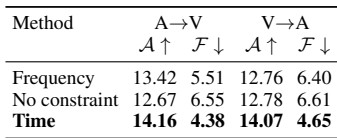

| Method | A→V | | V→A | |
| --- | --- | --- | --- | --- |
| | $\mathcal{A}\uparrow$ | $\mathcal{F}\downarrow$ | $\mathcal{A}\uparrow$ | $\mathcal{F}\downarrow$ |
| Frequency | 13.42 | 5.51 | 12.76 | 6.40 |
| No constraint | 12.67 | 6.55 | 12.78 | 6.61 |
| **Time** | **14.16** | **4.38** | **14.07** | **4.65** |

(a) Time chunk sizes    (b) Audio selection methods

Figure 10: **Variation of audio patch selection.** (a): Average retrieval task performance on various time chunk sizes. (b): Average retrieval task performance on various audio selection methods.

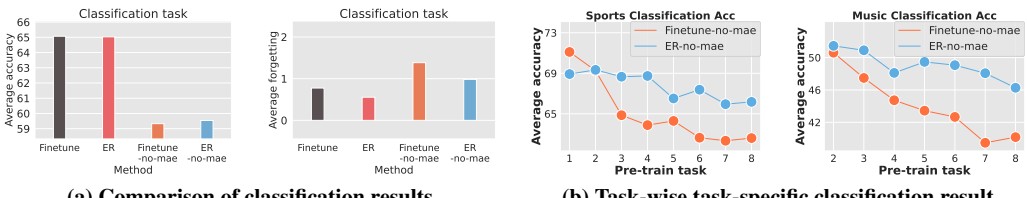

(a) Comparison of classification results    (b) Task-wise task-specific classification result

Figure 11: **Audiovisual classification results without MAE loss term.** (a): We compare the audiovisual classification results from the models with the MAE loss term (Finetune, ER) and the models without it (Finetune-no-mae, ER-no-mae). (b): We visualize changes in the task-specific average classification accuracy of the sequentially pre-trained model without the MAE loss term.

In pursuit of this balance, we conduct various experiments on the audio patch selection approach. The width of the time chunk assumes significance; a chunk that is too narrow could disrupt time continuity, while one that is excessively wide might not concisely capture core information. To validate our approach and assess the efficacy of time-wise chunk selection, we conduct two distinct sets of experiments.

The first experiment involves evaluating the model's performance across varying time chunk widths. A noteworthy observation from Figure 10 (a): adopting a size of 2 results in a noticeable performance decline. This potentially signifies the criticality of upholding the local correlation inherent in audio patches. Moving on to the second experiment, we explore various selection methods, inspired by the spectrogram masking techniques detailed in (Huang et al., 2022b). We test two variants of audio patch selection: Frequency indicates an approach of choosing audio patches frequency-wise, while No-constraint indicates selecting audio patches without any constraints; applying the same patch selection procedure as in the video patch selection. As shown in Figure 10 (b), time-wise selection exhibits superior performance compared to alternative audio selection methodologies, meaning that preserving audio information in time-chunk minimizes loss of audio properties.

**Pre-training without MAE objective.** We conduct a comprehensive analysis of our classification results as presented in Table 4, alongside the results reported in Cossu et al. (2022). These results demonstrate the robustness of the MAE objective against catastrophic forgetting in classification tasks. To further investigate the findings, we conduct an experiment where we omit the MAE objective from our pre-training objectives. For this experiment, we follow the same lifelong pre-training procedure using the VGGSound dataset but exclusively pre-train the model by minimizing the masked contrastive loss. Subsequently, we fine-tune the model for the classification task using the same procedure as our main experiments.

In Figure 11 (a), a substantial difference exists between models pre-trained without the MAE loss (Finetune-no-mae, ER-no-mae) and those pre-trained with the MAE loss (Finetune, ER). The former degrades an average accuracy with higher forgetting than the latter. Furthermore, in Figure 11 (b), the task-specific classification accuracy of the former shows a comparably noticeable decline as the model is sequentially pre-trained on the VGGSound tasks. These findings imply that the representations learned from the MAE loss are not confined to specific tasks, but rather possess higher transferability in comparison to those obtained through the masked contrastive loss.

Table 7: Results of audiovisual zero-shot retrieval task on shuffled VGGSound and AudioSet. The best and the second best results are highlighted in **bold** and underline, respectively.

| | Method | VGGSound | | | | | | | | AudioSet | | | | | | | |
| | | R@1 | | R@5 | | R@10 | | Avg | | R@1 | | R@5 | | R@10 | | Avg | |
| | | $\mathcal{A}\uparrow$ | $\mathcal{F}\downarrow$ | $\mathcal{A}\uparrow$ | $\mathcal{F}\downarrow$ | $\mathcal{A}\uparrow$ | $\mathcal{F}\downarrow$ | $\mathcal{A}\uparrow$ | $\mathcal{F}\downarrow$ | $\mathcal{A}\uparrow$ | $\mathcal{F}\downarrow$ | $\mathcal{A}\uparrow$ | $\mathcal{F}\downarrow$ | $\mathcal{A}\uparrow$ | $\mathcal{F}\downarrow$ | $\mathcal{A}\uparrow$ | $\mathcal{F}\downarrow$ |
|---|---|---|---|---|---|---|---|---|---|---|---|---|---|---|---|---|---|
| Audio-to-Video | Finetune | 0.80 | 4.15 | 2.96 | 12.23 | 5.05 | 16.91 | 2.94 | 11.10 | 1.50 | 4.72 | 5.49 | 10.41 | 9.80 | 11.91 | 5.60 | 9.01 |
| | ER | 3.89 | 3.06 | 12.10 | 6.55 | 18.30 | 7.74 | 11.43 | 5.78 | 4.52 | 3.16 | 12.72 | 6.93 | 18.83 | 8.00 | 12.02 | 6.03 |
| | MIR | 4.02 | 2.97 | 12.54 | 6.16 | 17.99 | 8.09 | 11.52 | 5.74 | 4.69 | 2.95 | 13.22 | 6.50 | 18.98 | 8.81 | 12.30 | 6.09 |
| | DER++ | 4.23 | 3.35 | 12.92 | 7.31 | 18.62 | 9.45 | 11.92 | 6.70 | 4.32 | 4.27 | 12.29 | 8.46 | 18.74 | 10.18 | 11.78 | 7.64 |
| | GMED | 3.90 | 2.94 | 11.51 | 7.41 | 17.65 | 8.87 | 11.02 | 6.41 | 4.70 | 2.48 | 12.56 | 4.55 | 18.62 | 5.05 | 11.96 | 4.03 |
| | CLS-ER | 3.94 | 3.35 | 12.96 | 7.19 | 18.09 | 10.66 | 11.66 | 7.07 | 5.16 | 2.97 | 14.33 | 6.88 | 20.24 | 8.74 | 13.24 | 6.20 |
| | LUMP | 4.06 | 2.18 | 13.21 | 4.66 | 19.34 | 5.58 | 12.20 | 4.14 | 4.45 | 3.40 | 13.05 | 6.25 | 19.45 | 7.28 | 12.32 | 5.64 |
| | FLAVA (Ours) | 4.72 | 2.89 | 14.17 | 5.74 | 19.94 | 5.74 | 12.94 | 4.79 | 4.97 | 3.47 | 13.91 | 5.59 | 20.30 | 6.70 | 13.06 | 5.25 |
| | FLAVA+ (Ours) | 4.90 | 3.19 | 16.42 | 4.72 | 23.49 | 5.89 | 14.94 | 4.60 | 5.77 | 3.90 | 17.51 | 4.49 | 23.72 | 7.07 | 15.67 | 5.15 |
| | Multitask | 6.45 | – | 20.19 | – | 29.01 | – | 18.55 | – | 8.28 | – | 24.14 | – | 33.74 | – | 22.05 | – |
| Video-to-Audio | Finetune | 0.78 | 3.77 | 3.00 | 11.68 | 5.21 | 15.86 | 3.00 | 10.44 | 1.42 | 5.11 | 6.54 | 10.30 | 10.43 | 13.48 | 6.13 | 9.63 |
| | ER | 3.57 | 2.76 | 11.66 | 7.67 | 16.75 | 10.76 | 10.66 | 7.06 | 4.01 | 4.31 | 12.47 | 7.27 | 19.32 | 9.26 | 11.93 | 6.95 |
| | MIR | 3.35 | 3.15 | 11.37 | 7.74 | 16.62 | 10.11 | 10.45 | 7.00 | 4.25 | 3.43 | 12.92 | 6.93 | 19.43 | 9.78 | 12.20 | 6.71 |
| | DER++ | 4.08 | 3.10 | 12.78 | 9.02 | 18.77 | 11.30 | 11.88 | 7.81 | 4.31 | 4.35 | 12.60 | 9.59 | 18.93 | 12.27 | 11.95 | 8.74 |
| | GMED | 3.42 | 3.80 | 11.45 | 7.76 | 17.06 | 9.94 | 10.64 | 7.17 | 4.20 | 1.87 | 12.97 | 6.04 | 19.98 | 8.11 | 12.38 | 5.34 |
| | CLS-ER | 3.49 | 3.85 | 12.28 | 8.05 | 17.75 | 11.31 | 11.17 | 7.74 | 4.85 | 5.48 | 13.37 | 9.17 | 19.69 | 11.36 | 12.64 | 8.67 |
| | LUMP | 3.98 | 1.67 | 12.44 | 5.17 | 18.11 | 7.27 | 11.51 | 4.70 | 4.23 | 4.06 | 13.53 | 6.09 | 19.27 | 9.53 | 12.34 | 6.56 |
| | FLAVA (Ours) | 4.18 | 2.54 | 13.81 | 6.56 | 19.90 | 8.88 | 12.63 | 5.99 | 4.86 | 2.92 | 14.20 | 6.41 | 20.00 | 9.82 | 13.02 | 6.38 |
| | FLAVA+ (Ours) | 5.28 | 1.81 | 15.35 | 6.33 | 21.97 | 8.01 | 14.20 | 5.38 | 5.57 | 3.80 | 16.67 | 6.96 | 23.91 | 9.28 | 15.38 | 6.68 |
| | Multitask | 6.85 | – | 21.93 | – | 30.63 | – | 19.80 | – | 8.05 | – | 25.81 | – | 35.60 | – | 23.15 | – |

**Shuffle task orders.** In addition to the main experiment results presented in Table 1, we conduct supplementary investigations with the intention of enhancing the reliability of our findings. Specifically, we carry out experiments on shuffled task sequences. For the VGGSound experiment, we randomize the original pre-train task sequence, leading to modified order: music→others part1→home&nature→sports→others part2→vehicle→animals→people. Likewise, in the case of the AudioSet experiment, we apply a similar task sequence shuffling, resulting in the following order: nature→human→home→vehicle→music→animal→others. Note that the VGGSound experiment is conducted on 36 batch size, unlike the main VGGSound experiment which is conducted on 48 batch size. We present the corresponding audiovisual zero-shot retrieval task results in Table 7. Our method shows competitive or better performance compared to other baselines, which coincides with the results in Table 1. This indicates that our method is robust under varying conditions, thereby enhancing the credibility of our analysis.

**Audiovisual Event Localization.** We conduct an audiovisual event localization (AVE) task to showcase the effectiveness of our method in precisely aligning audio and video streams. Following the experimental setup outlined in Lin et al. (2023), we utilize the AVE dataset (Tian et al., 2018) for the experiment. To assess whether continually pre-trained models can adapt to the downstream task involving the unseen dataset, we use model checkpoints obtained from continually pre-training until the final task (others part2) within the VGGSound experiment. The training process adheres to the hyperparameters described in Table 6, wherein the backbone model remains frozen while training the linear classifier. We present the summarized result in Table 8. This result demonstrates that our method surpasses other baseline methods. This underscores the strength of our method in adapting the downstream task that necessitates a sophisticated grasp of audio-video alignment at a high level.

Table 8: AVE result. The best result is highlighted in **bold**.

| | Method | Acc |
|---|---|---|
| AVE | Finetune | 52.56 |
| | ER | 54.98 |
| | DER++ | 55.81 |
| | GMED | 55.98 |
| | LUMP | 55.06 |
| | FLAVA (Ours) | **56.68** |
| | FLAVA+ (Ours) | **56.68** |
| | Multitask | 57.73 |

**Sound Source Localization.** To evaluate the model's ability to detect sound sources within visual scenes, we perform a sound source localization task with the AVE (Tian et al., 2018) dataset. Specifically, we use checkpoints of the model continually pre-trained up to the final task (others part2) within the VGGSound experiment. We follow the same attention map visualization approach in (Gong et al., 2023), which uses a cosine similarity matrix. Given that the AVE dataset represents unseen data for the pre-trained models, it allows us to gauge the extent to which the model has acquired general knowledge of audio-video correlations. The visualization of our findings is presented in Figure 12.

It is notable that all methods fail to accurately pinpoint the exact location of the sound sources. This limitation primarily stems from the inherent limitations of the backbone model. In Gong et al. (2023) Appendix I, the CAV model fails in the sound source localization task. Hence the backbone model has restricted potential to extend into audiovisual downstream tasks such as audiovisual parsing (Tian et al., 2020) and audiovisual segmentation (Zhou et al., 2022). Nevertheless, a compelling discovery emerges from our analysis in Figure 12-(e): the AVM module, continually pre-trained alongside the backbone, markedly surpasses the backbone model's capacity in identifying potential sound sources. It is noteworthy that even though the last pre-training task (others part2) lacks semantic relevance to the visual semantics portrayed in Figure 12, the AVM module demonstrates effectiveness in detecting sound sources in the visual scenes. This discovery suggests that the AVM module not only aids in capturing informative and forget-robust patches but also opens up new possibilities for adapting the pre-trained models to diverse audiovisual downstream tasks.

## F  HYPERPARAMTER TUNING RESULTS

**Patch sampling ratio.**    Central to our approach is the identification of tokens that exhibit a high localized alignment with their corresponding modality pairs while being robust to catastrophic forgetting of learned representation, enabling the retention of meaningful information. Achieving the right balance in the sampling ratio is critical: an excessively low sampling ratio hinders the model from accessing essential data, while an overly high ratio hampers the model's ability to disregard redundant or forget-inducing information.

For the audio sampling ratio, we systematically assess three options —37.5%, 50%, and 62.5%— while maintaining the video sampling ratio $\rho_v$ at 50%. Table 9 shows that sampling 50% of audio patches ensures high performance compared to the other sampling ratios. It is noteworthy that the other sampling ratios still yield competitive performance compared to the baselines. As we transition to optimizing the sampling ratio for video patches, we conduct experiments using three sampling ratios -37.5%, 50%, and 62.5%- alongside the audio sampling ratio $\rho_a$ at 50%. As demonstrated in Table 9, employing a 50% video sampling ratio ensures high performance.

Table 9: Retrieval result by sampling ratios.

| Ratio(%) | A→V | | V→A | |
|---|---|---|---|---|
| | $\mathcal{A}\uparrow$ | $\mathcal{F}\downarrow$ | $\mathcal{A}\uparrow$ | $\mathcal{F}\downarrow$ |
| $\rho_a$   37.5 | 13.76 | 4.77 | 13.52 | 5.53 |
| **50** | **14.16** | **4.38** | **14.07** | **4.65** |
| 62.5 | 13.77 | 5.04 | 13.46 | 5.06 |
| $\rho_v$   37.5 | 13.35 | 5.57 | 13.39 | 5.93 |
| **50** | **14.16** | **4.38** | **14.07** | **4.65** |
| 62.5 | 13.82 | 4.50 | 13.53 | 5.27 |

**Inference temperature in AVM module.**    In our approach, we actively harness cross-attention maps from the AVM module computed in Equation 2. During inference, we set the temperature hyperparameter $\beta$ to 0.4 for the VGGSound experiments. To examine the significance of $\beta$, we explore a range of the hyperparameter values, specifically 0.1, 0.4, and 0.5. The results, as summarized in Table 10, indicate that the optimal temperature values typically reside within the range of approximately 0.1 to 0.4. This suggests the need for

Table 10: Retrieval result by temperature values.

| $\beta$ | A→V | | V→A | |
|---|---|---|---|---|
| | $\mathcal{A}\uparrow$ | $\mathcal{F}\downarrow$ | $\mathcal{A}\uparrow$ | $\mathcal{F}\downarrow$ |
| 0.1 | 13.91 | 5.42 | **14.23** | 4.97 |
| **0.4** | **14.16** | **4.38** | 14.07 | **4.65** |
| 0.5 | 13.37 | 5.27 | 13.50 | 5.84 |

heightened emphasis on discriminative audio and video patches in order that those patches are more frequently selected in our selection framework in Equation 6 and in Algorithm 2.

## G  ADDITIONAL ANALYSIS OF MODALITY GAP

In the main paper, we examine the performance improvements of our approach in the context of audio-video continual pre-training with respect to the modality gap. In this section, we conduct a more detailed analysis; covering differences in the modality gap (Figure 13 (a)), exploring the modality gap within the AudioSet dataset (Figure 13 (b)), and providing additional visualizations of the modality gap to support the effectiveness of our approach (Figure 13 (c)).

In Figure 13 (a), our approach stands out with the smallest average modality gap difference. However, our approach does not exhibit high resistance to modality gap fluctuations within the AudioSet experiment. An interesting observation emerges when comparing the average modality gap difference with the average forgetting in Table 1; a smaller average modality gap difference seems to correspond to lower average forgetting in the zero-shot retrieval tasks. This aligns with the relatively high average

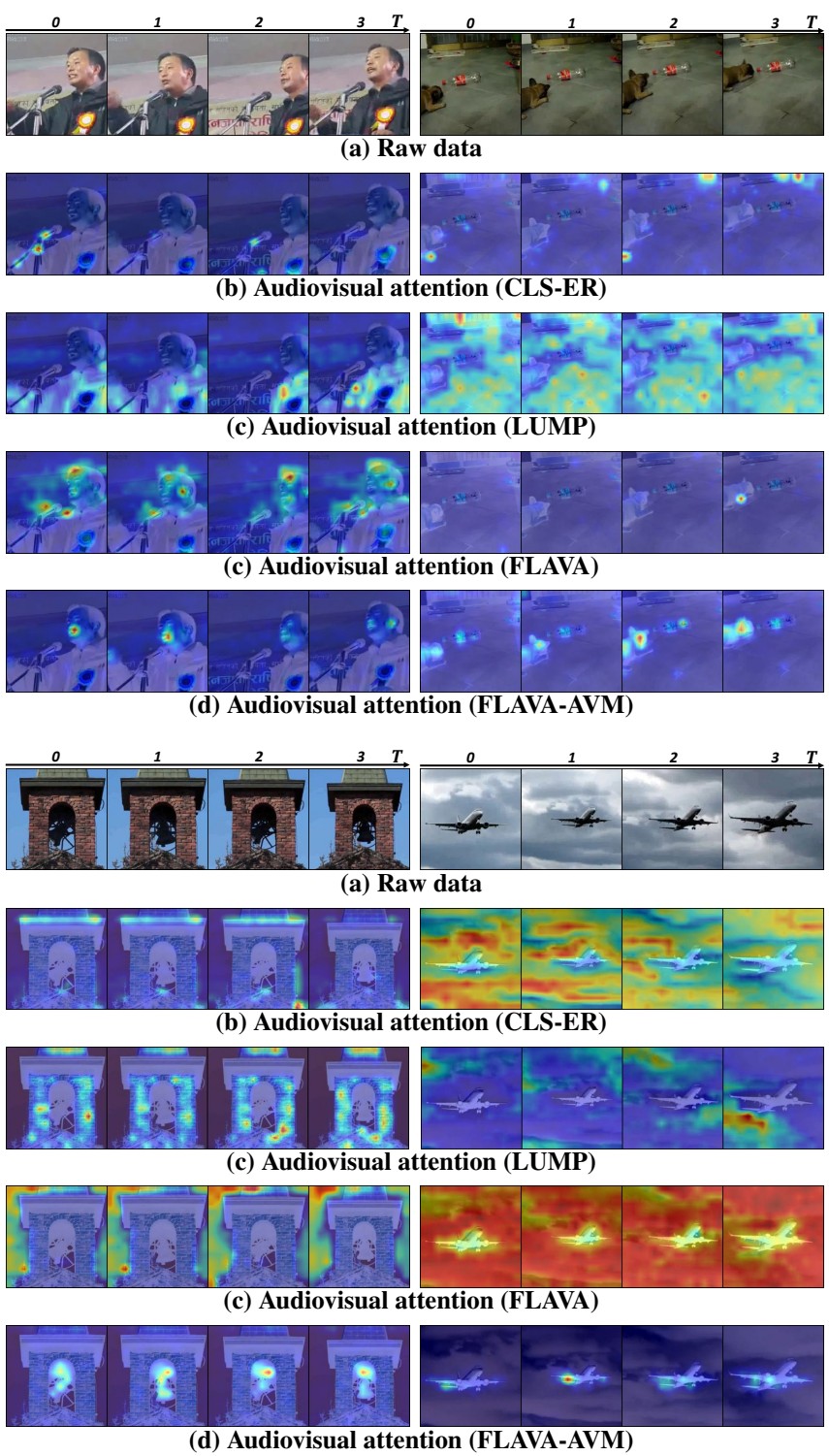

Figure 12: **Sound source localization** (a) Examples of raw video frames. (b) (d) We visualize cross-attention maps using cosine similarity between each video patch and averaged audio embedding. (e) We use the AVM module continually pre-trained with our backbone to visualize cross-attention maps. The AVM module is much more effective in capturing potential sound sources compared to the ability of the backbone to capture the sources.

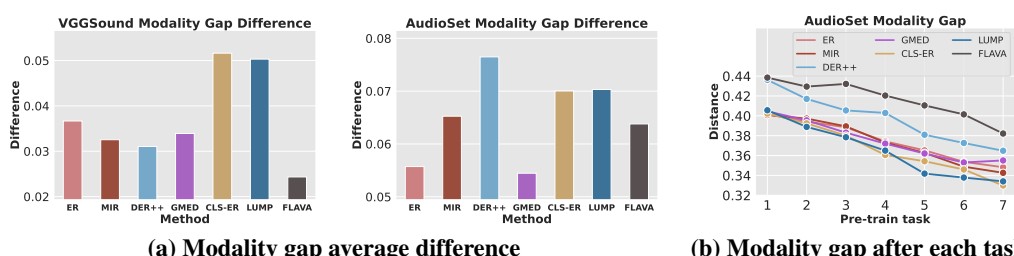

**(a) Modality gap average difference**

**(b) Modality gap after each task**

Figure 13: **Modality gap estimation. (a)**: Average modality gap difference between the modality gap estimated at the completion of the last task and the modality gap estimated at the completion of each task. **(b)**: Estimation of modality gap after the completion of each task (AudioSet).

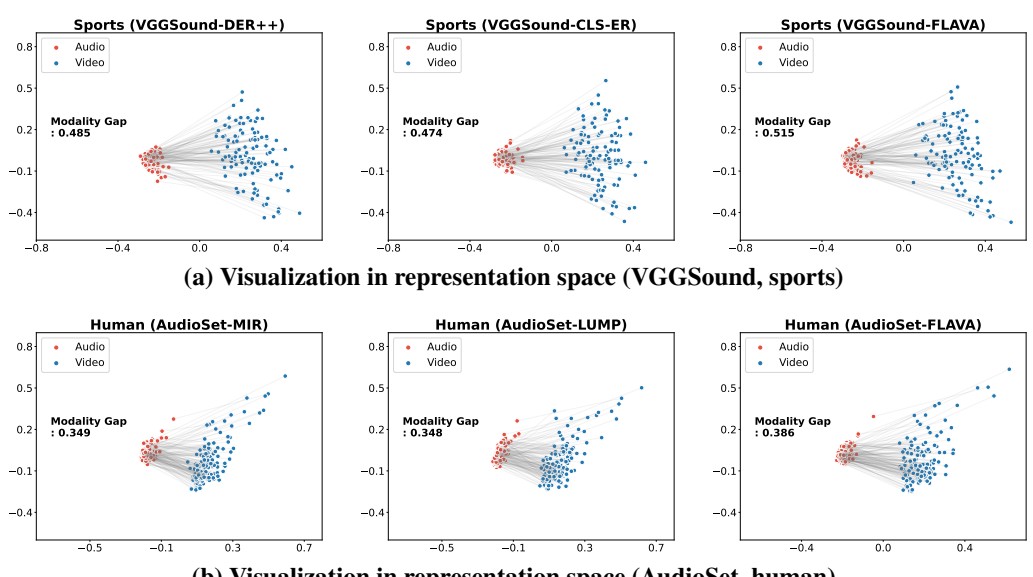

**(a) Visualization in representation space (VGGSound, sports)**

**(b) Visualization in representation space (AudioSet, human)**

Figure 14: **Modality gap visualization. (a)**: Visualizations of the modality gap corresponding to the sports task with the model pre-trained up to the last task in the VGGSound experiment. **(b)**: Visualization of the modality gap corresponding to the human task with the model pre-trained up to the last task in the AudioSet experiment.

forgetting of our approach in the AudioSet experiment, suggesting that the modality gap difference holds potential as a metric for assessing the extent of forgetting in audio-video correlation. Meanwhile, our approach consistently maintains the highest modality gap in all pre-train tasks (Figure 13 (b)), which explains the high average accuracy of our approach in the AudioSet retrieval tasks.

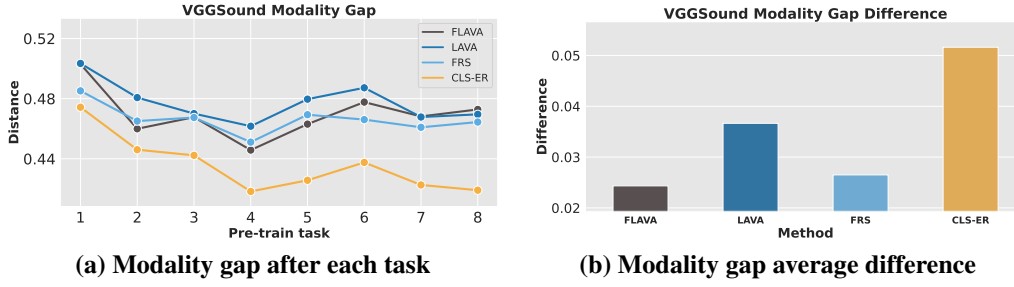

**(a) Modality gap after each task**

**(b) Modality gap average difference**

Figure 15: **Modality gap estimation for each component of our proposed method. (a)**: Estimation of modality gap after completing each task. **(b)**: Average difference in modality gap between the completion of the last task and the completion of each individual task.

We take our analysis a step further by visually representing the modality gap. In Figure 14 (a), we visualize the evaluation of audio-video data pairs from the sports task in the VGGSound experiments. Similarly, in Figure 14 (b), we visualize data from the human task in the AudioSet experiments. In both visualizations, we use the models that completed the continual pre-training phase. Remarkably, our approach consistently yields a larger gap in both cases. This suggests that the modality gap established from the initial task (sports, human) is effectively maintained, enabling the models to distinguish between different modalities, ultimately leading to enhanced performance.

We estimate the modality gap of two key components within our proposed method: LAVA (Localized audio video alignment §4.1) and FRS (Forget robust selection §4.2). The LAVA consistently exhibits the highest modality gap across the tasks, as depicted in Figure 15-(a). This underscores the effectiveness of the proposed method in §4.1 in identifying patches that demonstrate high localized alignment with their modality pairs. Consequently, the LAVA achieves better audio and video clustering within the multi-modal representation space, resulting in enhanced average accuracy in Table 3. This observation strongly supports our claim that the method outlined in §4.1 adeptly selects informative multi-modal patches from raw data.

The FRS illustrates a relatively minor modality gap difference, as indicated in Figure 15-(b). During the continual pre-training, the modality gap between the audio and video exhibits robustness to the effect of changing distribution. Hence, the model maintains learned audio-video alignment. This explains the small average forgetting exhibited by the FRS in Table 3. It affirms our claim that the method introduced in §4.2 proficiently selects forget-robust patches.

## H    AUDIO PATCH SELECTION PSEUDO CODE

**Algorithm 2** Audio time chunk selection in a PyTorch-like Style.

```
# I_a:     audio patch importance score matrix
# P_a:     audio pruning probability matrix
# L_c:     audio time chunk size
# kappa_a:  target length of audio selection
# num_time:  the number of tokens in time dimension
# num_freq:  the number of tokens in frequency dimension
def audio_time_chunk_selection(I_a,P_a):
    F_a=bernoulli(P_a)
    F_a=F_a.reshape(num_time, num_freq)
    F_a_t=F_a.sum(dim=1) # # of pruned patches
    I_a_t=I_a.reshape(num_time, num_freq)
    I_a_t=I_a_time.sum(dim=1) # Time-wise importance
    I_a_c=avg_pool(I_a_t, kernel_size=L_c) # Chunk-wise importance
    num_chunk=len(I_a_c)
    t_select=multinomial(I_a_c, num_samples=num_chunk)
    num_tokens=0
    for j in range(num_chunk):
        t=t_select[j]
        num_prune=F_a_t[t*L_c:(t+1)*L_c].sum() # # of pruned patches
        num_tokens+=(num_time*num_freq - num_prune) # Count # of patches
        if num_tokens > kappa_a:
            F_last=F_a[t*L_c:(t+1)*L_c].view(-1)
            F_last_accum=cumsum(flip(~F_last))
            prune_tail_idx= F_last_accum == num_tokens-kappa_a
            F_last[-(prune_tail_idx+1):]=True # Prune tail of last chunk
        F_a[t*L_c:(t+1)*L_c]=F_last.reshape(num_time,num_freq)
        for k in range(j+1, num_chunk):
            t_prune=t_select[k]
            F_a[t_prune*L_c:(t_prune+1)*L_c]=True
        break
    F_a=F_a.view(-1).float()
    S_tilde_a=argsort(F_a) # Forget-robust audio sorted indices
    return S_tilde_a
```

## I    VISUALIZATION OF FADING AUDIO-VISUAL ATTENTION

As shown in Figure 2 of the main paper, we tackle the problem of forgetting past audio-video correlation by visualizing the attention maps. In Figure 17, we provide additional examples that vividly illustrate the challenge of forgetting past correlation as the model undergoes pre-training on sequential tasks.

In the top-left example of Figure 17, we observe a video example where a person is engaged in rope skipping. The initial attention map concentrated on the feet ((b)). However, as the model adapts to new tasks, the attention map is shifted solely to the person's face ((c)), implying the gradual erosion of the correlation between the sound of rope skipping and the corresponding jumping motion. In the top-right example of Figure 17, the attention map undergoes an intriguing shift towards an unrelated caption in the first two frames ((c)). Moving on to the middle-left example in Figure 17, the model initially demonstrates a keen understanding of the xylophone's location where the sound originates ((b)). However, subsequent training on additional tasks weakens auditory attention, and the model fails to locate the sounding region ((c)). This challenge becomes more pronounced when multiple sounding objects are involved. In the middle-right example in Figure 17, we explore a scenario where a child is singing alongside a man playing the guitar. The initial visual attention map correctly identifies both the guitar and the child's mouth. Nevertheless, as the model undergoes continuous training, the correlation between the singing voice and the child's visual presence diminishes, and the model connects the sound with the guitar only ((c)). Similarly, in the bottom-left example of Figure 17, the visual attention map shifts from the horse to the human, accompanied by the weakening of auditory attention towards the horse's clip-clop sound ((b)). Lastly, in the bottom-right example of Figure 17, despite the presence of only one prominent sounding object, the bird, the visual attention map is activated at the uncorrelated object. However, our approach successfully mitigates this forgetting problem, as demonstrated in (d) of the example, where the attention maps remain consistent with the initial attention maps.

## J    VISUALIZATION OF SPURIOUS AUDIO-VISUAL ATTENTION BY PAST DATA

In the main paper, we discuss the potential causes of catastrophic forgetting of continual audio-video representation learning. The observation in Figure 3 suggests that during the continual pre-training, the model may learn spurious audio-video correlation for new data distributions based on learned multimodal knowledge, and the current spurious correlation may overwrite the previously learned correlation. In Figure 16, we provide various examples in order to support our findings.

In the top-left and top-right examples of Figure 16, when exposed to audio involving human sports activities, attention maps shift from musical instruments ((b)) to human faces in visual regions ((c)). Similarly, in the middle-left example, when given an audio sample of human activities, the visual attention map shifts from the vehicle ((b)) to the crowds of people ((c)). In the middle-right and bottom-left examples, the visual attention maps initially focus on the relevant objects for the corresponding audio ((b)). However, when given completely unrelated audio, the attention maps shift to meaningless regions ((c)). The bottom-right example shows human conversation and volcanic eruption. For the current audio-video pair, the attention map is activated in response to both elements, with the audio attention map primarily focusing on the volcanic eruption sound ((b)). However, when given a video that consists only of human conversation, the audio attention map shifts to the timeline where the human conversation happened in the current pair ((c)).

## K    LIMIATIONS

Our approach involves an extra inference step for patch selection, leveraging the AVM module on top of the backbone model. While this significantly reduces GPU memory consumption, it does incur additional computational overhead, yielding a relatively small improvement in throughput. To address this challenge, one potential solution is to develop a student model that integrates the AVM module and utilizes knowledge distillation to transfer audio-video representation from the backbone model. Recognizing the importance of enhancing efficiency, we acknowledge the necessity for future research to explore effective strategies for leveraging the AVM module. This avenue for improvement is a key component of our future research agenda.

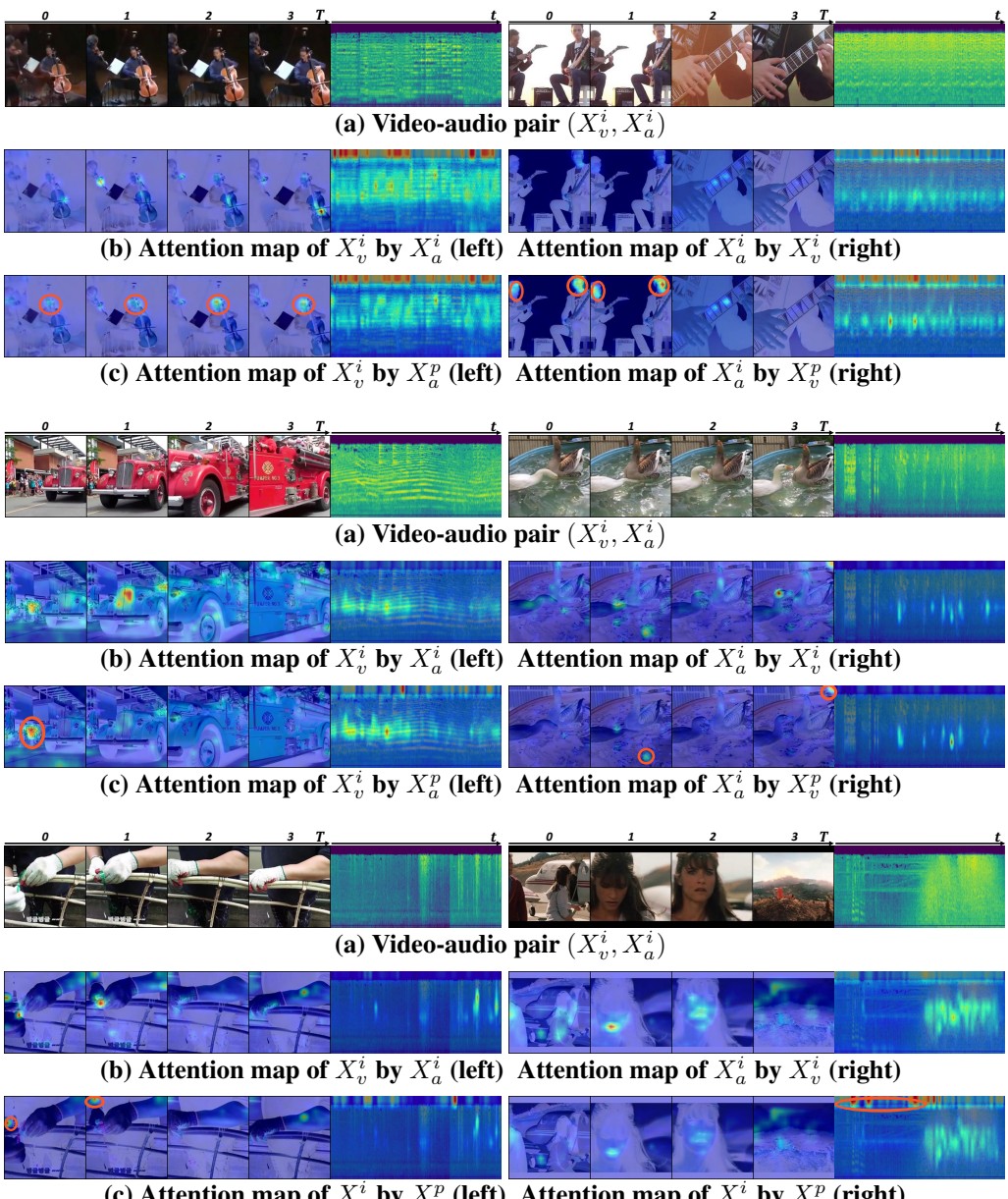

Figure 16: **Examples of false attention by past data**: **(a)**: Visualization of video (left) and audio (right) pairs. **(b)**: Audio-to-video attention map (left) and video-to-audio attention map(right). **(c)**: Attention maps by past audio (left) and past video (right). The area of spurious correlation is marked in the orange circle.

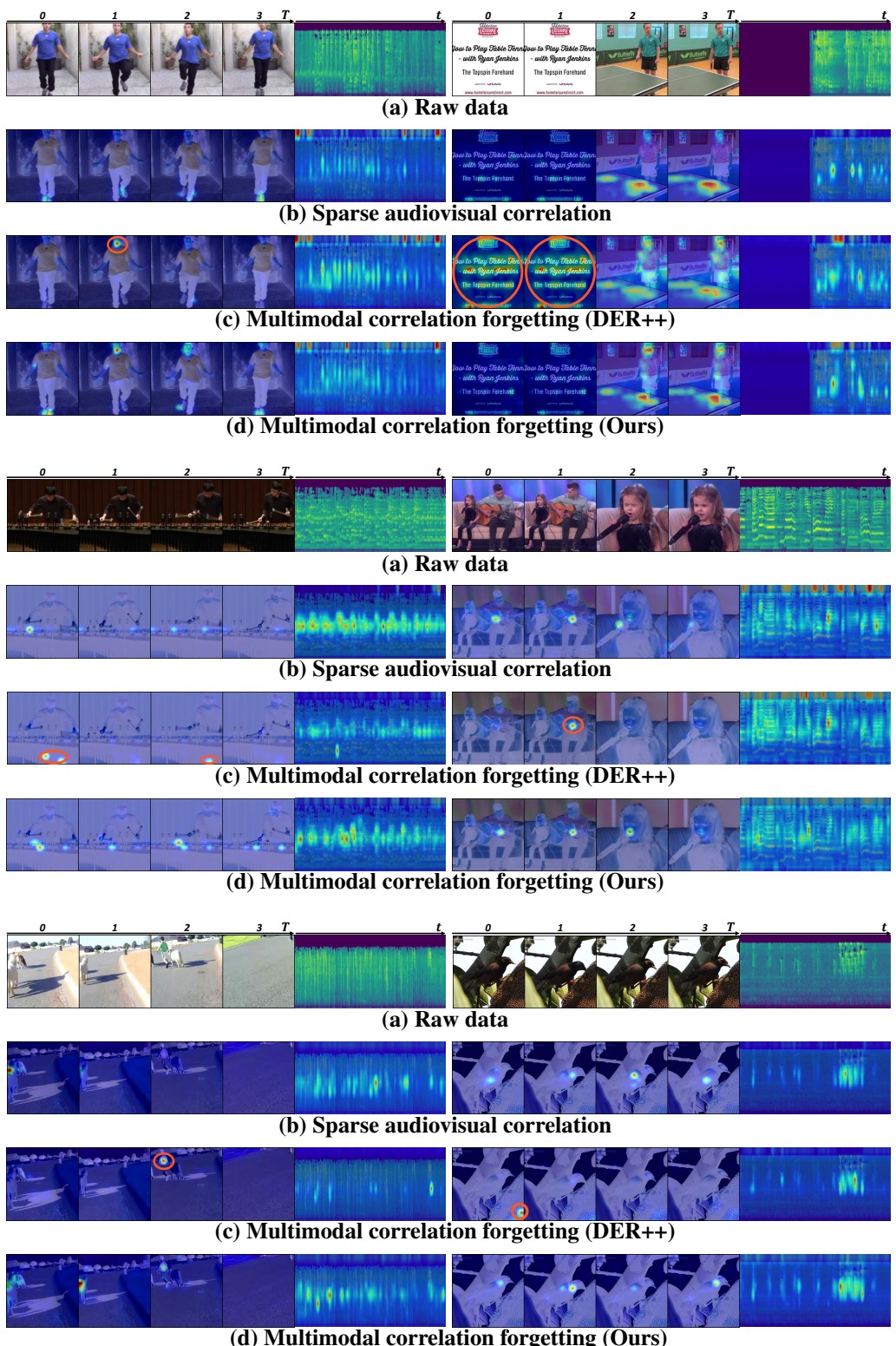

Figure 17: **Visualization of cross-attention maps. (a)** Examples of raw data pairs. We visualize cross-attention maps of the pairs in **(b)**. The closer the color is to red, the higher the attention score. While the baseline model using DER++ attends to entirely different parts as can be seen in **(c)**, our method attends to a similar part even after being trained on two additional tasks as presented in **(d)**. The wrong attention region is marked in an orange circle.

