# OpenReview forum: "Lifelong Audio-video Masked Autoencoder with Forget-robust Localized Alignments"
_ICLR.cc/2024/Conference — Submitted to ICLR 2024_

### Official Review · Reviewer_4AjC · 2023-11-01

**Soundness:** 2 fair
**Presentation:** 2 fair
**Contribution:** 3 good
**Rating:** 5
**Confidence:** 4

**Summary:**

This paper proposes two key challenges in learning audio-video data with continuously changing semantic categories: sparse spatiotemporal correlation and representational forgetting of audio-video relationships. This paper further proposes a framework for lifelong learning in audio-visual scenes, named FLAVA, it contains two important components: (1) A lightweight trainable audio-video matching (AVM) module, which performs cross-modal attention operation to obtain cross-modal similarity. (2) A rank-based forget-robust patches selection module. Experiments on multiple audio-visual datasets demonstrate the effectiveness of the proposed method.

**Strengths:**

+ Lifelong learning in audio-visual scenes is a very meaningful research topic.
+ The proposed two challenges (sparse spatiotemporal correlation and representational forgetting of audio-video relationships) are interesting and they can bring some insights to our community.
+ Extensive experiments show the effectiveness of the proposed method.

**Weaknesses:**

- The paper does not introduce the proposed method very clearly, the writing of the paper should be polished.
- The paper claims that "this is the first work that addresses a continual representation learning problem for audio-video tasks and identifies the crucial challenges that the new problem has." Hence, previous works about audio-visual continual learning (e.g., audiovisual continuous speech recognition,  audiovisual continuous emotion recognition) should be introduced in the related work part. The differences between this paper and previous works about multimodal continual learning should also be further refined.
- Experiments in section 3.2 are not convincing enough, unpaired data naturally creates misleading cross-modal attention maps. These experiments do not fully explain the reason for representational forgetting.
- The paper says: "In order to assess the relative importance of current data and past data, we further compute past-data-induced attention maps." However, in continual learning, past data is usually unavailable, so how to compute past-data-induced attention maps? This is a very important question and it should be explained in detail.

**Questions:**

Pls see the weakness part.

---

> ### Author Response · Authors · 2023-11-17
>
> Thank you for your review and your constructive comments. During the rebuttal period, we have made every effort to address your concerns. The detailed responses are below:
>
> ---
>
> > **Weakness 1: The paper does not introduce the proposed method very clearly, the writing of the paper should be polished.**
>
> $\rightarrow$ Thank you for your opinion. We have made efforts to more clearly describe our approach in sections 4.2 and 4.3, as well as Figure 4, and update them in our latest revision. We add a line indicator of Equation 4 to corresponding sentences to help you understand the algorithm of forget-robust patch selection in Section 4.2. In the below, we follow Section 4.2 and Section 4.3 step-by-step to provide more details.
>
> Section 4.2 explains **the process of obtaining the current and past attention map information reflecting inter-task (past & current) importance** (*Section 4.2, paragraph 1, line 16*). Using this importance, the forget-robust patch selection aims to prune patches from current data that have a higher correlation with previous steps. The purpose of this pruning is to retain the past correlation knowledge while effectively learning new data (*Section 3.2, line 11~16*).
>
> In Section 4.3, **we utilize the importance scores and pruning probability matrices to find the forget-robust and informative audio and video patches** (*Section 4.3, paragraph 1, line 1* and *Section 4.3, paragraph 3, line 1*). In the case of the audio, we have to preserve local correlation among audio patches due to the temporal continuity in the spectrogram (*Section 4.3, paragraph 1, line 2*). Hence, we follow the process below.
>
> 1. Process the audio importance score ($\textbf{I}\_{a}$) and audio pruning indicator ($\textbf{F}\_{a}$) in time-wise to obtain $\textbf{I}^{t}\_{a}$ and $\textbf{F}^{t}\_{a}$. (Section 4.3, paragraph 1).
>
> 2. Group $\textbf{I}^{t}\_{a}$ in time chunks with length $\textbf{L}\_c$ to preserve the time continuity information. This grouping process generates a chunk-wise importance score ($\textbf{I}^{c}\_{a}$). (Section 4.3, paragraph 2, line 1~4)
>
> 3. Based on $\textbf{I}^{c}\_{a}$, we employ multinomial probability distribution to sample the time chunks that are most likely to be informative. (Section 4.3, paragraph 2, line 5)
>
> 4. Iterate through the sampled time chunks and accumulate the number of selected audio patches while excluding the forget-inducing audio patches referring to $\textbf{F}^{t}\_{a}$.
>
> 5. We iterate until the number of selected audio patches reaches our target number, which is $\kappa\_{a}$. (Section 4.3, paragraph 2, line 6-7)
>
> For video patches are selected in a much simpler manner.
>
> 1. For every element in the video importance score ($\textbf{I}\_{v}$) that aligns with indices marked as True in the video pruning indicator ($\textbf{F}\_{v}$), we set the value to zero. This process is to exclude the forget-inducing video patches in our video patch selection, yielding $\tilde{\textbf{I}\}_{v}$. (Section 4.3, paragraph 3, line 4)
>
> 2. Apply a multinomial probability distribution on $\tilde{\textbf{I}}\_{v}$ to select video patches that are most likely to be informative. (Section 4.3, paragraph 3, line 6)
>
> ---

---

> ### Author Response · Authors · 2023-11-17
>
> > **Weakness 2: The differences between this paper and previous works about multimodal continual learning should also be further refined.**
>
> $\rightarrow$ To our current knowledge, investigations into audiovisual continual learning have emerged relatively recently [1, 2]. Since the recent works tackle the problem of audiovisual continual learning in a supervised learning setup (*Related Work, paragraph 2, line 10*), the claim that our work represents a new exploration into continual audio-video representation learning is valid.
>
> As far as we understand, the suggested works (audiovisual continuous speech recognition [3] or audiovisual continuous emotion recognition [4]) seek to automatically predict the labels in a temporally continuous manner, which are not audiovisual continual learning and thus not related to our work.
>
> The primary difference between the two recent works[1, 2] and our work lies in the focus: **the former focuses on audio-visual supervised continual learning** where the model is continuously trained on the audiovisual tasks with class labels. Conversely, **our work focuses on audio-visual unsupervised continual learning** where the model is continuously pre-trained with self-supervised loss terms and the pre-trained model is estimated on various downstream tasks.
>
> For a more comprehensive grasp of these distinctions, we have included Figure 7 and *Appendix B, Comparison between SCL setup and UCL setup* in our latest revision. This section delineates and contrasts supervised continual learning with unsupervised continual learning, facilitating a clearer understanding of our approach.
>
> [1] Mo et al,. Class-incremental grouping network for continual audio-visual learning, ICCV 2023.
> [2] Pian et al., Audio-visual class-incremental learning, ICCV 2023.
> [3] Rouditchenko et al., AV-CPL: Continuous Pseudo-labeling for Audio-Visual Speech Recognition
> [4] Zhang et al., Continuous Emotion Recognition with Audio-visual Leader-follower Attentive Fusion, ICCV workshop 2021
>
> ---
>
> > **Weakness 3:  Experiments in section 3.2 are not convincing enough, unpaired data naturally creates misleading cross-modal attention maps. These experiments do not fully explain the reason for representational forgetting.**
>
> $\rightarrow$ It is crucial to highlight that **the misalignment of past-audio-guided video attention maps are located in misleading locations where uncorrelated sound sources exist** (*Section 3.2, line7*). Consider Figure 17 top row: despite the presence of musical instruments in the visual scene, when presented with uncorrelated human voice audio data, the audio-guided video attention maps primarily focus on human features. **Originally, such misalignment might not pose significant challenges in conventional audio-video representation learning**, as **static and generalized data distribution** typically aids the model in distinguishing between different semantics (human-playing-musical-instrument, human-conversation, etc) that share similar visual or audio characteristics (human features).
>
> **However, within the continual audio-video representation learning paradigm, these misalignments significantly deteriorate the learned representation.** During continual pre-training, the model encounters **shifting and biased data distributions**. Consequently, training on the current task leads to the weakening of the audio-visual correlations from the previous tasks. For example, when the current task involves data solely featuring humans playing musical instruments, the model would potentially link auditory data of musical sounds to visual data containing either humans or musical instruments. **If the past task consists of human-conversational data, training on the current task leads to the weakening of the audio-visual correlation of human visuals and human voice learned from the past task.** Instead, the model potentially connects between human visual features and musical sound features from the current ask. This leads to the forgetting of the audio-video representation learned from the past task. To address this issue, we prioritize selecting forget-robust patches by comparing the current-data-induced cross-attention map and the past-data-induced cross-attention map. (*Section 4.2*)
>
> ---

---

> ### Author Response · Authors · 2023-11-17
>
> > **Weakness 4: In continual learning, past data is usually unavailable, so how do we compute past-data-induced attention maps? This is a very important question and it should be explained in detail.**
>
> $\rightarrow$ Rehearsal-based continual learning, where the model temporally stores a small fraction of previous task data in a small-sized buffer during pre-training, is one of the most effective and popular/well-known directions in CL tasks [1,2,3,4], achieving a good trade-off between performance and efficiency. And we follow the same assumption since this is also valid in our scenario.
>
> From the rehearsal memory, we use the past data to compute past-data-induced attention maps. **Specifically, we decide to exploit the past discriminative queries stored in the rehearsal memory, which was calculated during the pre-training of the past tasks** (*Section 4.2, line 15*), since processing the raw past data to compute the past attention maps could be a waste of computational resources. Storing the past information only increases negligible amounts of memory (*Section 5.2, paragraph 2, line 8*).
>
> The computation of past-data-induced attention maps is as follows:
>
> 1. We use importance scores ($\textbf{I}\_{a}, \textbf{I}\_{v}$) from *Section 4.1* to extract patch indices whose patches have high correlation with their modality pair. This process yields sorted indices for audio and video, respectively. ($\textbf{S}\_{a}, \textbf{S}\_{v}$) (*Section 4.2, line 8*)
>
> 2. With the sorted indices, we collect discriminative keys ($\hat{\textbf{k}}\_{a}, \hat{\textbf{k}}\_{v}$) from the key embeddings ($\textbf{k}\_{a},\textbf{k}\_{v}$) of the current data. (*Section 4.2, line 10*)
>
> 3. Similarly, we also collect discriminative queries from the query embeddings ($\textbf{q}\_{a},\textbf{q}\_{v}$) of the current data. We subsequently apply weighted mean on the collected queries using the importance scores as weight. This process yields the final discriminative queries ($\hat{\textbf{q}}\_{a}, \hat{\textbf{q}}\_{v}$). (*Section 4.2, line 11*)
>
> 4. Using the discriminate keys and queries in 2. and 3., we compute the current-data-induced attention maps ($\hat{\textbf{A}}\_{a}, \hat{\textbf{A}}\_{v}$) that represent the importance of each current patch to the current modality pair. (*Section 4.2, line 13*)
>
> 5. We sample the past discriminative queries ($\hat{\textbf{q}}\_{a}^{p}, \hat{\textbf{q}}\_{v}^{p}$) from the rehearsal memory, which were stored during the pre-training of the past tasks.
>
> 6. Using the queries, we compute the past-data-induced attention maps by combining the discriminative keys from current data ($\hat{\textbf{k}}\_{a}, \hat{\textbf{k}}\_{v}$) and the past queries ($\hat{\textbf{q}}\_{a}^{p}, \hat{\textbf{q}}\_{v}^{p}$). (*Section 4.2, line 15*) The past-data-induced attention maps exhibit the importance of each current patch to the past modality pair.
>
> We have added a line indicator of Equation 4 to corresponding sentences to help you understand the process of calculating the past-data-induced attention maps in Section 4.2 in our latest revision.
>
> [1] Jin et al., Gradient-based Editing of Memory Examples for Online Task-free Continual Learning, NeurIPS 2021
> [2] Yoon et al., Online Coreset Selection for Rehearsal-based Continual Learning, ICLR 2022
> [3] Caccia et al., New Insights on Reducing Abrupt Representation Change in Online Continual Learning, ICLR 2022
> [4] Jeeveswaran et al., BiRT: Bio-inspired Replay in Vision Transformers for Continual Learning, ICML 2023
>
> ---

---

> ### Author Response · Authors · 2023-11-21
> **Dear Reviewer 4AjC - A Gentle Reminder**
>
> Dear Reviewer 4AjC,
>
> We are sincerely grateful to you for reading our response.
>
> > During the rebuttal period,
> - We have refined the content in *Section 4* for a better presentation of our proposed method. Especially, we have added a line indicator of Equation 4 to corresponding sentences to improve readability and clarify the description of the method line by line.
> - We have also provided a step-by-step procedure in Section 4.3  to help you catch a comprehensive grasp of the patch selection mechanism.
> - We have clarified differences between the recent related works concerning audio-video supervised continual learning. **The recent works focus on audio-visual supervised continual learning while our work focuses on audio-visual unsupervised continual learning.** For a more comprehensive grasp of these distinctions, we have included Figure 7 and *Appendix B, Comparison between SCL setup and UCL setup* that contrasts supervised continual learning with unsupervised continual learning, facilitating a clearer understanding of our approach.
> - We have provided examples where the misleading cross-modal attention maps, which originally might not pose a significant challenge in audio-video representation learning, **deteriorate the learned representation in the continual audio-video representation learning scenarios.** We believe this supports the forgetting of audio-video correlation issues that we raised through Section 3.2.
> - We have clarified that **rehearsal-based continual learning is one of the popular and standard approaches for solving continual learning.** We have further addressed your concerns about the past-data-induced attention maps by clarifying **the use of the past discriminative queries stored in the rehearsal memory to compute the maps**, providing explanations on each step of the procedure to help you understand our process.
>
> We remain committed to further improving the quality of our paper by addressing any remaining concerns and suggestions where necessary. With that in mind, If you might have any further feedback, please let us know. We would be grateful for the opportunity to address them and make our work a more solid and valuable contribution to the field of audio-video continual representation learning.
>
> **Also, we would like to kindly suggest your reconsideration of the rating, if you feel that our work does not have major concerns with respect to evaluation, resources, reproducibility, and ethical considerations.** We understand that the criteria for rating a paper can sometimes be subjective; however, we believe that most of your concerns are effectively addressed as long as there are no major issues.
>
> We thank you so much for your time and effort in reviewing our paper, and your constructive feedback that has greatly contributed to improving our paper.
>
> Warm Regards,
> Authors

---

> ### Author Response · Authors · 2023-11-22
> **Thank you for your review; Today is the end of the discussion phase.**
>
> Dear Reviewer 4AjC,
>
> We sincerely appreciate your efforts in reviewing our paper, and your constructive comments. We have responded to your comments and faithfully reflected them in the revision.
>
> As you know, now we have only one day to have interactive discussions. Could you please go over our responses and the revision since the end of the final discussion phase is approaching? Please let us know if there is anything else we need to clarify or provide.
>
> Best,
> Authors
>
> ---

---

> ### Author Response · Authors · 2023-11-23
> **Discussion phase ends within 8 hours**
>
> Dear Reviewer 4AjC,
>
> We really appreciate your effort in reviewing our submission again. Since the discussion period for ICLR 2024 ends within 8 hours, we politely ask you to read our new responses by any chance. Please understand that we have made our best effort to address your concerns during this period.
>
> Also, we would like to kindly suggest **a reconsideration of the initial rating (reject: 5)**, if you agree that **most concerns/misunderstandings raised by the initial review are resolved**. We strongly believe that most of your concerns are effectively addressed as long as there are **no significant issues to be negative**.
>
> We thank you so much for your time and effort in reviewing our paper and for the constructive feedback that has greatly improved it.
>
> Best,
> Authors

---

### Official Review · Reviewer_YJJY · 2023-11-05

**Soundness:** 3 good
**Presentation:** 3 good
**Contribution:** 2 fair
**Rating:** 5
**Confidence:** 4

**Summary:**

This work tackles the continual learning of audio-video representations with attention-based localized alignment and forget-robust multimodal patch selection strategies. Experiments on VGG-sound and AudioSet show its effectiveness.

**Strengths:**

The general structure is clear. The method is simple in general. It’s easy to follow. The performance of both the two proposed modules is obvious. This work also provides a thorough analysis of both the method and experiment.

**Weaknesses:**

I haven't specifically focused my research on lifelong learning, but from a methodological perspective, this work primarily involves the direct utilization of basic attention mechanisms. The measurement of relative importance is determined by the levels of attention results, which is a relatively common approach.

**Questions:**

(1) The best performance on VGGSound should be at least 66.+% now, which is much higher than 58.65% here. What’s the main reason of this gap? Due to that there are more supervised labels in those works? If so, what will the performance be like of this work if we also provide labels? Will it also be improved by a large margin to close to 70%?
(2) Beyond the proposed specific trainable module, is it possible to introduce this proposed manner into the learning of large-scale audio-visual models? What would be the results like of using this strategy to train large-scale models? Will is lose its effect when coming to large-scale data and models?

---

> ### Author Response · Authors · 2023-11-17
>
> Thank you for your review and your constructive comments. During the rebuttal period, we have made every effort to address your concerns. The detailed responses are below:
>
> ---
>
> > **Weakness 1: I haven't specifically focused my research on lifelong learning, but from a methodological perspective, this work primarily involves the direct utilization of basic attention mechanisms, which is relatively common.**
>
> $\rightarrow$ We respectfully disagree with the Reviewer's claim that our approach simply relies on a basic attention mechanism. First, we emphasize that the utilization of attention maps as a versatile tool for solving a variety of problems has been extensively validated in many studies [1,2,3]. The **appropriate/effective utilization of attention mapping is the core originality of these studies,** and the novelty of our proposed approach lies in utilizing this tool to solve the problem of learning continuous audio-video representations in an environment where task identity is not provided.
>
> In particular, our method is sufficiently novel with multiple unique contributions, and leveraging attention maps is only a small part of our methodology. From the motivation and problem statement perspectives, we first clarify the critical challenges in multimodal (audio-video) continual pre-training; *1) sparse spatiotemporal correlation between the audio-video pairs*, and *2) representational forgetting of audio-video relationships*. Furthermore, we discover the potential root causes of the representational forgetting by visualizing the cross-attention maps induced by past data (*Section 3.2*) and use these observations in the attention maps to formulate our approach.
>
> Next, from a methodology perspective, our FLAVA contains multiple unique and meaningful core components. To solve the challenge of *1) sparse spatiotemporal correlation between the audio-video pairs*, we actively leverage the cross-attention information to compute importance score matrices to locate the highly correlated audio(video) patches with respect to their video(audio) data. **Pre-training with the selected patches from the importance score enables the model to achieve better audio and video clustering within the multi-modal representation space**, as we analyzed in *Appendix G, paragraph 4*.
>
> To address the challenge of *2) representational forgetting of audio-video relationships*, we compute inter-task (current-past) importance by estimating a cross-modal attention map of current data from past data (*Section 4.2*). While various continual learning approaches leverage the attention maps from the past model, which is trained up to the previous task, to perform knowledge distillation or regularization for the purpose of alleviating forgetting [3][4], **we have our own originality that we extend to investigating the impact of past tasks on the current task attention map activations.** To estimate the impact, we exploit the query embeddings from past data stored in the rehearsal memory to activate cross-attention maps on the current data, and compare the past-data-induced cross-attention maps with the current-data-induced cross-attention maps for selecting forget-robust patches among the current data. This strategy significantly improves forgetting robustness during multimodal continual pre-training.
>
> Furthermore, **our method harnesses a video-guided audio cross-attention map and introduces an approach to retain the local correlation information during patch pruning.** (*Section 4.3*). Although the audio-guided video cross-attention map has been commonly used in various fields [3], the video-guided audio cross-attention map and its utilization have been understudied. We analyze the audio-guided video cross-attention map and suggest a way of computing it in order to preserve the local correlation in the audio spectrogram. **We also conducted an ablation study of our proposed audio patch selection to give more insights into audio patch selection (*Appendix E, Audio patch selection strategy*).**
>
> [1] Liu et al., Swin Transformer: Hierarchical Vision Transformer using Shifted Windows, ICCV 2021
> [2] Lee et al., Sparse Token Transformer with Attention Back Tracking, ICLR 2023
> [3] Pian et al., Audio-Visual Class-Incremental Learning, ICCV 2023
> [4] Pelsin et al., Towards Exemplar-Free Continaul Learning in Vision Transformers: an Account of Attention, Functional and Weight Regularization, CVPR workshop 2022
>
> ---

---

> ### Author Response · Authors · 2023-11-17
>
> > **Question 1: The best performance on VGGSound should be at least 66.+% now, which is much higher than 58.65% here. What’s the main reason of this gap?**
>
> $\rightarrow$ **This disparity arises due to fundamental differences in the experimental setups.** Comparing the outcome of our continual representation learning directly with results from current representation learning doesn’t yield a meaningful one-to-one comparison.
>
> The 66.+% accuracy stems from models that are trained with a full dataset, whereas our reported accuracy is based on a continual learning scenario. In our approach, we use a subset of the dataset and split it into various tasks to train the model on the sequence of tasks.
>
> For clearer comprehension of continual learning, we have included Figure 7 and provide details of unsupervised continual learning in *Appendix B, Comparison between SCL setup and UCL setup* in our latest revision
>
> ---
>
> > **Question 2:  Is it possible to introduce this proposed manner into the learning of large-scale models? What would be the results like of using this strategy to train large-scale models?**
>
> $\rightarrow$ **Our approach has the potential to enhance the efficiency and sustainability of large-scale audio-video models.**
>
> Our approach processes the unlabeled videos to learn audio-video correlations with much smaller GPU memory occupancy (Table 2). Training large-scale models requires processing web-scale data and huge computational costs. Hence, the large-scale model can efficiently learn audio-video knowledge from web-scale video data without human intervention.
>
> Moreover, for the large-scale models to be sustainable in the future, the models should continually learn new knowledge. In our approach, we continually pre-train the audio-video model efficiently by selecting informative and forget-robust patches from the current task. Hence, **our approach exhibits high adaptability to new concepts by catching audio-video correlations from sparse spatiotemporal correlations in videos while preserving past audio-video knowledge by learning forget-robust patches in the videos.**
>
> ---

---

> ### Author Response · Authors · 2023-11-21
> **Dear Reviewer YJJY - A Gentle Reminder**
>
> Dear Reviewer YJJY,
>
> We are sincerely grateful to you for reading our response.
>
> > During the rebuttal period,
> - We have clarified the uniqueness of our work. We would like to emphasize that **the recent studies using attention maps have their core originality in appropriate and effective utilization of attention maps.** We have further emphasized that our method selects patches based on cross-attention map information that empirically results in achieving better audio and video clustering within the multi-modal representation space. Especially, **we have our own originality that we extend to investigating the impact of past tasks to the current task attention map activations to solve representational forgetting issues in continual audio-video representation learning.** We have also listed the uniqueness of our method in the main discussion.
> - We have clarified that **the disparity of the performance arises due to fundamental differences in the experimental setups.** The one you referred to follows conventional self-supervised learning on static datasets, while our setting follows a continual representation learning setup in which the model is self-supervised with consecutive tasks. For clarification, we have addressed Figure 7 and details of unsupervised continual learning in *Appendix B, Comparison between SCL and UCL setup*.
> - We have discussed that **our approach has the potential to enhance the efficiency and sustainability of large-scale audio-video models.** Our approach exhibits high adaptability to new concepts by catching audio-video correlations from sparse spatiotemporal correlations in videos while preserving past audio-video knowledge by learning forget-robust patches in the videos.
>
> We remain committed to further improving the quality of our paper by addressing any remaining concerns and suggestions where necessary. With that in mind, If you might have any further feedback, please let us know. We would be grateful for the opportunity to address them and make our work a more solid and valuable contribution to the field of audio-video continual representation learning.
>
> **Also, we would like to kindly suggest your reconsideration of the rating, if you feel that our work does not have major concerns with respect to evaluation, resources, reproducibility, and ethical considerations.** We understand that the criteria for rating a paper can sometimes be subjective; however, we believe that most of your concerns are effectively addressed as long as there are no major issues.
>
> We thank you so much for your time and effort in reviewing our paper, and your constructive feedback that has greatly contributed to improving our paper.
>
> Warm Regards,
> Authors

---

> ### Author Response · Authors · 2023-11-22
> **Thank you for your review; Today is the end of the discussion phase.**
>
> Dear Reviewer YJJY,
>
> We sincerely appreciate your efforts in reviewing our paper, and your constructive comments. We have responded to your comments and faithfully reflected them in the revision.
>
> As you know, now we have only one day to have interactive discussions. Could you please go over our responses and the revision since the end of the final discussion phase is approaching? Please let us know if there is anything else we need to clarify or provide.
>
> Best,
> Authors
>
> ---

---

> ### Author Response · Authors · 2023-11-23
> **Discussion phase ends within 8 hours**
>
> Dear Reviewer YJJY,
>
> We really appreciate your effort in reviewing our submission again. Since the discussion period for ICLR 2024 ends within 8 hours, we politely ask you to read our new responses by any chance. Please understand that we have made our best effort to address your concerns during this period.
>
> Also, we would like to kindly suggest **a reconsideration of the initial rating (reject: 5)**, if you agree that **most concerns/misunderstandings raised by the initial review are resolved**. We strongly believe that most of your concerns are effectively addressed as long as there are **no significant issues to be negative**.
>
> We thank you so much for your time and effort in reviewing our paper and for the constructive feedback that has greatly improved it.
>
> Best,
> Authors

---

### Official Review · Reviewer_hYwV · 2023-11-07

**Soundness:** 3 good
**Presentation:** 2 fair
**Contribution:** 3 good
**Rating:** 8
**Confidence:** 2

**Summary:**

In this paper, the authors propose a lifelong audio-visual masked autoencoder model: FLAVA.  It can continually learn multimodal representations from a video stream containing audio-video pairs, even while the distribution of the data continually shifts over time. FLAVA addresses the challenges of continual audio-video representation learning by proposing two novel ideas: Localized Alignment and Forget-robust multimodal patch selection. FLAVA outperforms the state-of-the-art continual learning methods on several benchmark datasets in continual audio-video representation learning.

***Post-rebuttal***

Thank the authors for responding to my questions! My major concerns have been addressed.

**Strengths:**

+ Self-supervised audio-visual continual learning is an important topic in multimodal learning, and this work addresses the issue of forgetting in such scenarios.

 + The authors clearly motivate the need for their work and provide vivid examples of the audio-visual alignment of forgetting issues.

+ The proposed method outperforms compared continual learning approaches on several benchmark datasets.

**Weaknesses:**

+ The paper writing can be further improved. Sections 4.2 and 4.3 are difficult to follow. Please clarify the following:
(1) What do you mean by "relative importance of current data and past data"?
(2) How is past data used in Section 4.2?
(3) How are past discriminative queries selected?
(4) With increasing continual learning steps, how can past data from previous steps be better leveraged to improve memory usage?
Why can the proposed method tackle the issues mentioned in Figures 1, 2, and 3?

+ I saw the authors use a fixed task order for continual pre-training. I wonder whether the order matters.

+ Two concurrent related works [1, 2] have addressed audio-visual continual learning, the second of which also observed and addressed audio-visual alignment forgetting issues. The authors can discuss the relevance and differences among these works in more detail, especially the differences between the proposed method and the second work. Although it is clear that the works are concurrent, more discussions would be helpful to distinguish between the different works.

+ On the audiovisual classification task, why are the improvements of the proposed method marginal?


[1] Mo, Shentong, Weiguo Pian, and Yapeng Tian. "Class-incremental grouping network for continual audio-visual learning." Proceedings of the IEEE/CVF International Conference on Computer Vision. 2023.

[2] Pian, Weiguo, et al. "Audio-visual class-incremental learning." Proceedings of the IEEE/CVF International Conference on Computer Vision. 2023.

**Questions:**

See Weaknesses.

---

> ### Author Response · Authors · 2023-11-17
>
> Thank you for your review and your constructive comments. During the rebuttal period, we have made every effort to address your concerns. The detailed responses are below:
>
> ---
>
> > **Weakness 1-1: What do you mean by “relative importance of current and past data” in Section 4.2?**
> > **Weakness 1-2: How is past data used in Section 4.2?**
>
> $\rightarrow$ In Section 4.2, we outline our proposed method for selecting forget-robust patches (*Section 4, line 4*). We sample past data from the rehearsal memory, containing past discriminative queries, importance scores, and pruning probability matrices (Algorithm 1, line 6). To find audio and video patches that are semantically closer to the past tasks than the current task, we suggest exploiting attention maps using the current and past data. **We compute the importance of each patch to the past and current tasks by comparing the past and current attention maps, which is used for selecting forget-robust patches for current data.**
>
> ---
>
> > **Weakness 1-3: How are past discriminative queries selected?**
>
> $\rightarrow$ We utilize reservoir sampling that randomly samples and replaces past instances during training. **Hence, we randomly sample past audio-video pairs together with past discriminative queries, importance scores, and pruning probability matrices from the rehearsal memory** during the continual pre-training phase, following the random sampling in our baselines (e.g., LUMP, DER++, CLS-ER, ...). We sincerely apologize for the confusion that arose due to our oversight. To rectify this, we have made updates to *Section 5.1, Baselines, line 4* of our latest revision.
>
> ---
>
> > **Weakness 1-4: With increasing continual learning steps, how can past data from previous steps be better leveraged to improve memory usage?**
>
> $\rightarrow$ Improving the rehearsal memory usage could be one of the important problems in rehearsal-based continual learning since an efficient way of storing past data could enrich the diversity within the rehearsal memory that has memory constraints. This becomes much more important in audio-video continual learning since video data size is much larger than the size of an image. **One way of achieving efficient rehearsal memory usage is to store the pruned data during the continual pre-training**, rather than preserving raw data within the memory. During the rebuttal period, we explored this approach and summarized the zero-shot retrieval task results in the Tables below (*acc* indicates average accuracies, *fgt* indicates average forgetting). FLAVA+ implements our patch selection strategy while its memory stores the selected patches instead of the raw data.
>
> Comparison between FLAVA and FLAVA+ (first, and second rows in the Tables) elucidates a clear trade-off associated with storing pruned data in the rehearsal memory. Despite the effectiveness of our method in selecting informative patches from audio-video data, pruning the patches would result in a loss of the data’s contextual information. Consequently, replaying the pruned data from the memory yields a weakened effect in retaining past knowledge due to the diminished total amounts of information. However, this strategy significantly saves nearly half of the memory during continual pre-training while remaining highly competitive compared to other baselines. **Thus, in scenarios where the model faces stricter memory constraints, FLAVA+ presents itself as a potentially advantageous approach.**
>
> Moreover, when we increase the number of past instances in FLAVA+ to align with the memory size of FLAVA, FLAVA+ outperforms FLAVA by a large margin (first, and third rows in the Tables). This significant improvement arises from the augmented volume of data in the rehearsal memory, **allowing the memory to effectively maintain audio-video knowledge from past tasks, simultaneously enhancing data diversity in current task training.**
>
> This approach offers an orthogonal direction to our original contribution, demonstrating our method's potential to efficiently utilize rehearsal memory.
>
> VGGSound retrieval task
>
> | Method   | Compress | # of past data | Memory size (GB) | A$\rightarrow$V acc | A$\rightarrow$V fgt | V$\rightarrow$A acc | V$\rightarrow$A fgt |
> |---------------|------------|-----------|-----------|-----------|-----------|-----------|-----------|
> | FLAVA | x | 2k | 5.63 | 14.16 | **4.38** | 14.07 | 4.65 |
> | FLAVA+ | o | 2k | 2.89 | 13.29 | 6.11 | 13.07 | 6.50 |
> | FLAVA+ | o | 4k | 5.78 | **15.44** | 4.62 | **15.53** | **3.89** |
>
> AudioSet retrieval task
>
> | Method   | Compress | # of past data | Memory size (GB) | A$\rightarrow$V acc | A$\rightarrow$V fgt | V$\rightarrow$A acc | V$\rightarrow$A fgt |
> |---------------|------------|-----------|-----------|-----------|-----------|-----------|-----------|
> | FLAVA | x | 5k | 14.08 | 12.35 | 6.95 | 11.29 | 8.38 |
> | FLAVA+ | o | 5k | 7.23 | 11.56 | 8.25 | 10.21 | 10.48 |
> | FLAVA+ | o | 10k | 14.45 | **15.26** | **5.74** | **14.47** | **6.70** |
>
> ----

---

> ### Author Response · Authors · 2023-11-17
>
> > **Weakness 1-5: Why can the proposed method tackle the issues mentioned in Figures 1,2, and 3?**
>
> $\rightarrow$ Our method mainly focuses on finding locally aligned and forget-robust patches in the raw data from the current task to solve the sparse spatiotemporal correlation and representation forgetting problems suggested in Figures 1,2, and 3 (*Section 4, first paragraph*).
> - Figure 1 illustrates the real-world scenarios where the video data distribution is dynamically shifting, and thus the model should adapt to the new distribution without forgetting the past knowledge when past data is not available. This scenario underscores the necessity for continual audio-video representation learning. Audio-video continual pre-training raises two challenges, which are *1) sparse spatiotemporal correction, and 2) representational forgetting of audio-video relationships* (*Introduction, paragraph 3, line 2*). The first challenge indicates the trait of the video data where only a few regions in the visual scene are correlated with its audio, as we observed in Figure 2. Hence, **we suggest using attention maps as importance scores to statistically sample locally aligned patches** (*Section 4-1*).
>
> - The second challenge is closely related to Figure 2 and Figure 3. In Figure 2, we observe that during continual pre-training, the model forgets the correct audio-video correlation ( *Introduction, paragraph 3, line 5*). In Figure 3, we find the potential root cause of the forgetting: misaligned audio-video correlation between the current and past tasks. This misalignment could lead to learning spurious audio-video correlation, or the past audio-video correlation could be overwritten (*Section 3.2, line 12*). Hence, **we focus on selecting the forget-inducing that aims to prune patches from current data that have a higher correlation with previous tasks** (*Section 4-2*).
>
> ---

---

> ### Author Response · Authors · 2023-11-17
>
> > **Weakness 2: I saw the authors use a fixed task order for continual pre-training. I wonder whether the order matters.
>  I saw the authors use a fixed task order for continual pre-training. I wonder whether the order matters.**
>
> $\rightarrow$ Thank you for your meaningful comment about our experiment. We carry out experiments on shuffled task sequences for continual pre-training, and the results are shown in the below Table (*acc* indicates average accuracy, *fgt* indicates average forgetting). We underscore **the best** and ***the second best*** performances in the Table. **Our method shows competitive or better performance compared to other baselines.** We present the task results in Table 7 of our latest revision. This indicates that **our method is robust under varying conditions**, thereby enhancing the credibility of our analysis.
>
> VGGSound Audio-to-Video Result
>
> | Method | R@1 acc | R@1 fgt | R@5 acc | R@5 fgt | R@10 acc | R@5 fgt | Avg acc | Avg fgt |
> |---------------|------------|-----------|-----------|-----------|-----------|-----------|-----------|-----------|
> | Finetune | 0.80 | 4.15 | 2.96 | 12.23 | 5.05 | 16.91 | 2.94 | 11.10 |
> | ER | 3.89 | 3.06 | 12.10 | 6.55 | 18.30 | 7.74 | 11.43 | 5.78 |
> | MIR | 4.02 | 2.97 | 12.54 | 6.16 | 17.99 | 8.09 | 11.52 | 5.74 |
> | DER++ | 4.23 | 3.35 | 12.92 | 7.31 | 18.62 | 9.45 | 11.92 | 6.70 |
> | GMED | 3.90 | 2.94 | 11.51 | 7.41 | 17.65 | 8.87 | 11.02 | 6.41 |
> | CLS-ER | 3.94 | 3.35 | 12.96| 7.19 | 18.09 | 10.66 | 11.66 | 7.07 |
> | LUMP | 4.06 | **2.18** | 13.21 | **4.66** | 19.34 | **5.58** | 12.20 | **4.14** |
> | FLAVA (Ours) | ***4.72*** | ***2.89*** | ***14.17*** | 5.74 | ***19.94*** | ***5.74*** | ***12.94*** | 4.79 |
> | FLAVA+ (Ours) | **4.90** | 3.19 | **16.42** | ***4.72*** | **23.49** | 5.89 | **14.94** | ***4.60*** |
>
>
> VGGSound Video-to-Audio Result
>
> | Method | R@1 acc | R@1 fgt | R@5 acc | R@5 fgt | R@10 acc | R@5 fgt | Avg acc | Avg fgt |
> |---------------|------------|-----------|-----------|-----------|-----------|-----------|-----------|-----------|
> | Finetune | 0.78 | 3.77 | 3.00 | 11.68 | 5.21 | 15.86 | 3.00 | 10.44 |
> | ER | 3.57 | 2.76 | 11.66 | 7.67 | 16.75 | 10.76 | 10.66 | 7.06 |
> | MIR | 3.35 | 3.15 | 11.37 | 7.74 | 16.62 | 10.11 | 10.45 | 7.00 |
> | DER++ | 4.08 | 3.10 | 12.78 | 9.02 | 18.77 | 11.30 | 11.88 | 7.81 |
> | GMED | 3.42 | 3.80 | 11.45 | 7.76 | 17.06 | 9.94 | 10.64 | 7.17 |
> | CLS-ER | 3.49 | 3.85 | 12.28 | 8.05 | 17.75 | 11.31 | 11.17 | 7.74 |
> | LUMP | 3.98 | **1.67** | 12.44 | **5.17** | 18.11 | **7.27** | 11.51 | **4.70** |
> | FLAVA (Ours) | ***4.18*** | 2.54 | ***13.81*** | 6.56 | ***19.90*** | 8.88 | ***12.63*** | 5.99 |
> | FLAVA+ (Ours) | **5.28** | ***1.81*** | **15.35** | ***6.33*** | **21.97** | ***8.01*** | **14.20** | ***5.38*** |
>
> AudioSet Audio-to-Video Result
>
> | Method | R@1 acc | R@1 fgt | R@5 acc | R@5 fgt | R@10 acc | R@5 fgt | Avg acc | Avg fgt |
> |---------------|------------|-----------|-----------|-----------|-----------|-----------|-----------|-----------|
> | Finetune | 1.50 | 4.72 | 5.49 | 10.41 | 9.80 | 11.91 | 5.60 | 9.01 |
> | ER | 4.52 | 3.16 | 12.72 | 6.93 | 18.83 | 8.00 | 12.02 | 6.03 |
> | MIR | 4.69 | ***2.95*** | 13.22 | 6.50 | 18.98 | 8.81 | 12.30 | 6.09 |
> | DER++ | 4.32 | 4.27 | 12.29 | 8.46 | 18.74 | 10.18 | 11.78 | 7.64 |
> | GMED | 4.70 | **2.48** | 12.56 | ***4.55*** | 18.62 | **5.05** | 11.96 | 4.03 |
> | CLS-ER | ***5.16*** | 2.97 | ***14.33*** | 6.88 | 20.24 | 8.74 | 13.24 | 6.20 |
> | LUMP | 4.45 | 3.40 | 13.05 | 6.25 | 19.45 | 7.28 | 12.32 | 5.64 |
> | FLAVA (Ours) | 4.97 | 3.47 | 13.91 | 5.59 | ***20.30*** | ***6.70*** | ***13.06*** | 5.25 |
> | FLAVA+ (Ours) | **5.77** | 3.90 | **17.51** | **4.49** | **23.72** | 7.07 | **15.67** | ***5.15*** |
>
> AudioSet Video-to-Audio Result
>
> | Method | R@1 acc | R@1 fgt | R@5 acc | R@5 fgt | R@10 acc | R@5 fgt | Avg acc | Avg fgt |
> |---------------|------------|-----------|-----------|-----------|-----------|-----------|-----------|-----------|
> | Finetune | 1.42 | 5.11 | 6.54 | 10.30 | 10.43 | 13.48 | 6.13 | 9.63 |
> | ER | 4.01 | 4.31 | 12.47 | 7.27 | 19.32 | 9.26 | 11.93 | 6.95 |
> | MIR | 4.25 | 3.43 | 12.92 | 6.93 | 19.43 | 9.78 | 12.20 | 6.71 |
> | DER++ | 4.31 | 4.35 | 12.60 | 9.59 | 18.93 | 12.27 | 11.95 | 8.74 |
> | GMED | 4.20 | **1.87** | 12.97 | **6.04** | 19.98 | **8.11** | 12.38 | **5.34** |
> | CLS-ER | 4.85 | 5.48 | 13.37 | 9.17 | 19.69 | 11.36 | 12.64 | 8.67 |
> | LUMP | 4.23 | 4.06 | 13.53 | ***6.09*** | 19.27 | 9.53 | 12.34 | 6.56 |
> | FLAVA (Ours) | ***4.86*** | ***2.92*** | ***14.20*** | 6.41 | ***20.00*** | 9.82 | ***13.02*** | ***6.38*** |
> | FLAVA+ (Ours) | **5.57** | 3.80 | **16.67** | 6.96 | **23.91** | 9.28 | **15.38** | 6.68 |
>
> ---

---

> ### Author Response · Authors · 2023-11-17
>
> > **Weakness 3-1: Two concurrent related works [1, 2] have addressed audio-visual continual learning, the second of which also observed and addressed audio-visual alignment forgetting issues. The authors can discuss the relevance and differences among these works in more detail.**
>
> $\rightarrow$ Thank you for noticing us in the first related work. We have included it in our citation of the latest revision. The primary difference between the suggested two works and our work lies in the focus: **the former focuses on audiovisual supervised continual learning** where the model is continuously trained on the audiovisual tasks with class labels. Conversely, **our work focuses on audiovisual unsupervised continual learning** where the model is continuously pre-trained with self-supervised loss terms and the pre-trained model is estimated on various downstream tasks.
>
> For a more comprehensive grasp of these distinctions, we have included Figure 7 and *Appendix B, Comparison between SCL setup and UCL setup* in our latest revision. This section delineates and contrasts supervised continual learning with unsupervised continual learning, facilitating a clearer understanding of our approach.
>
> [1] Mo, Shentong, Weiguo Pian, and Yapeng Tian. "Class-incremental grouping network for continual audio-visual learning." Proceedings of the IEEE/CVF International Conference on Computer Vision. 2023.
>
> [2] Pian, Weiguo, et al. "Audio-visual class-incremental learning." Proceedings of the IEEE/CVF International Conference on Computer Vision. 2023.
>
> ---
>
> > **Weakness 3-2: The authors can discuss the differences between the proposed method and the second work. Although it is clear that the works are concurrent, more discussions would be helpful to distinguish between the different works.**
>
> $\rightarrow$ The differences between the AV-CIL (second work) and our work are also clear as follows:
>
> - **In our setting, the absence of class labels allows for a direct assessment of audio-video correlation, enabling a visual observation of the extent of forgetting.** The AV-CIL observation regarding the forgetting of audio-visual correlation is rooted in a supervised learning setting. Consequently, we cannot disregard the influence of class labels that guide visual attention, similar to traditional supervised continual learning.
>
> - The AV-CIL observes the phenomenon where visual attention maps from previous tasks could vanish during continual training. In our work, we too have observed a parallel phenomenon, evident in both Figure 2 and Figure 16. **Importantly, our focus extends to investigating the impact of previous tasks on the current task attention map activations as illustrated in Figure 3 and Figure 17.**
>
> - Motivated by the observation in Figure 3, we suggest an approach aimed at selecting patches that are informative and forget-robust. The AV-CIL focuses on distilling past knowledge into the current model using the past model.
>
> - The AV-CIL relies on the assumption that the model possesses task boundary information throughout its training process, which is required to separate the past model and current model. Conversely, our approach operates within a task-free setting (*Section 3.1, paragraph 1, line 4*) where the model does not know the task boundary information. **The task-free setting is a notably more challenging scenario**, requiring the model to adapt dynamically without relying on explicit task boundaries.
>
> - The AV-CIL framework overlooks the integration of a video-guided audio attention map, a crucial element in comprehending audiovisual tasks. Conversely, **our approach places equal emphasis on harnessing the potential of the video-guided audio attention map.**
>
> ---

---

> ### Author Response · Authors · 2023-11-17
>
> > **Weakness 4: On the audiovisual classification task, why are the improvements of the proposed method marginal?**
>
> $\rightarrow$ **Our method primarily focuses on an effective approach to retaining and learning audio-video correlation.** Hence, our method is especially effective in tasks that require high-level audio-video alignment (i.e., audiovisual retrieval task). On the other hand, the classification task requires discriminative features to classify a given video into its label, rather than focusing on correlations of audio-video multimodality, so the improvement is not dramatic. On the other hand, we achieved the highest average accuracy on the classification task for both datasets, which is statistically significant when considering the standard deviation.
>
> During the rebuttal period, we also suggest FLAVA+, an extension of FLAVA, where its memory stores the selected patches. We update the details of FLAVA+ (Section 4.3) and experiment results of FLAVA+ (Table 1, 4, 7) in our latest revision. We also summarize the classification performance of FLAVA+ in the below Table. **FLAVA+ significantly outperforms baseline methods in large margins.** The introduction of FLAVA+ represents an orthogonal direction of our initial contribution. It represents our method’s potential to efficiently utilize rehearsal memory.
>
> Audiovisual classification tasks
>
> | Method   | VGGSound acc | VGGSound fgt | AudioSet acc | AudioSet fgt |
> |---------------|------------|-----------|------------|-----------|
> | FLAVA | 58.65 | **0.64** | 65.66 | 0.89 |
> | FLAVA+ | **59.07** | 0.80 | **66.05** | **0.78** |
>
> We also performed additional audiovisual downstream tasks (audiovisual event localization, sound source localization) during the rebuttal period, and found that the proposed method outperformed the other baselines (*Appendix E, Audiovisual Event Localization and Sound Source Localization*), supporting the superiority of the proposed method. The Table below shows the audiovisual event localization result.
>
> Audiovisual event localization task
>
> | Method | Acc |
> |---------------|------------|
> | Finetune | 52.56 |
> | ER | 54.98 |
> | DER++ | 55.81 |
> | GMED | 55.98 |
> | LUMP | 55.06 |
> | **FLAVA (Ours)** | **56.68** |
> | **FLAVA+ (Ours)** | **56.68** |
>
> ---

---

> ### Author Response · Authors · 2023-11-21
> **Dear Reviewer hYwV - A Gentle Reminder**
>
> Dear Reviewer hYwV,
>
> We are sincerely grateful to you for reading our response.
>
> > During the rebuttal period,
> - We have clarified the meaning of “relative importance of current data and past data”, and that we have randomly sampled past data from the rehearsal memory to compute the past attention maps.
> - We have additionally suggested **an efficient approach to improving memory usage by storing the delicately selected patches through our method in the rehearsal memory, rather than saving the raw data.** Through extensive experiments, we have shown that the proposed approach outperforms baselines, allowing the memory to effectively maintain audio-video knowledge from past tasks, simultaneously enhancing data diversity in current task training.
> - We have clarified how our method resolves the problems raised in Figures 1,2 and 3.
> - We have included the experiment results on **shuffled task order**, as you requested, and our method shows competitive or better performance compared to other baselines. We believe this result indicates that **our method is robust under varying conditions.**
> - We have clarified that **our work focuses on audiovisual unsupervised continual learning while the two focus on supervised continual learning with audiovisual data.** For a more comprehensive grasp of the difference, we have included Figure 7 and *Appendix B, Comparison between SCL setup and UCL setup* that delineates and contrasts supervised continual learning with unsupervised continual learning.
> - We have discussed that the audiovisual classification task focuses on classifying a given video into its label using discriminative features, while **our method is particularly effective in tasks requiring high-level audio-video alignment.**, resulting in not dramatic improvement of our proposed method on the classification task compared to the retrieval task.
>
> We remain committed to further improving the quality of our paper by addressing any remaining concerns and suggestions where necessary. With that in mind, If you might have any further feedback, please let us know. We would be grateful for the opportunity to address them and make our work a more solid and valuable contribution to the field of audio-video continual representation learning.
>
> **Also, we would like to kindly suggest your reconsideration of the rating, if you feel that our work does not have major concerns with respect to evaluation, resources, reproducibility, and ethical considerations.** We understand that the criteria for rating a paper can sometimes be subjective; however, we believe that most of your concerns are effectively addressed as long as there are no major issues.
>
> We thank you so much for your time and effort in reviewing our paper, and your constructive feedback that has greatly contributed to improving our paper.
>
> Warm Regards,
> Authors

---

> ### Author Response · Authors · 2023-11-22
> **Thank you for your review; Today is the end of the discussion phase.**
>
> Dear Reviewer hYwV,
>
> We sincerely appreciate your efforts in reviewing our paper, and your constructive comments. We have responded to your comments, faithfully reflected them in the revision, and provided additional experimental results that you have requested.
>
> As you know, now we have only one day to have interactive discussions. Could you please go over our responses and the revision since the end of the final discussion phase is approaching? Please let us know if there is anything else we need to clarify or provide.
>
> Best,
> Authors
>
> ---

---

> ### Author Response · Authors · 2023-11-23
> **Discussion phase ends within 8 hours**
>
> Dear Reviewer hYwV,
>
> We really appreciate your effort in reviewing our submission again. Since the discussion period for ICLR 2024 ends within 8 hours, we politely ask you to read our new responses by any chance. Please understand that we have made our best effort to address your concerns during this period.
>
> Also, we would like to kindly suggest **increasing the initial rating**, if you agree that **most concerns raised by the initial review are resolved**. We strongly believe that most of your concerns are effectively addressed as long as there are **no significant issues** now.
>
> We thank you so much for your time and effort in reviewing our paper and for the constructive feedback that has greatly improved it.
>
> Best,
> Authors

---

### Official Review · Reviewer_Q9gf · 2023-11-09

**Soundness:** 3 good
**Presentation:** 2 fair
**Contribution:** 3 good
**Rating:** 5
**Confidence:** 3

**Summary:**

The motivation of this paper is clear and meaningful. The authors bring up two critical challenges for continuous audio-visual learning: (1) sparse spatiotemporal correlation between the video-audio pairs, and 2) representational forgetting of audio-video relationships. To demonstrate these two challenges, the authors give a visualization of cross-attention maps, which can illustrate that the traditional model will forget the correct relation between these two modalities. The authors also propose a novel model named Forget-robust Localized Audio-Video Alignments to alleviate these two challenges.

**Strengths:**

The motivation of this paper is clear and meaningful.

**Weaknesses:**

The paper claims that they can achieve better audio-visual lifelong alignment. However, the authors choose retrieval and classification as their downstream tasks, which can not effectively demonstrate the superior of the propose model. Retrieval and classification tasks only require global connection between audio and visual features, to further illustrate the effectiveness of the model, the authors should choose more convincing tasks, such as audio-visual event localization, audio-visual parsing, audio-visual segmentations, all these tasks have corresponding datasets. It will be better if the proposed model can achieve promising results on these downstream tasks.
The ablation studies in experiments are nor sufficient. The authors should analyze more about each components in the propose method.

**Questions:**

In figure 3, the authors claim that they use similar audio in (c). What are the criteria for selecting similar audio, and how to ensure that there will be no other interfering noise between the selected audio and the original audio.

---

> ### Author Response · Authors · 2023-11-17
>
> Thank you for your review and your constructive comments. During the rebuttal period, we have made every effort to address your concerns. The detailed responses are below:
>
> ---
>
> > **Weakness 1-1: The paper claims they can achieve better audio-visual lifelong alignment. To further illustrate the effectiveness of the model, the authors should choose more convincing tasks, such as audio-visual event localization, audio-visual parsing, audio-visual segmentations. It will be better if the proposed model can achieve promising results on these downstream tasks.**
>
> $\rightarrow$ Thank you for your meaningful comments about our experiments. We agree with your opinion and thus provide the result of the audiovisual event localization task with the AVE [1] dataset in the Table below. It allows us to assess the model’s adaptability to precisely aligning audio-visual information within the unseen video. The analysis of the experiment result is summarized in *Appendix E, Audiovisual Event Localization*.
>
> The result demonstrates that **our method surpasses other baseline methods in the audiovisual event localization task**. This underscores the strength of our method in adapting the downstream task that necessitates a sophisticated grasp of audio-video alignment at a high level.
>
> | Method | Acc |
> |---------------|------------|
> | Finetune | 52.56 |
> | ER | 54.98 |
> | DER++ | 55.81 |
> | GMED | 55.98 |
> | LUMP | 55.06 |
> | **FLAVA (Ours)** | **56.68** |
> | **FLAVA+ (Ours)** | **56.68** |
>
> In addition, we conduct qualitative analysis on the audiovisual alignment through the sound source localization task with the AVE dataset, which is illustrated in *Appendix E, Sound source localization*. The results show the effectiveness of the suggested AVM module that potentially opens up new possibilities for adapting continually pre-trained models to diverse audiovisual tasks like audiovisual segmentation.
>
> We will conduct additional in-depth experiments on the proposed audiovisual tasks and update the Appendix in the final revision.
>
> [1] Tian et al., Audio-Visual Event Localization in Unconstrained Videos, ECCV 2022
>
> ---
>
> > **Weakness 1-2: The ablation studies in experiments are not sufficient. The authors should analyze more about each component in the proposed method.**
>
> $\rightarrow$ While we analyzed the effect of each component in Table 3 and in *Appendix F*, we agree that a more rigorous analysis would be helpful. Hence, we have additionally estimated the modality gap of each component of our proposed method: LAVA (Localized audio video alignment in *Section 4.1*) and FRS (Forget-robust selection in *Section 4.2*) in Figure 15 in our latest revision.
>
> Here, we observe that the LAVA consistently exhibits the highest modality gap across the tasks (Figure 15 (a)), meaning that the LAVA achieves better audio and video clustering within the multi-modal representation space. **This strongly supports our claim that the method in *Section 4.1* adeptly selects informative multi-modal patches from raw data.**
>
> In the case of the FRS, it shows a comparably small modality gap difference (Figure 15 (b)). This implies that during the continual pre-training, the modality gap is robust to the effect of changing data distribution. **This also supports our claim that the method introduced in *Section 4.2* proficiently selects forget-robust patches.**
>
> We have described a detailed explanation of the analysis in *Appendix G, paragraphs 4-5* in the latest revision. We strongly believe this additional in-depth analysis further strengthens our paper.
>
> ---
>
> > **Question 1: In Figure 3, the authors claim that they use similar audio in (c). What are the criteria for selecting similar audio, and how to ensure that there will be no other interfering noise between the selected audio and the original audio.**
>
> $\rightarrow$ **We randomly sample past audio-video pairs from the rehearsal memory, which stores the subset of data from the past tasks, during the continual pre-training phase, following the reservoir sampling approach in our baselines** (e.g., LUMP, DER++, GMED, CLS-ER, ...). We sincerely apologize for the confusion that arose due to our oversight. To rectify this, we have made updates to *Section 5.1, Baselines, line 4* of our latest revision.
>
> We follow the same strategy to visualize Figure 3 Left; the current task is 'music' while the past task is 'sports' in our VGGSound experiment (*Appendix B, Audiovisual Dataset Configuration, paragraph 1, line 12*). Hence, the figure demonstrates the activation of regions within the present data that exhibit correlations with the randomly sampled past data.
>
> If you add some more details on the issue of  "interfering noise between the select audio and the original audio", we will do our best to solve your question.
>
> ---

---

> ### Author Response · Authors · 2023-11-21
> **Dear Reviewer Q9gf - A Gentle Reminder**
>
> Dear Reviewer Q9gf,
>
> We are sincerely grateful to you for reading our response.
>
> > During the rebuttal period,
> - We have experimented on **the audiovisual event localization task** and found that our method surpasses other baseline methods. We have also visualized **the sound source localization task** results and found that our suggested small trainable module can effectively catch potential sound sources. These results underscore **the strength of our method in adapting the downstream task that necessitates a sophisticated grasp of audio-video alignment at a high level.**
> - We have additionally conducted **more analysis of each component of our method through the modality gap.** We have observed that our first component ensures better audio and video clustering within the multi-modal representation space, while our second component shows robustness to the changing data distribution. We believe that **these results strongly support our claim that our method adeptly selects informative multi-modal patches from raw data and proficiently selects forget-robust patches.**
> - We have clarified that we randomly sample past audio-video pairs from the rehearsal memory, following the reservoir sampling approaches that our baselines employed.
>
> We remain committed to further improving the quality of our paper by addressing any remaining concerns and suggestions where necessary. With that in mind, If you might have any further feedback, please let us know. We would be grateful for the opportunity to address them and make our work a more solid and valuable contribution to the field of audio-video continual representation learning.
>
> **Also, we would like to kindly suggest your reconsideration of the rating, if you feel that our work does not have major concerns with respect to evaluation, resources, reproducibility, and ethical considerations.** We understand that the criteria for rating a paper can sometimes be subjective; however, we believe that most of your concerns are effectively addressed as long as there are no major issues.
>
> We thank you so much for your time and effort in reviewing our paper, and your constructive feedback that has greatly contributed to improving our paper.
>
> Warm Regards,
> Authors

---

> ### Author Response · Authors · 2023-11-22
> **Thank you for your review; Today is the end of the discussion phase.**
>
> Dear Reviewer Q9gf,
>
> We sincerely appreciate your efforts in reviewing our paper, and your constructive comments. We have responded to your comments, faithfully reflected them in the revision, and provided additional experimental results that you have requested.
>
> As you know, now we have only one day to have interactive discussions. Could you please go over our responses and the revision since the end of the final discussion phase is approaching? Please let us know if there is anything else we need to clarify or provide.
>
> Best,
> Authors
>
> ---

---

> ### Author Response · Authors · 2023-11-23
> **Discussion phase ends within 8 hours**
>
> Dear Reviewer Q9gf,
>
> We really appreciate your effort in reviewing our submission again. Since the discussion period for ICLR 2024 ends within 8 hours, we politely ask you to read our new responses by any chance. Please understand that we have made our best effort to address your concerns during this period.
>
> Also, we would like to kindly suggest **a reconsideration of the initial rating (reject: 5)**, if you agree that **most concerns raised by the initial review are resolved**. We strongly believe that most of your concerns are effectively addressed as long as there are **no significant issues** now.
>
> We thank you so much for your time and effort in reviewing our paper and for the constructive feedback that has greatly improved it.
>
> Best,
> Authors

---

### Official Review · Reviewer_nu6v · 2023-11-09

**Soundness:** 2 fair
**Presentation:** 1 poor
**Contribution:** 2 fair
**Rating:** 5
**Confidence:** 3

**Summary:**

This work proposes audio-visual continual learning with self-supervised learning, building off of CAV-MAE. Compared with the recent AV-CIL work that proposes supervised audio-visual continual learning, this method doesn't require labels during the pre-training stage. The authors show that the standard audio-visual model is prone to "forgetting" when fine-tuned on a new task, and their approach mitigates this problem.

Overall, I think this is interesting work, but I have concerns about the motivation and experiments.

**Strengths:**

- This work extends CAV-MAE with an AVM (audio-visual matching module) which shows good qualitative cross-modal localization abilities (ie. the ability to localize visual sound sources). CAV-MAE could not achieve this capability out-of-the-box (see CAV-MAE Appendix I which shows poor sound source localization results).
- Implementation details and analysis of the model are provided.

**Weaknesses:**

- I don't understand the motivation of audio-visual continual learning. I think large scale audio-video pre-training data is enough to learn generic representations for different categories of sounds. It seems like an unrealistic constraint to train the model on one category of video at a time (ie. music, sports, etc...), when we could just pool together data from all of the categories and learn a more general representation from the start (especially because the model doesn't require class labels and it's easy to get unlabeled videos). Moreover, the performance of the proposed method is worse than the "multi-task" result where the model simultaneously trains on the data from all of the different tasks. Also, the retrieval performance is much worse than reported in CAV-MAE even with a smaller / easier retrieval set size used in this work, although the present work does not explain why.
- The proposed method assumes access to audio-visual data (ie. memory) from the previous tasks, which doesn't seem like a realistic scenario. I think a more realistic scenario is to have a black-box model without access to the pre-training data, and then the model is presented with a new task / training data. Otherwise, if we have access to the data from the previous tasks, why not just combine the data from the new task and the old tasks to train the model? I didn't see a comparison with this kind of approach. Besides, in Figure 9, the proposed method improves with a larger memory size (ie. more access to data from previous tasks), which further shows the benefit of training on combined data from different tasks.
- As someone familiar with audio-visual learning but not continual learning, I did not find the explanation of the other methods in the main results adequate. The difference with the compared methods should be explained more.
- The writing / explanation of the method is not clear enough. I didn't understand Sections 4.2 and Sections 4.3; the writing should be improved with more high-level explanation. It would be helpful if each line in equation (4) was explained separately. Algorithm 1 was helpful for a high-level understanding, perhaps more detail could be added there.
- The proposed method is only significantly better than the baseline methods for continual learning (ie. LUMP) on the VGGSound retrieval task. For the AudioSet retrieval task and VGGSound / AudioSet classification tasks, the improvement is small.
- The experiments leave questions unanswered (see my questions below).
- There are some distracting typos ("vidoe," "Fintune")

**Questions:**

- The pre-training / fine-tuning / task / evaluation splits of the datasets should be more clear. Can you provide a table in the appendix with the precise number of clips for each split? Specifically, I am wondering how the training and evaluation data differs with CAV-MAE.
- Why are the tasks set up to be "zero-shot" by excluding classes from all of the continual learning datasets? I don't understand how this measures "forgetting" since the model isn't being tested on classes that it was trained on. It would make more sense to me to have an evaluation set per continual learning subset and test the model on evaluation sets corresponding to tasks it has already seen (and average the results).
- What is the difference between your method and the baseline methods in terms of the design?
- Is it possible to compare with AV-CIL on the classification task, since that requires supervised fine-tuning?
- Why is the Multitask retrieval result worse than CAV-MAE? The classification results on AudioSet and VGGSound are much higher than in CAV-MAE, is it due to a different evaluation set?
- Why is the average forgetting much smaller for the classification task compared to the retrieval?
- It would be nice to see a table breaking down the performance of each task at each stage in the continual learning process, similar to Figure 10b but for all tasks (ie. stage on the rows, task on the columns, and the upper diagonal should be filled). I'd like to see this for the fine-tuning method and the proposed method to understand how soon the simple fine-tuning strategy deteriorates.
- How does the order of the tasks change the performance? Have you tried other orders?

Misc. questions
- For the AVM objective, the binary cross entropy loss requires less memory than the contrastive loss. Does the batch size and number of positive / negative pairs impact the performance? Have you tried training with the contrastive loss?
- What is "ER" in Figure 10?
- How is modality gap estimated?
- What are the model sizes and number of parameters?

---

> ### Author Response · Authors · 2023-11-17
>
> Thank you for your review and your constructive comments. During the rebuttal period, we have made every effort to address your concerns. The detailed responses are below:
>
> ---
>
> > **Weakness 1-1: I don't understand the motivation of audio-visual CL, as it seems like an unrealistic constraint to train the model on one category of video at a time, when we could just pool together data from all of the categories and learn a more general representation from the start.**
>
> $\rightarrow$ We want to politely emphasize that **continual self-supervised learning with audio-visual multimodality** is an **undoubtedly crucial scenario for the real world**. As other reviewers totally agreed, *“lifelong learning in audio-visual scenes is a very meaningful research topic”* **(Reviewer 4AjC)** and especially *“self-supervised audio-visual continual learning is an important topic in multimodal learning”* **(Reviewer hYwV)**, solving the above problem gives strong motivation to effectively pre-train the audio-video model continuously.
>
> This is because **in the real world, new categories of audio-video semantics constantly emerge** as the world is dynamic. For example, people very recently recognized the unique motor sound of electric vehicles after they became commercialized. We should continuously update the model with new knowledge (described in *the second paragraph, third line in the Introduction*).
>
> The easiest way for a model to learn new knowledge is to train it from scratch with new and old data. However, new **concepts/data/semantic categories emerge every day**, and training (from scratch) with new and old data requires **immense computational costs** with **financial waste** due to redundant knowledge learning. Moreover, permanently storing all past audio-video data for future retraining can lead to **serious privacy issues** and **memory constraints**. This becomes even more critical for large-scale audio-video pre-training, which consumes industry-level computing power and web-scale video data to pre-train the model.
>
> This is why, in other words, **audio-video continual self-supervised learning is important and promising for the real world**.
>
> [1] Konishi et al., Parameter-Level Soft-Masking for Continual Learning, ICML 2023
> [2] Wang et al., Hierarchical Decomposition of Prompt-Based Continual Learning: Rethinking Obscured Sub-optimality, NuerIPS 2023
> [3] Khattak et al., Self-regulating Prompts: Foundational Model Adaptation without Forgetting, ICCV 2023
>
> ---
>
> > **Weakness 1-2: The performance of the proposed method is worse than the "multi-task" result where the model simultaneously trains on the data from all different tasks.**
>
> $\rightarrow$ We provide the “Multi-task” baseline as **an upper bound performance**, assuming that a model has the privilege to know and access **all future tasks in advance**, which is unrealistic in the real world.
>
> ---
>
> > **Weakness 1-3 & Question 5: The retrieval performance is much worse than reported in CAV-MAE, even with a smaller / easier retrieval set size used in this work, although the present work does not explain why.**
>
> $\rightarrow$ We believe you may confuse our problem statements. The setting of CAV-MAE is for conventional self-supervised learning on static datasets. That is, **the setting is totally different from ours** and a direct comparison with the number in the CAV-MAE paper isn't meaningful. For clarification, we summarize the differences between the setting of ours and CAV-MAE as follows:
>
> - We use the subset of the VGGSound and AudioSet datasets and split them into multiple tasks. Then, we continually train the model with the sequence of the split datasets, while the CAV-MAE trains the model on the complete AudioSet-2M dataset.
>
> - The retrieval evaluation data size is identical in the VGGSound experiment and smaller in the AudioSet to exclude retrieval evaluation samples that overlap in two different tasks.
>
> - For evaluation, we record the task-specific performance at each end of the task. Then, we estimate the difference between the performances from the last task and the task-specific performances to quantify the amount of representational forgetting of the model. We also average the performance from the last task to estimate the average accuracy (*Section 5.1, Evaluation Metrics*). Conversely, the CAV-MAE estimates the performance of the model pre-trained with the complete dataset.
>
> ---

---

> ### Author Response · Authors · 2023-11-17
>
> > **Weakness 2-1: The proposed method assumes access to audio-visual data (ie. memory) from the previous tasks, which doesn't seem like a realistic scenario.**
> > **Weakness 2-3: If we have access to the data from the previous tasks, why not just combine the data from the new task and the old tasks to train the model? In Figure 9, the proposed method improves with a larger memory size, which further shows the benefit of training on combined data from different tasks.**
>
>
> $\rightarrow$ Storing all videos and training from scratch has huge computational costs, privacy issues, and memory constraints, as discussed in Weakness 1-1. To alleviate these problems, rehearsal-based continual learning, where the model temporally stores a small fraction of previous task data in a small-sized buffer **during training**, has become one of the most effective and popular/well-known directions in CL tasks [1,2,3,4], achieving a good trade-off between performance and efficiency. And we follow the same assumption since this is also valid in our scenario.
>
> An ever-growing dataset that the Reviewer suggested **needs an assumption of having an infinite amount of resources, which is clearly far from the realistic scenario.** The purpose of the experiment in Figure 9 is to show the **trade-off** between rehearsal memory size and downstream task performance (Appendix E, Effect of Rehearsal Memory), not to demonstrate a **one-sided benefit of expanding storage size**.
>
> [1] Jin et al., Gradient-based Editing of Memory Examples for Online Task-free Continual Learning, NeurIPS 2021
> [2] Yoon et al., Online Coreset Selection for Rehearsal-based Continual Learning, ICLR 2022
> [3] Caccia et al., New Insights on Reducing Abrupt Representation Change in Online Continual Learning, ICLR 2022
> [4] Jeeveswaran et al., BiRT: Bio-inspired Replay in Vision Transformers for Continual Learning, ICML 2023
>
> ---
>
> > **Weakness 2-2:  I think a more realistic scenario is to have a black-box model without access to the pre-training data, and then the model is presented with new task/training data.**
>
> $\rightarrow$ The suggested scenario is a simple pretraining-finetuning strategy that has two critical limitations in view of continuity. First, given the pre-trained model, the model cannot transfer the knowledge across different downstream tasks since the model aims to transfer the pre-trained representation to the downstream tasks individually. Second, the user requires individual fine-tuned models according to the downstream tasks, requiring N times larger memory budgets proportional to the number of downstream tasks N.
>
> Due to these reasons and as we explicated in *Weakness 1-1*, **the lifelong learnable & sustainable model for real-world scenarios should adapt to new knowledge continuously.**
>
> ---
>
>
> > **Weakness 3 & Question 3 & Question 10: Detailed explanations of continual learning baselines and what is the difference between your method and the baseline methods in terms of the design?**
>
> $\rightarrow$ Thank you for your suggestion. We agreed that providing further details on our baselines is helpful, and we have included explanations of all baselines in *Appendix A, Baselines* in our revision. Here, we focus on describing the differences between two main baselines:
>
> DER++ [1] matches stored logits from past tasks with the current ones, ensuring a smoother transition and preventing knowledge shifts in the logits during training. CLS-ER [2] maintains two identical models to retain short-term memories and long-term memories in each model, and the model with long-term memories transfers retained knowledge to the other to alleviate forgetting of past knowledge.
>
> The originality of our method is that **we select patches that are audio-visually aligned and forget-robust effectively, allowing efficient continual pre-training with these selected important patches only**. Specifically, we exploit instance-wise cross-attention maps to select informative audio-video patches and inter-task (i.e., between current and past tasks) cross-attention maps to select patches that are robust to representational forgetting. Comparably, other baselines pre-train the model using the unprocessed data and compute to update all patches without considering the spatiotemporal correlation between audio and visual modality, resulting in suboptimal retrieval performance with higher forgetting.
>
>
> [1] Buzzega et al., Dark experience for general continual learning: a strong, simple baseline, NeurIPS 2020
> [2] Arani et al., Learning fast, learning slow: A general continual learning method based on complementary learning system, ICLR 2022
>
> ---

---

> ### Author Response · Authors · 2023-11-17
>
> > **Weakness 4-1: It would be helpful if each line in equation (4) was explained separately. Algorithm 1 was helpful for a high-level understanding, perhaps more detail could be added there.**
>
> $\rightarrow$ Thank you for your suggestion! We have added a line indicator of Equation 4 to corresponding sentences to help you understand the algorithm of forget-robust patch selection in Section 4.2. **We have also added more explanations in Section 4.2, Section 4.3, Figure 4, and Algorithm 1 to help you better understand the overall process.** In the below, we follow Section 4.2 and Section 4.3 step-by-step to provide more details. Please let us know if there's anything we can do to make the description clearer! We'll be happy to incorporate your suggestions.
>
> ---
>
> > **Weakness 4-2: The writing/explanation of the method is not clear enough. I didn't understand Sections 4.2 and 4.3.**
>
> $\rightarrow$ Thank you for your opinion. We have made efforts to more clearly describe our approach in sections 4.2 and 4.3, as well as Figure 4, and update them in our latest revision.
>
> Section 4.2 explains **the process of obtaining the current and past attention map information reflecting inter-task (past & current) importance** (*Section 4.2, paragraph 1, line 16*). Using this importance, the forget-robust patch selection aims to prune patches from current data that have a higher correlation with previous steps. The purpose of this pruning is to retain the past correlation knowledge while effectively learning new data (*Section 3.2, line 11~16*).
>
> In Section 4.3, **we utilize the importance scores and pruning probability matrices to find the forget-robust and informative audio and video patches** (*Section 4.3, paragraph 1, line 1* and *Section 4.3, paragraph 3, line 1*). In the case of the audio, we have to preserve local correlation among audio patches due to the temporal continuity in the spectrogram (*Section 4.3, paragraph 1, line 2*). Hence, we follow the process below.
>
> 1. Process the audio importance score ($\textbf{I}\_{a}$) and audio pruning indicator ($\textbf{F}\_{a}$) in time-wise to obtain $\textbf{I}^{t}\_{a}$ and $\textbf{F}^{t}\_{a}$. (Section 4.3, paragraph 1).
>
> 2. Group $\textbf{I}^{t}\_{a}$ in time chunks with length $\textbf{L}\_c$ to preserve the time continuity information. This grouping process generates a chunk-wise importance score ($\textbf{I}^{c}\_{a}$). (Section 4.3, paragraph 2, line 1~4)
>
> 3. Based on $\textbf{I}^{c}\_{a}$, we employ multinomial probability distribution to sample the time chunks that are most likely to be informative. (Section 4.3, paragraph 2, line 5)
>
> 4. Iterate through the sampled time chunks and accumulate the number of selected audio patches while excluding the forget-inducing audio patches referring to $\textbf{F}^{t}\_{a}$.
>
> 5. We iterate until the number of selected audio patches reaches our target number, which is $\kappa\_{a}$. (Section 4.3, paragraph 2, line 6-7)
>
> For video patches are selected in a much simpler manner.
>
> 1. For every element in the video importance score ($\textbf{I}\_{v}$) that aligns with indices marked as True in the video pruning indicator ($\textbf{F}\_{v}$), we set the value to zero. This process is to exclude the forget-inducing video patches in our video patch selection, yielding $\tilde{\textbf{I}\}_{v}$. (Section 4.3, paragraph 3, line 4)
>
> 2. Apply a multinomial probability distribution on $\tilde{\textbf{I}}\_{v}$ to select video patches that are most likely to be informative. (Section 4.3, paragraph 3, line 6)
>
> ---

---

> ### Author Response · Authors · 2023-11-17
>
> > **Question 1-1: The pre-training / fine-tuning / task / evaluation splits of the datasets should be more clear. Can you provide a table in the appendix with the precise number of clips for each split?**
>
> $\rightarrow$ As the reviewer suggested, **we additionally provide the additional table for statistics of the dataset in *Figure 6* in our latest revision.** We use a subset of the VGGSound and the AudioSet dataset and split it according to their high-level category information. The finetuning data is identical to the pre-training data, as commonly practiced in other self-supervised learning research [1, 2] when evaluating the learned representation from the pre-training data. In the case of the zero-shot retrieval task, we use a subset of the evaluation retrieval dataset from the CAV-MAE [1].
>
> [1] Gong et al., Contrastive Audio-Visual Masked Autoencoder, ICLR 2023
>
> [2] He et al., Masked Autoencoders Scalable Vision Leaners, CVPR 2022
>
> ---
>
> > **Question 1-2: I am wondering how the training and evaluation data differs with CAV-MAE.**
>
> $\rightarrow$ **We use the subset of the VGGSound and AudioSet for continual pre-training according to their high-level category information** (*Appendix B, Audiovisual Dataset Configuration*). We split the VGGSound dataset into 8 tasks based on the category labels (sports, music, ...) and each task consists of 6k-8k video clips from 20 different classes. The AudioSet follows a similar process. We follow the official class hierarchy information to split it into 7 tasks. Since the AudioSet is a multi-label dataset, we exclude the samples that are included in more than one task in order to prevent the past task semantics from leaking into the current task. **In contrast, the CAV-MAE uses the whole VGGSound and AudioSet dataset**.
>
> ---
>
> > **Question 2: Why are the tasks set up to be "zero-shot" by excluding classes from all of the continual learning datasets? I don't understand how this measures "forgetting" since the model isn't being tested on classes that it was trained on.**
>
> $\rightarrow$ We believe this misunderstanding is from the confusion of the scenario of continual self-supervised learning. For a clearer understanding, in Figure 7 in our latest revision, we have illustrated the supervised and the unsupervised continual learning setups in both the training and evaluation setup. We also elaborate on it in *Appendix B, Comparison between two different setups* in our latest revision.
>
> We evaluate continual self-supervised learning methods on audio-video multimodality using zero-shot retrieval tasks to assess the continually pre-trained model’s ability to align audio and video into the multi-modal representation space and quantify the amount of forgetting. In the zero-shot retrieval task, we do not need class labels, but we ask the continually pre-trained model to find the correct audio(or video) pair given video(or audio) at each end of the pre-train task.
>
> To assess the task-specific retrieval task performance, we split the retrieval evaluation data according to the category information, as we split the training and evaluation dataset. In this way, **we test the model on the whole retrieval evaluation data and extract the performance corresponding to tasks it has already seen.**
>
> We compare each maximum of task-specific ($i$) performance of the pre-trained models ($\underset{t\in\{1,\ldots, \mathcal{T}-1\}}{\text{max}}\textit{acc}\_{t,i}$) with the task-specific performance at the completion ($\mathcal{T}$) of the tasks ($\textit{acc}\_{\mathcal{T},i}$) (*Section 5.1, Evaluation Metrics*, Figure 7). In this way, we can estimate how the model's representation ability has changed during the continual pre-training, which is quantified as *forgetting*. This process is also applied to the audio-video classification task.
>
> ---
>
> > **Question 4: Is it possible to compare with AV-CIL on the classification task, since that requires supervised fine-tuning?**
>
> $\rightarrow$ The AV-CIL follows (1) supervised continual learning, (2) trains its own backbone structure, (3) requires class labels to implement task-wise knowledge distillation losses, and (4) assumes class-incremental learning, which leverages task boundary information during training, while our setting follows (1) unsupervised continual learning, (2) uses CAV-MAE backbone and freeze the backbone in downstream tasks, (3) pre-trained on contrastive loss & MAE objectives without class labels, and (4) assumes task-free learning. Hence, the AV-CIL itself is not applicable in our setting, and direct comparison is out of our interest.
>
> For a more comprehensive grasp of these distinctions, we have included Figure 7 and *Appendix B, Comparison between SCL setup and UCL setup* in our latest revision. This section delineates and contrasts supervised continual learning with unsupervised continual learning, facilitating a clearer understanding of our approach.
>
> ---

---

> ### Author Response · Authors · 2023-11-17
>
> > **Question 6: Why is the average forgetting much smaller for the classification task compared to the retrieval?**
>
> $\rightarrow$ This is because **classification tasks have a chance to rectify/update the classifier to the target tasks, while retrieval tasks do not.**
>
> The classification tasks require an additional cost of fine-tuning process to estimate the learned audio-video joint representation. Through fine-tuning with the labels, the model can adapt to the target tasks. On the other hand, the zero-shot retrieval task offers scalability without the need for labels or extra training costs, yet it is more susceptible to representational forgetting. Consequently, simultaneously acquiring new knowledge while effectively mitigating forgetting is a more challenging problem in the zero-shot retrieval task.
>
> ---
>
> > **Question 7: It would be nice to see a table breaking down the performance of each task at each stage in the continual learning process, similar to Figure 10b but for all tasks (ie. stage on the rows, task on the columns, and the upper diagonal should be filled).**
>
> $\rightarrow$ The Table below presents the VGGSound classification accuracy matrix, where the row indicates the pre-train task identity and the column indicates the task name for the task-specific performance estimation.
>
> | Task name \ Pre-train stage   | Sports | Music | Vehicle | People | Animals | Home&Nature | Others_part1 | Others_part2 |
> |---------------|------------|-----------|-----------|-----------|-----------|-----------|-----------|-----------|
> | Sports | 75.81 | 75.36 | 74.42 | 74.12 | 73.78 | 73.48 | 72.91 | 72.43 |
> | Music | - | 56.34 | 54.91 | 53.73 | 53.03 | 52.37 | 52.63 | 52.52 |
> | Vehicle | - | - | 56.73 | 56.07 | 55.96 | 55.15 | 56.55 | 55.48 |
> | People | - | - | - | 52.28 | 51.78 | 50.15 | 49.54 | 49.81 |
> | Animals | - | - | - | - | 55.82 | 54.33 | 55.52 | 53.72 |
> | Home & Nature | - | - | - | - | - | 46.95 | 46.83 | 45.32 |
> | Others_part1 | - | - | - | - | - | - | 65.66 | 66.78 |
> | Others_part2 | - | - | - | - | - | - | - | 60.04 |
>
> We also provide the AudioSet classification accuracy matrix in the Table below.
>
> | Task name \ Pre-train stage   | Human | Vehicle | Nature | Animal | Others | Home | Music |
> |---------------|------------|-----------|-----------|-----------|-----------|-----------|-----------|
> | Human | 52.07  | 51.21 | 49.77 | 51.27 | 51.41 | 50.60 | 49.44 |
> | Vehicle | - | 70.14 | 67.65 | 67.53 | 67.19 | 67.04 | 66.57 |
> | Nature | - | - | 78.51 | 79.07 | 77.32 | 77.64 | 77.34 |
> | Animal | - | - | - | 65.33 | 62.42 | 61.46 | 60.36 |
> | Others | - | - | - | - | 50.92 | 50.69 | 50.63 |
> | Home | - | - | - | - | - | 69.00 | 69.55 |
> | Music | - | - | - | - | - | - | 74.00 |
>
> As can be seen, the Finetune method apparently suffers from catastrophic forgetting of joint audio-video representation.
>
> ---
>
> > **Question 9: For the AVM objective, the binary cross entropy loss requires less memory than the contrastive loss. Does the batch size and number of positive/negative pairs impact the performance? Have you tried training with the contrastive loss?**
>
> $\rightarrow$ We believe our method could be relatively robust to the impact of batch size (i.e., the number of positive/negative pairs) since FLAVA doesn't rely on contrastive loss but training the model along with the masked reconstruction objective. **It is known that the representation learned from the MAE objective has higher transferability compared to the representation learned from the contrastive loss [1].** This relaxed dependency of the batch size is also observed in the CAV-MAE (Table 1, comparison between CAV-MAE and CAV-MAE-scale+) [2].
>
> [1] Xie, et al., Revealing the dark secrets of masked image modeling, CVPR 2023
> [2] Gong et al., Contrastive Audio-Visual Masked Autoencoder, ICLR 2023
>
> ---

---

> ### Author Response · Authors · 2023-11-17
>
> > **Question 8: How does the order of the tasks change the performance? Have you tried other orders?**
>
> $\rightarrow$ Thank you for your meaningful comment about our experiment. We carry out experiments on shuffled task sequences for continual pre-training, and the results are shown in the Table below (*acc* indicates average accuracy, *fgt* indicates average forgetting). We underscore **the best** and ***the second best*** performances in the Table. **Our method shows competitive or better performance compared to other baselines.** We present the task results in Table 7 of our latest revision. This indicates that **our method is robust under varying conditions**, thereby enhancing the credibility of our analysis.
>
> VGGSound Audio-to-Video Result
>
> | Method | R@1 acc | R@1 fgt | R@5 acc | R@5 fgt | R@10 acc | R@5 fgt | Avg acc | Avg fgt |
> |---------------|------------|-----------|-----------|-----------|-----------|-----------|-----------|-----------|
> | Finetune | 0.80 | 4.15 | 2.96 | 12.23 | 5.05 | 16.91 | 2.94 | 11.10 |
> | ER | 3.89 | 3.06 | 12.10 | 6.55 | 18.30 | 7.74 | 11.43 | 5.78 |
> | MIR | 4.02 | 2.97 | 12.54 | 6.16 | 17.99 | 8.09 | 11.52 | 5.74 |
> | DER++ | 4.23 | 3.35 | 12.92 | 7.31 | 18.62 | 9.45 | 11.92 | 6.70 |
> | GMED | 3.90 | 2.94 | 11.51 | 7.41 | 17.65 | 8.87 | 11.02 | 6.41 |
> | CLS-ER | 3.94 | 3.35 | 12.96| 7.19 | 18.09 | 10.66 | 11.66 | 7.07 |
> | LUMP | 4.06 | **2.18** | 13.21 | **4.66** | 19.34 | **5.58** | 12.20 | **4.14** |
> | FLAVA (Ours) | ***4.72*** | ***2.89*** | ***14.17*** | 5.74 | ***19.94*** | ***5.74*** | ***12.94*** | 4.79 |
> | FLAVA+ (Ours) | **4.90** | 3.19 | **16.42** | ***4.72*** | **23.49** | 5.89 | **14.94** | ***4.60*** |
>
>
> VGGSound Video-to-Audio Result
>
> | Method | R@1 acc | R@1 fgt | R@5 acc | R@5 fgt | R@10 acc | R@5 fgt | Avg acc | Avg fgt |
> |---------------|------------|-----------|-----------|-----------|-----------|-----------|-----------|-----------|
> | Finetune | 0.78 | 3.77 | 3.00 | 11.68 | 5.21 | 15.86 | 3.00 | 10.44 |
> | ER | 3.57 | 2.76 | 11.66 | 7.67 | 16.75 | 10.76 | 10.66 | 7.06 |
> | MIR | 3.35 | 3.15 | 11.37 | 7.74 | 16.62 | 10.11 | 10.45 | 7.00 |
> | DER++ | 4.08 | 3.10 | 12.78 | 9.02 | 18.77 | 11.30 | 11.88 | 7.81 |
> | GMED | 3.42 | 3.80 | 11.45 | 7.76 | 17.06 | 9.94 | 10.64 | 7.17 |
> | CLS-ER | 3.49 | 3.85 | 12.28 | 8.05 | 17.75 | 11.31 | 11.17 | 7.74 |
> | LUMP | 3.98 | **1.67** | 12.44 | **5.17** | 18.11 | **7.27** | 11.51 | **4.70** |
> | FLAVA (Ours) | ***4.18*** | 2.54 | ***13.81*** | 6.56 | ***19.90*** | 8.88 | ***12.63*** | 5.99 |
> | FLAVA+ (Ours) | **5.28** | ***1.81*** | **15.35** | ***6.33*** | **21.97** | ***8.01*** | **14.20** | ***5.38*** |
>
> AudioSet Audio-to-Video Result
>
> | Method | R@1 acc | R@1 fgt | R@5 acc | R@5 fgt | R@10 acc | R@5 fgt | Avg acc | Avg fgt |
> |---------------|------------|-----------|-----------|-----------|-----------|-----------|-----------|-----------|
> | Finetune | 1.50 | 4.72 | 5.49 | 10.41 | 9.80 | 11.91 | 5.60 | 9.01 |
> | ER | 4.52 | 3.16 | 12.72 | 6.93 | 18.83 | 8.00 | 12.02 | 6.03 |
> | MIR | 4.69 | ***2.95*** | 13.22 | 6.50 | 18.98 | 8.81 | 12.30 | 6.09 |
> | DER++ | 4.32 | 4.27 | 12.29 | 8.46 | 18.74 | 10.18 | 11.78 | 7.64 |
> | GMED | 4.70 | **2.48** | 12.56 | ***4.55*** | 18.62 | **5.05** | 11.96 | 4.03 |
> | CLS-ER | ***5.16*** | 2.97 | ***14.33*** | 6.88 | 20.24 | 8.74 | 13.24 | 6.20 |
> | LUMP | 4.45 | 3.40 | 13.05 | 6.25 | 19.45 | 7.28 | 12.32 | 5.64 |
> | FLAVA (Ours) | 4.97 | 3.47 | 13.91 | 5.59 | ***20.30*** | ***6.70*** | ***13.06*** | 5.25 |
> | FLAVA+ (Ours) | **5.77** | 3.90 | **17.51** | **4.49** | **23.72** | 7.07 | **15.67** | ***5.15*** |
>
> AudioSet Video-to-Audio Result
>
> | Method | R@1 acc | R@1 fgt | R@5 acc | R@5 fgt | R@10 acc | R@5 fgt | Avg acc | Avg fgt |
> |---------------|------------|-----------|-----------|-----------|-----------|-----------|-----------|-----------|
> | Finetune | 1.42 | 5.11 | 6.54 | 10.30 | 10.43 | 13.48 | 6.13 | 9.63 |
> | ER | 4.01 | 4.31 | 12.47 | 7.27 | 19.32 | 9.26 | 11.93 | 6.95 |
> | MIR | 4.25 | 3.43 | 12.92 | 6.93 | 19.43 | 9.78 | 12.20 | 6.71 |
> | DER++ | 4.31 | 4.35 | 12.60 | 9.59 | 18.93 | 12.27 | 11.95 | 8.74 |
> | GMED | 4.20 | **1.87** | 12.97 | **6.04** | 19.98 | **8.11** | 12.38 | **5.34** |
> | CLS-ER | 4.85 | 5.48 | 13.37 | 9.17 | 19.69 | 11.36 | 12.64 | 8.67 |
> | LUMP | 4.23 | 4.06 | 13.53 | ***6.09*** | 19.27 | 9.53 | 12.34 | 6.56 |
> | FLAVA (Ours) | ***4.86*** | ***2.92*** | ***14.20*** | 6.41 | ***20.00*** | 9.82 | ***13.02*** | ***6.38*** |
> | FLAVA+ (Ours) | **5.57** | 3.80 | **16.67** | 6.96 | **23.91** | 9.28 | **15.38** | 6.68 |
>
> ---

---

> ### Author Response · Authors · 2023-11-17
>
> > **Weakness 5: The proposed method is only significantly better than the baseline methods for continual learning (ie. LUMP) on the VGGSound retrieval task. For the AudioSet retrieval task and VGGSound / AudioSet classification tasks, the improvement is small.**
>
> $\rightarrow$ We respectfully disagree with the Reviewer's comment that the improvement is small. We would like to emphasize that for the AudioSet video-to-audio zero-shot retrieval task, our method **outperforms the strongest baselines at 5.42%**, and our method **ranks the highest R@1 score in all experiments, including the VGGSound and AudioSet**. We also rank the highest average accuracy in the classification tasks for both datasets, which is **statistically significant observing the standard deviation**.
>
> During the rebuttal period, we also suggest FLAVA+, an extension of FLAVA, where its memory stores the selected patches. We update the details of FLAVA+ (Section 4.3) and experiment results of FLAVA+ (Table 1, 4, 7) in our latest revision. We also summarize the performance of FLAVA+ in the below Tables. **FLAVA+ significantly outperforms baseline methods in large margins.** The introduction of FLAVA+ represents an orthogonal direction of our initial contribution. It represents our method’s potential to efficiently utilize rehearsal memory.
>
> VGGSound zero-shot retrieval task
>
> | Method   | A$\rightarrow$V acc | A$\rightarrow$V fgt | V$\rightarrow$A acc | V$\rightarrow$A fgt |
> |---------------|------------|-----------|-----------|-----------|
> | FLAVA | 14.16 | **4.38** | 14.07 | 4.65 |
> | FLAVA+ | **15.44** | 4.62 | **15.53** | **3.89** |
>
> AudioSet zero-shot retrieval task
>
> | Method   | A$\rightarrow$V acc | A$\rightarrow$V fgt | V$\rightarrow$A acc | V$\rightarrow$A fgt |
> |---------------|------------|-----------|-----------|-----------|
> | FLAVA | 12.35 | 6.95 | 11.29 | 8.38 |
> | FLAVA+ | **15.26** | **5.74** | **14.47** | **6.70** |
>
> Audiovisual classification tasks
>
> | Method   | VGGSound acc | VGGSound fgt | AudioSet acc | AudioSet fgt |
> |---------------|------------|-----------|------------|-----------|
> | FLAVA | 58.65 | **0.64** | 65.66 | 0.89 |
> | FLAVA+ | **59.07** | 0.80 | **66.05** | **0.78** |
>
> In addition, we conduct additional audiovisual downstream tasks (audiovisual event localization, sound source localization) during the rebuttal period, found that and our method surpasses other baselines (*Appendix E, Audiovisual Event Localization and Sound Source Localization*), which supports the superiority of our proposed method. The Table below summarizes the audiovisual event localization task results.
>
> Audiovisual event localization task
>
> | Method | Acc |
> |---------------|------------|
> | Finetune | 52.56 |
> | ER | 54.98 |
> | DER++ | 55.81 |
> | GMED | 55.98 |
> | LUMP | 55.06 |
> | **FLAVA (Ours)** | **56.68** |
> | **FLAVA+ (Ours)** | **56.68** |
>
> ---
>
> > **Weakness 7: Typos (“vidoe”, “Fintune”)**
>
> $\rightarrow$ We thank you for finding the typos that would be a disturbance in explaining our findings. We have corrected the typos in the latest revision.
>
> ---
>
> > **Question 11:  How is modality gap estimated?**
>
> $\rightarrow$ We use evaluation data from each task to extract audio and video embeddings from each encoder (*Section 5-2, paragraph 5, line 11*). From the accumulated embeddings, we calculate the normalized L2 distance (i.e., cosine similarity) between the audio embedding cluster and the video embedding cluster. The details of the modality gap are explained in [1], as we referred to in our paper.
>
> [1] Mind the gap: Understanding the modality gap in multi-modal contrastive representation learning, NuerIPS 2022
>
> ---
>
> > **Question 12: What are the model sizes and number of parameters?**
>
> $\rightarrow$ The backbone model size is 707.8MB, with the number of parameters 185M. The AVM module increases only 3.18% of the total backbone size. This is highly efficient compared to methods that need additional backbone models like CLS-ER, which is one of our baseline methods. We have added the model size statistics in *Appendix D, paragraph 3, line 5* in the latest revision.

---

> ### Author Response · Authors · 2023-11-21
> **Dear Reviewer nu6v - A Gentle Reminder**
>
> Dear Reviewer nu6v,
>
> We are sincerely grateful to you for reading our response.
>
> > During the rebuttal period,
> - We have clarified the motivation of continual learning (CL), and especially **why continual audio-video representation learning is important and promising for real-world scenarios.**
> - We have clarified recent CL works to validate that **the rehearsal-based CL is one of the most effective and popular/well-known directions in CL.**
> - We have provided **the major differences between our setting and the setting that follows conventional self-supervised learning** on static datasets, adding Figure 6-(c) and Figure 7 for clarification.
> - We have further clarified the details of our baseline methods and explanation of our proposed method in Section 4. We believe that **these revisions will significantly improve the readability of our work with better understanding.**
> - We have clarified why AV-CIL could not be our baseline, showing major differences. For a more comprehensive grasp of these distinctions, **we have also included Figure 7 and *Appendix B, Comparison between SCL setup and UCL setup* that contrast supervised continual learning with unsupervised continual learning.**
> - We have provided the downstream task results of audiovisual event localization and sound source localization. We have also introduced FLAVA+ which surpasses other baselines by a large margin. We believe these results support the superiority of our method.
> - We have provided the experiment results on **shuffled task order**, as you requested, that our method shows competitive or better performance compared to other baselines.
>
> We remain committed to further improving the quality of our paper by addressing any remaining concerns and suggestions where necessary. With that in mind, If you might have any further feedback, please let us know. We would be grateful for the opportunity to address them and make our work a more solid and valuable contribution to the field of audio-video continual representation learning.
>
> **Also, we would like to kindly suggest your reconsideration of the rating, if you feel that our work does not have major concerns with respect to evaluation, resources, reproducibility, and ethical considerations.** We understand that the criteria for rating a paper can sometimes be subjective; however, we believe that most of your concerns are effectively addressed as long as there are no major issues.
>
> We thank you so much for your time and effort in reviewing our paper, and your constructive feedback that has greatly contributed to improving our paper.
>
> Warm Regards,
> Authors

---

> ### Author Response · Authors · 2023-11-22
> **Thank you for your review; Today is the end of the discussion phase.**
>
> Dear Reviewer nu6v,
>
> We sincerely appreciate your efforts in reviewing our paper, and your constructive comments. We have responded to your comments, faithfully reflected them in the revision, and provided additional experimental results that you have requested.
>
> As you know, now we have only one day to have interactive discussions. Could you please go over our responses and the revision since the end of the final discussion phase is approaching? Please let us know if there is anything else we need to clarify or provide.
>
> Best,
> Authors
>
> ---

---

> ### Author Response · Authors · 2023-11-23
> **Discussion phase ends within 8 hours**
>
> Dear Reviewer nu6v,
>
> We really appreciate your effort in reviewing our submission again. Since the discussion period for ICLR 2024 ends within 8 hours, we politely ask you to read our new responses by any chance. Please understand that we have made our best effort to address your concerns during this period.
>
> Also, we would like to kindly suggest **a reconsideration of the initial rating (reject: 3)**, if you agree that **most concerns/misunderstandings raised by the initial review are resolved**. We strongly believe that most of your concerns are effectively addressed as long as there are **no significant issues** now.
>
> We thank you so much for your time and effort in reviewing our paper and for the constructive feedback that has greatly improved it.
>
> Best,
> Authors

---

### Author Response · Authors · 2023-11-19

To show that our approach enables efficient utilization of the rehearsal memory, we introduce FLAVA+ which implements FLAVA and saves the selected patches in the rehearsal memory.

> **Enhancing rehearsal memory efficiency with FLAVA+**

Efficient utilization of the rehearsal memory is important in rehearsal-based continual learning, particularly in audio-video continual learning scenarios due to the substantially larger size of video data compared to images. The effective storage of past data can notably augment the diversity of data within the memory. To tackle this challenge, we also propose FLAVA+, an extension of FLAVA, where its memory stores the selected patches instead of the raw data. We explicate the algorithm in *Section 4.3, Algorithm 1* in our latest revision. FLAVA+ accommodates an increased number of instances based on sampling ratios ($\rho\_{a}, \rho\_{v}$) to match the memory size of FLAVA. We update the experimental results on FLAVA+ in *Table 1, 4 7, and 8* in our latest revision.

We compare the VGGSound and AudioSet experiment results between FLAVA and FLAVA+ in the Table below. FLAVA+ outperforms FLAVA by a large margin. This significant improvement arises from the augmented volume of data in the rehearsal memory, allowing the memory to effectively maintain audio-video knowledge from past tasks, simultaneously enhancing data diversity in current task training.

The introduction of FLAVA+ represents a distinct and complementary direction to our initial contribution, demonstrating the efficacy of efficient rehearsal memory utilization. It represents our method's potential to efficiently utilize rehearsal memory.

VGGSound zero-shot retrieval task

| Method   | A$\rightarrow$V acc | A$\rightarrow$V fgt | V$\rightarrow$A acc | V$\rightarrow$A fgt |
|---------------|------------|-----------|-----------|-----------|
| FLAVA | 14.16 | **4.38** | 14.07 | 4.65 |
| FLAVA+ | **15.44** | 4.62 | **15.53** | **3.89** |

AudioSet zero-shot retrieval task

| Method   | A$\rightarrow$V acc | A$\rightarrow$V fgt | V$\rightarrow$A acc | V$\rightarrow$A fgt |
|---------------|------------|-----------|-----------|-----------|
| FLAVA | 12.35 | 6.95 | 11.29 | 8.38 |
| FLAVA+ | **15.26** | **5.74** | **14.47** | **6.70** |

Audiovisual classification tasks

| Method   | VGGSound acc | VGGSound fgt | AudioSet acc | AudioSet fgt |
|---------------|------------|-----------|------------|-----------|
| FLAVA | 58.65 | **0.64** | 65.66 | 0.89 |
| FLAVA+ | **59.07** | 0.80 | **66.05** | **0.78** |

---

We hope the above makes our contributions clear and corrects any factual misunderstandings. We thank you again for reviewing our work.

---

---

### Author Response · Authors · 2023-11-19

We sincerely appreciate your time and effort in reviewing our paper, as well as the constructive comments and valuable suggestions. **To convince the novelty of our work, we provide additional details and experiments in the submitted pdf file**, which include

- Enhanced explanation (*Section 4.2 and 4.3*)
- Suggestion of FLAVA+ for efficient rehearsal memory utilization (*Section 4.3, Algorithm 1*)
- Detail of the baseline methods (*Appendix A, Baselines*)
- Comparison between supervised continual learning and unsupervised continual learning (*Appendix B, Comparison between SCL setup and UCL setup*)
- Experiment results on the shuffled task order (*Appendix E, Shuffle task orders*)
- Downstream task performance on audiovisual event localization task (*Appendix E, Audiovisual Event Localization*)
- Visual results of sound source localization task (*Appendix E, Sound Source Localization*)
- Analysis of two key components of our approach with the modality gap (*Appendix G, paragraph 4-5*)

---

To highlight the strengths of our work, we have incorporated the Reviewers’ comments on our work, which we present succinctly below.

- The suggested AVM module demonstrates strong cross-modal localization abilities, particularly in localizing visual sound sources, which CAV-MAE struggled with. (*Reviewer nu6v*)
- Clear and meaningful motivation is presented, addressing critical challenges in continual audio-visual learning with illustrative visualizations that bring some insights. (*Reviewer Q9gf, hYwV, 4AjC*)
- Extensive experiments demonstrate the effectiveness of the proposed method, accompanied by comprehensive analysis of both the method and experiment. (*Reviewer hYwV, YJJY, 4AjC*)
-  The general structure is clear, and the method is simple in general (*Reviewer YJJY*)

---

To clarify the audio-video continual representation learning setting, we recap the major differences between our work and the audiovisual supervised continual learning.

> **Audiovisual supervised continual learning vs. Audiovisual continual representation learning**

The audiovisual supervised continual learning [1, 2] revolves around updating a model based on new audiovisual tasks (audiovisual classification task, audiovisual event localization, etc) that introduce unseen classes or domains while ensuring the preservation of past audiovisual knowledge within the model. This adaptation requires human-annotated labels, but acquiring high-quality human annotations is expensive and not scalable to large datasets.

Conversely, our audiovisual representation (unsupervised) continual learning focuses on **adapting new audiovisual representations from new datasets with altered data distributions while maintaining the learned audiovisual representations from previous tasks**. It continuously pre-trains the model with self-supervised loss terms and the pre-trained model is estimated on various audiovisual downstream tasks such as audiovisual zero-shot retrieval task, audiovisual classification task, audiovisual event localization, and sound source localization.

[1] Mo et al., Class-incremental grouping network for continual audio-visual learning, ICCV 2023.
[2] Pian et al., Audio-visual class-incremental learning, ICCV 2023

---

To understand how past data is retained and utilized during continual learning, we delve into the role of rehearsal memory. By examining recent studies, we aim to elucidate the rehearsal memory and the sampling mechanism used in the experiment.

> **The validity of rehearsal memory and how past instances are sampled**

In the realm of continual learning, rehearsal-based methods have been demonstrated to be more effective in challenging continual learning tasks. It assumes that the model can store a small subset of past task data during training on current task data [1, 2, 3, 4]. In our approach, same as other baseline methods, we leverage rehearsal memory to preserve this subset of past data.

During the process of sampling past instances from the memory, both our method and other baseline techniques utilize reservoir sampling [4]. Reservoir sampling serves as a strategy to maintain a fixed-size memory buffer while handling an indefinite-length data stream. It ensures that every sample in the buffer has an equal chance of being selected. The buffer randomly samples and replaces entries without giving priority to specific samples in the data stream [1].

[1] Arani et al., Learning Fast, Learning Slow: A General Continual Learning Method Based on Complementary Learning System, ICLR 2022
[2] Jeeveswaran et al., BiRT: Bio-inspired Replay in Vision Transformers for Continual Learning, ICML 2023
[3] Caccia et al., New Insights on Reducing Abrupt Representation Change in Online Continual Learning, ICLR 2022
[4] Jin et al., Gradient-based Editing of Memory Examples for Online Task-free Continual Learning, NeurIPS 2021
[5] Vitter., Random sampling with a reservoir, ACM 1985.

---

---

### Meta-Review · Area_Chair_aqcj · 2023-11-26

**Metareview:**

This paper proposes a lifelong audio-visual masked autoencoder model: FLAVA. to continually learn multimodal representations from audio-visual pairs at the scenaior where the distribution of the data continually shifts over time. The author pointed out two challenges in lifelong audio-visual representation learning: (1) sparse spatiotemporal correlation between the video-audio pairs, and 2) representational forgetting of audio-video relationships. FLAVA addresses these challenges by proposing two ideas: 1) a lightweight trainable audio-video matching (AVM) module, which performs attention-based localized alignment with cross-modal similarity, and 2) rank-based forget-robust multimodal patch selection. FLAVA outperforms SOTA continual learning methods on VGG-sound and AudioSet, showing its effectiveness.

Strength:
* The proposed two challenges are critical for the success of lifelong audio-visual representation learning. The paper brings some insights to the community. The work is reasonably motivated.
* The paper conducts reasonable experiments and ablation study to support the effectiveness of FLAVA in continual learning setting. The results seem to confirm the hypotheses that are laid out in the introduction.
* Implementation details are provided, which increases reproducability.

Weakness:
* My main concern about this work lies in the impact and its limited interests to ICLR community. Although I do think continual/life-long learning is an interesting problem, the area is relatively niche. This can be somewhat told ~half of the reviewers are not familiar with the topic or giving ambivalent ratings. Also, it's reasonable to define a problem where data distribution the models applied to changes over time. However, with the popularity of LLMs and self-supervised learning (SSL), we can see a trend of increasing model sizes along with pretraining data, and it's easy to imagine a world where most of available knowledge (available on Internet or proprietary database) will be used up for foundation model pretraining. There seems a disconnection between experiments conducted in this work and the trend of LLMs/SSL. Although reviewer nu6v might not be that familiar with lifelong learning, a very interesting question has indeen been raised, what if we just pool together data from all of the categories and learn a more general representation from the start. This is a reasonable question anyone in ICLR communtiy but not familiar with lifelong learning might have. I would encourage the authors try to answer such questions in future research, so that the work can induce more interests in the communtiy not just lifelong learning area.
* Sometimes signficant SOTA results trump the weakness of working on a niche area, since the results might indicate some more general breakthrough. However, I don't feel that after reading this paper (and the temperature in reviewer feedbacks).
* The writing could be improved. I understand one reviewer might find the method is simple in general and easy to follow. However, we always want more people to understand our papers for broader impact.

Even though reviewers seem less active in replying to authors, I had offline email exchanges with reviewers. They agreed (even the reviewers gave score 8) there is some concerns regarding the problem setting or data limitations. The reviewer who gave 8 to recognize the value and importance of the addressed research problem – self-supervised audio-visual continual learning. Valid results that demonstrate potential in this new direction.

**Justification For Why Not Higher Score:**

As mentioned in the weakness of this paper, this paper works on a niche area (lifelong learning). The results are not strong enough to justify a general interests to ICLR community. This can be told from the ambivalent review ratings. Another concerns are the writing. The authors seem have to explain the works with extra long comments, this indicates the authors can't convey their opinion and convince reviewers (or future readers) with limited paper length.

**Justification For Why Not Lower Score:**

N/A

---

### Decision · Program_Chairs · 2024-01-16

Reject